# Enhancer RNAs stimulate Pol II pause release by harnessing multivalent interactions to NELF

Vladyslava Gorbovytska[1,7], Seung-Kyoon Kim [2,3,7], Filiz Kuybu [1], Michael Götze[4], Dahun Um[2], Keunsoo Kang[5], Andreas Pittroff [1], Theresia Brennecke[1], Lisa-Marie Schneider [1], Alexander Leitner [4], Tae-Kyung Kim [2,6✉] & Claus-D. Kuhn [1✉]

Enhancer RNAs (eRNAs) are long non-coding RNAs that originate from enhancers. Although eRNA transcription is a canonical feature of activated enhancers, the molecular features required for eRNA function and the mechanism of how eRNAs impinge on target gene transcription have not been established. Thus, using eRNA-dependent RNA polymerase II (Pol II) pause release as a model, we here investigate the requirement of sequence, structure and length of eRNAs for their ability to stimulate Pol II pause release by detaching NELF from paused Pol II. We find eRNAs not to exert their function through common structural or sequence motifs. Instead, eRNAs that exhibit a length >200 nucleotides and that contain unpaired guanosines make multiple, allosteric contacts with NELF subunits -A and -E to trigger efficient NELF release. By revealing the molecular determinants of eRNA function, our study establishes eRNAs as an important player in Pol II pause release, and it provides new insight into the regulation of metazoan transcription.

[1] RNA Biochemistry, University of Bayreuth, Universitätsstrasse 30, 95447 Bayreuth, Germany. [2] Department of Life Sciences, Pohang University of Science and Technology (POSTECH), Pohang, Gyeongbuk 37673, Republic of Korea. [3] Department of Convergent Bioscience and Informatics, College of Bioscience and Biotechnology, Chungnam National University, Daejeon 34134, Republic of Korea. [4] Department of Biology, Institute of Molecular Systems Biology, ETH Zurich, 8093 Zurich, Switzerland. [5] Department of Microbiology, Dankook University, Cheonan 31116, Republic of Korea. [6] Institute of Convergence Science, Yonsei University, Seoul 03722, Republic of Korea. [7] These authors contributed equally: Vladyslava Gorbovytska, Seung-Kyoon Kim. ✉email: tkkim@postech.ac.kr; claus.kuhn@uni-bayreuth.de

Enhancers are DNA elements that direct metazoan tissue- and stimulus-specific gene expression by serving as transcription factor binding platforms. At the primary sequence level enhancers can be distant from their target genes; however, to achieve regulation, enhancers and promoters are brought into proximity by chromatin looping[1]. The regulatory information encoded in the composition of the enhancer-bound transcription factors is transmitted to the core transcription machinery via Mediator[2]. In addition to orchestrating transcription factor binding, enhancers themselves are transcribed bidirectionally by RNA polymerase II (Pol II), resulting in the production of large numbers of long non-coding RNAs (lncRNAs) that are termed enhancer RNAs (eRNAs)[3,4]. In general, eRNA expression has been shown to correlate with target gene activation, but whether eRNA molecules themselves are functional or whether it is the act of transcription that boosts enhancer activity is currently under debate[5–8].

Although considerable research has focused on histone modification and transcription factor binding profiles at enhancers, recent studies show that eRNA levels and the extent of their bidirectional transcription are most likely a better predictor of enhancer activity[9,10]. Despite their inherent instability[11], eRNAs were detected in different tissues and following numerous stimuli[3,4,12–14], and they were shown to correlate with the activation of diverse sets of target genes. This data strongly supports a direct, functional role for eRNAs, but their mechanistic role remains largely unexplained. This is in part due to the fact that eRNAs, similar to other lncRNAs, possess diverse mechanisms through which they exert their function. For example, in cell culture experiments eRNAs were shown to increase transcription rates by stimulating the histone acetyltransferase activity of CBP (CREB-binding protein)[15]. Furthermore, eRNAs cause the increased retention of transcription factors, such as YY1 and BRD4, at gene regulatory elements, thereby modulating their biological impact[16,17]. In earlier work, we characterized the impact of eRNAs on the stimulus-induced expression of immediate early genes (IEGs) in mouse primary neurons[18]. Our results suggested that neuronal eRNAs trigger the release of NELF from paused Pol II at IEGs, whereby they presumably facilitated the transition of Pol II into productive elongation. To achieve NELF release, we speculated that eRNAs might compete with nascent mRNAs for binding to the negative elongation factor (NELF). However, the molecular mechanism of how eRNAs affect paused Pol II and the structural and sequence characteristics that enable eRNA-driven NELF release remained unknown.

Here we answer these unsolved questions and reveal an unexpected allosteric mechanism through which eRNAs stimulate PoI II pause release. First, we characterize both the sequences and the structures of dozens of neuronal eRNAs by a combination of Exo-seq (5′-end RNA-seq) and SHAPE-MaP (selective 2′-hydroxyl acylation analyzed by primer extension and mutational profiling). Second, we then use electrophoretic mobility shift assays (EMSAs) to show that eRNAs efficiently trigger NELF release from the paused elongation complex (PEC) in vitro, but only if eRNAs are sufficiently long and contain several unpaired guanosines. Third, the use of protein-RNA crosslinking coupled to mass spectrometry on eRNA-bound NELF and paused elongation complexes uncovers that the length requirement for eRNA functionality stems from multiple eRNA binding sites along NELF subunits -A and -E. In order to efficiently detach NELF from Pol II, eRNAs must bind at least some of these sites simultaneously. These in vitro findings are supported by NELF-E-directed eCLIP-seq (enhanced UV crosslinking and immunoprecipitation sequencing) experiments in mouse primary neurons, which demonstrate that NELF is directly contacted by enhancer-derived eRNAs. By utilizing a reconstituted pause release assay, we further demonstrate that eRNA-driven NELF release results in transcription activation through the more efficient release of Pol II from the paused state. Complementing these in vitro findings once more in vivo, we find that NELF binding levels correlate with rapid and efficient transcriptional elongation in response to neuronal stimulation, suggesting that the eRNA-dependent release of NELF from paused Pol II could play a role in activity-induced transcription. To our knowledge, this study represents the first report that mechanistically links eRNAs to the core Pol II transcription machinery. Furthermore, by revealing the detailed molecular determinants that enable eRNAs to stimulate Pol II pause release, our study establishes the functional capacity of eRNAs in the regulation of metazoan transcription.

## Results

**Exo-seq allows for the assignment of eRNA transcription start sites with single-nucleotide precision.** To begin to decipher the molecular mode of action of eRNAs in mouse primary neurons, it was imperative to first determine their exact sequences. We thus performed global run-on sequencing (GRO-seq)[19] before and after neuronal stimulation by KCl treatment and profiled all resulting nascent RNAs (see Methods). We defined transcription units de novo from GRO-seq reads and assigned them either to annotated genes or to eRNAs based on the overlap between intergenically-located transcript units and regions enriched for histone H3 lysine 27 acetylation (H3K27ac), a histone mark previously found to label active enhancers[20]. A total of 9,028 annotated genes were overlapped with the transcript units defined de novo from GRO-seq reads. In addition, we identified a total of 1226 intergenic eRNA transcription units, of which 252 were activity-induced, as defined by a >1.5-fold increase of eRNA GRO-seq signal at any time point after KCl stimulation (Supplementary Data 1). As our GRO-seq data did not allow us to unambiguously determine the 5′-ends of eRNAs, we performed the Exo-seq protocol in addition, a technique dedicated to the assignment of RNA 5′-ends[21]. Indeed, after applying the program TSScall[10] to our Exo-seq data, we could determine the 5′-ends of eRNAs with single-nucleotide precision. We then intersected the list of eRNA TSSs (eTSS = enhancer transcription start site) with our GRO-seq-based list of 1226 intergenic eRNA transcription units to find 304 of them (281 for replicate 2) to exhibit well-defined 5′-ends (>20 reads per eTSS). Eighty-six of these eRNAs (79 for replicate 2) stem from activity-induced enhancers. Overall, the 5′-ends of both eRNAs and mRNAs - as defined by Exo-seq - were well-correlated with nascent transcripts detected by GRO-seq, as shown for two enhancer loci, Nr4a1 and Arc (Fig. 1a, b). In corroborating our analysis, we detected eRNA TSSs of IEGs such as Arc, Nr4a1, Junb, c-Fos (enhancers e1, e2, and e5), and Fosb, for some of which eRNA expression had been reported before[3]. After closer inspection of the Exo-seq reads at the selected eTSSs, we excluded all sites with pervasive or convergent transcription and compiled a final list of 33 high-quality eRNA candidates, including all the abovementioned IEG eRNAs. Seven of the 33 eRNA candidates, among them Junb and Nr4a1, showed distinct, likely alternative eTSSs. For these eRNAs we, therefore, included two separate eRNAs (termed (a) and (b)). In total, this resulted in a test set of 39 eRNAs (Fig. 1b and Supplementary Data 2).

**RNA structure probing uncovers that eRNAs do not share common structural motifs.** As we previously showed that target gene induction depends on eRNAs[18], yet we were unable to identify sequence motifs that would explain eRNA function, we speculated that eRNA might possess specific structural features

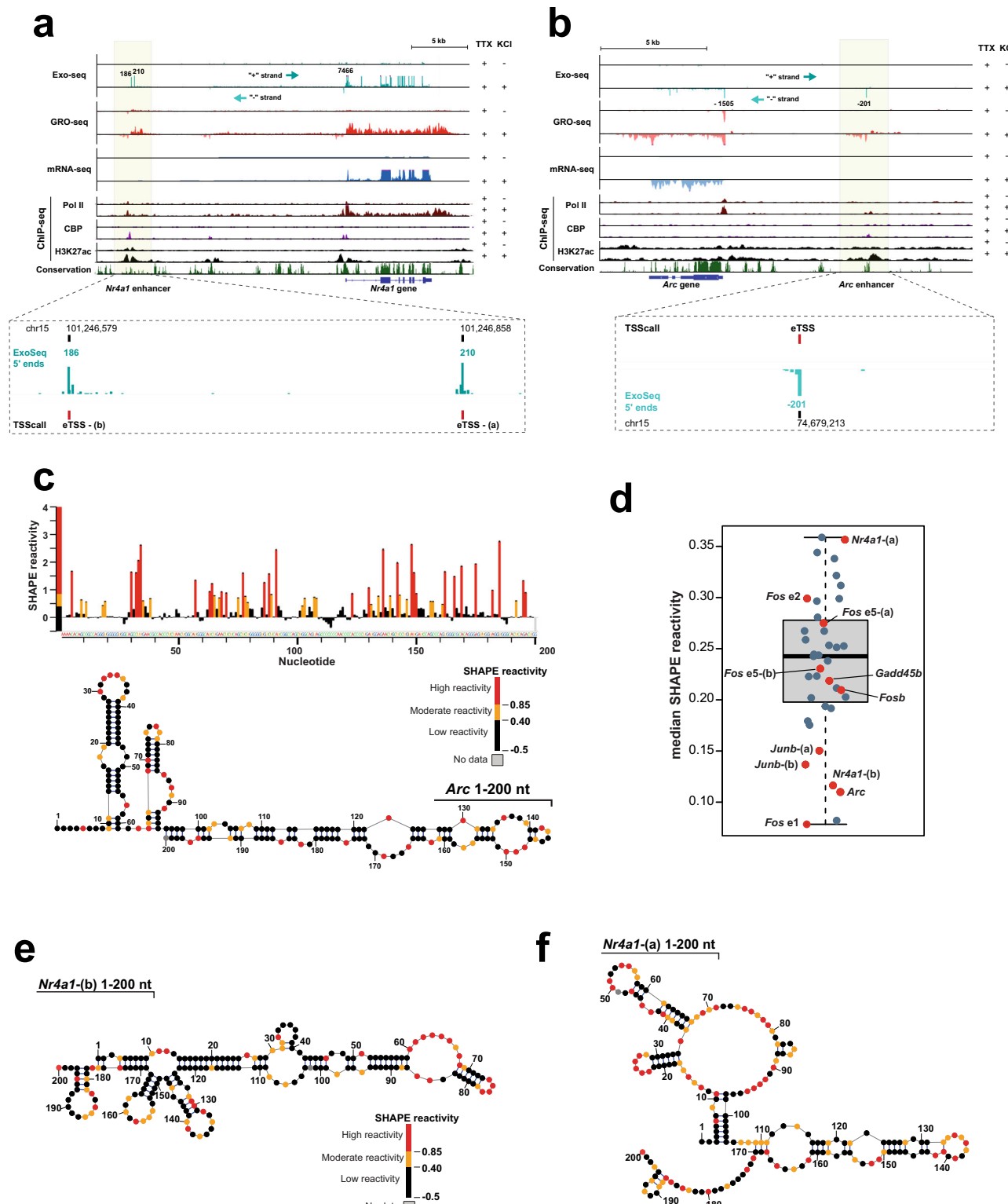

that are important for their function. We thus set out to determine the secondary structures of all 39 eRNA candidates, for which we had identified the precise eTSS (Fig. 1a, b and Supplementary Data 2). Although eRNAs are reported to have a median length of ~1 kb[11], we decided to focus our secondary structure analysis on the 5'-terminal 200 nt of all eRNAs for the following reasons: (1) While long-range RNA–RNA interactions exist, RNA base pairs in diverse lncRNAs and mRNAs predominantly form locally (<200 bp)[22,23], and (2) eRNAs are

capped and thus possess defined 5'-ends, whereas their 3'-ends are not polyadenylated and therefore prone to exosome-mediated decay[24]. The functionally relevant eRNA parts are therefore likely confined to their 5'-ends. To map all 39 eRNA structures simultaneously, we turned to chemical probing of RNA secondary structure read out by next-generation sequencing. Specifically, we performed the SHAPE-MaP protocol with in vitro transcribed[25], monodisperse eRNA (1–200) fragments (Supplementary Fig. 1a) by use of 1M7 (1-methyl-7-nitroisatoic anhydride) as a modifying

**Fig. 1 Secondary structure mapping of selected mouse neuronal enhancer RNAs.** The mouse *Nr4a1* (**a**) and *Arc* (**b**) gene and enhancer loci in genome browser view. Exo-, GRO-, and mRNA-seq signal from KCl-stimulated mouse cortical neurons is plotted alongside Pol II (8WG16 antibody), CBP, and H3K27ac ChIP-Seq data from publicly available sources[3, 20]. A zoom-in view of the Exo-seq data attests to the single-nucleotide resolution in determining eRNA TSSs (eTSS). The enhancer locus of *Nr4a1* revealed two distinct TSS sites (eTSS-(a) and eTSS-(b)). **c** SHAPE reactivity profile of *Arc* eRNA (1–200), as computed by using *Shapemapper2*[28]. Using the experimentally determined SHAPE reactivities as restraints, the corresponding secondary structure (shown below) was generated with *RNAStructure*[29]. Nucleotides are shown as circles and are colored according to their SHAPE reactivity (see the reactivity scale bar above the structure). **d** The distribution of median SHAPE reactivities for all individual 39 eRNAs that were subjected to SHAPE-MaP is summarized in a boxplot, showing individual data points ($n = 39$). Data points corresponding to prominent immediate early genes are marked in red. eRNAs that exhibit low median SHAPE reactivities (<0.1) are considered to be highly structured. **e, f** Secondary structures of variant (b) and variant (a) of *Nr4a1* eRNAs (1–200). Determined and plotted as in (**c**). Source data for (**c, d**) are provided in a Source Data file.

agent[26,27]. Following sequencing of our SHAPE-MaP libraries, we calculated mutation rates and SHAPE reactivities using ShapeMapper2[28] and computed SHAPE-restraint RNA secondary structures using RNAstructure[29] (Supplementary Fig. 1b, c and Supplementary Data 3). Unfortunately, eRNAs in cortical neurons proved refractory to sufficient levels of 1M7 modification, thus precluding us from confirming our data in vivo. While we found the enhancer RNA of the IEG *Arc* (*Arc* eRNA (1–200)) to predominantly consisting of highly structured parts, as reflected by its low median SHAPE reactivity (Fig. 1c, d), this is not the case for all eRNAs. For them, median SHAPE reactivities range from 0.08 for *Fos* e1 eRNA (1–200) to 0.36 for *Nr4a1*-(a) (1–200) eRNA, respectively (Fig. 1d). This broad distribution of eRNA structures is best compared to the RNA structure-ome of *E. coli*, whose SHAPE reactivities were shown to range from <0.1 for tRNAs and abundant non-coding RNAs all the way to 0.15–0.35 for coding mRNAs[23]. Interestingly, five of the top six most structured eRNAs in our dataset (median SHAPE reactivity <0.15) belong to IEGs (*Arc*, *Nr4a1*-(b), *Junb*, and *Fos* e1) (Fig. 1d). However, as eRNAs of other prominent IEGs, such as *Nr4a1*-(a), *Gadd45b,* and *Fosb*, are highly flexible (Fig. 1d), we conclude that neuronal eRNAs are not necessarily highly structured. That being said, our experimental data strongly argues against a general lack of eRNA structuredness, as was reported before using RNA fold predictions only[11]. In contrast, we find eRNAs to populate a wide range of structural spaces without any common structural motifs that might explain their function.

**NELF detachment from the paused elongation complex is markedly dependent on eRNA length**. The diverse nature of their structures did not offer a path forward in deciphering the mechanistic impact of eRNAs on paused Pol II (Fig. 1 and Supplementary Data 3). To remedy this, we reconstituted the mammalian paused elongation complex (PEC) in vitro[30,31]. To then study the effect of eRNAs on the PEC, we used electrophoretic mobility shift assays (EMSAs), as this technique offers both high sensitivity for detecting protein–RNA interactions and allows for the study of large macromolecular complexes such as the PEC (which has a size of 0.9 MDa). The PEC was stepwise assembled on a synthetic transcription bubble (Fig. 2a) that contained a 5′-³²P-labeled "nascent" RNA with a GC content of 28% and a length of 25 nt[30,31]. Our EMSA setup confirmed that nascent RNA needs to be >22 nt in length to allow for PEC assembly (Supplementary Fig. 2a)[32]. Following PEC assembly, we added the 5′-terminal 200 nt fragments of *Arc*, *Nr4a1*-(a), or *Nr4a1*-(b) eRNAs, respectively, and analyzed the mobility of formed complexes (Fig. 2b–d and Supplementary Fig. 2b). Intriguingly, at eRNA concentrations only about equimolar to PEC concentrations, we already observed NELF dissociation from the PEC (top gels in Fig. 2b–d). In contrast to NELF, DSIF (DRB sensitivity inducing factor) remained untouched and stably bound to Pol II after eRNA addition—even in presence of a tenfold excess of eRNA (top gels in Fig. 2b–d). Our results are in

agreement with established knowledge on pause release and they assert the validity of our assay setup for studying pausing in vitro[33–35]. We verified the presence of NELF and DSIF in our EMSA gels by performing supershift assays (Supplementary Fig. 2c). Moreover, we excluded any effect that our choice of nascent RNA might have on eRNA activity by performing EMSA experiments with a nascent RNA that had a GC content of 48% (Supplementary Fig. 2d). Next, to uncover the part or parts of each eRNA that are responsible for NELF dissociation, we shortened the eRNAs to their 1–100 and 1–50 variants (middle and lower gels in Fig. 2b–d). When using these shorter variants, we were very surprised to find that, whereas *Arc* eRNA and *Nr4a1*-(b) (1–100) fragments retained some of their NELF-dissociation potentials, both 1–50 fragments were unable to induce NELF dissociation from the PEC, even in 18-fold molar excess (middle and lower gels in Fig. 2b–d). To assess the dissociation potential of different eRNA fragments towards NELF, we quantified the appearance of the Pol II-DSIF complex following NELF dissociation (Supplementary Fig. 2c). Our analysis confirmed that eRNA (1–100) fragments exhibit a >10x higher apparent $K_d$ compared to their 1–200 siblings and, thus, possess a clearly reduced ability to displace NELF. In contrast to the *Nr4a1*-(b) (1–100) fragment, the *Nr4a1*-(a) (1–100) fragment was considerably more effective in dissociating NELF ($K_d = 2.02 \mu M$ vs. $0.14 \mu M$) (Fig. 2c–e). Further, the *Nr4a1*-(a) 1–50 fragment also displayed a mild NELF dissociation effect with a $K_d$ of 9.70 μM. Interestingly, while *Arc* and *Nr4a1*-(b) are highly structured with prominent double-stranded regions (Fig. 1d, e), *Nr4a1*-(a) exhibits a more flexible structure with long single-stranded regions (Fig. 1f). This hints at the possibility that RNA flexibility might facilitate the observed dissociative effect of eRNAs on the PEC. Taken together, we found eRNA-induced NELF dissociation to be critically dependent on eRNA length. However, due to their divergent structures and sequences, these results did not reveal the mechanism of eRNA-induced NELF dissociation.

**Unpaired guanosines are critical for the dissociative effect of eRNAs on the paused elongation complex**. To further decode the functional role of eRNA structure, we analyzed the almost entirely double-stranded *Arc* eRNA (96–200) fragment, a *Nr4a1*-(a) mutant (*Nr4a1*-(a) Δloop 12 mutant 1–102), in which single-stranded regions had been deleted, and an *Arc* mutant that lacked almost all secondary structure (*Arc* Δstem mutant 1–100) (Supplementary Fig. 2e). Despite their fundamentally different secondary structure, all mutants were able to detach NELF from Pol II equally well, clearly demonstrating that eRNA function does not depend on structure alone (Fig. 2f). Thus, to simplify the sequence and structure space of the tested RNAs, we turned to synthetic, low complexity RNAs and measured their effect on NELF dissociation. Surprisingly, none of the tested single-stranded RNAs that lack guanosines showed the ability to dissociate NELF from the PEC (Fig. 2g). In contrast, despite their short length G-containing poly(GU)$_{40}$ RNA and poly(GA)$_{48}$ RNA

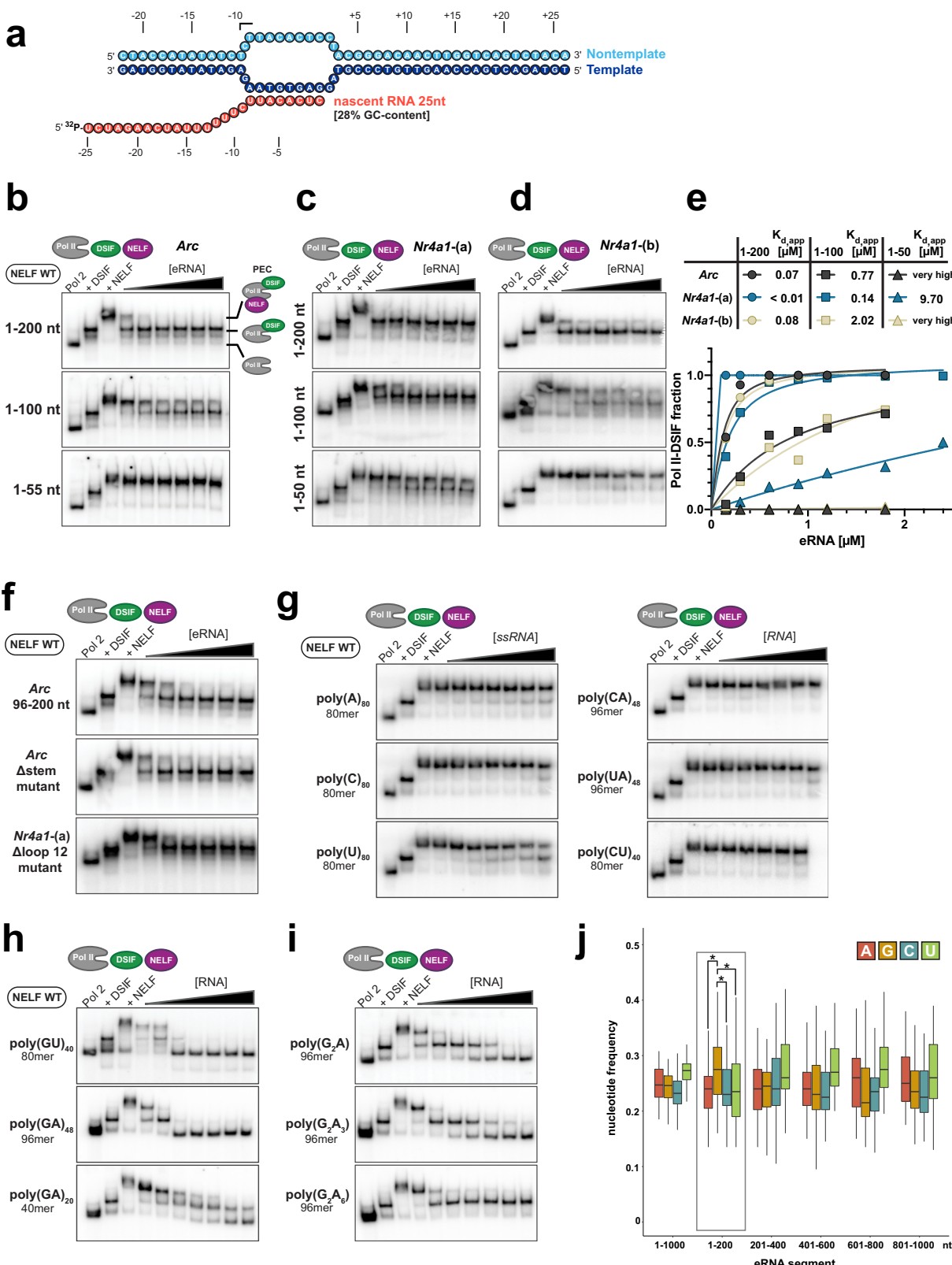

were able to displace NELF at low RNA concentrations and even led to DSIF dissociation at higher RNA concentrations (Fig. 2h). As observed before (Fig. 2b–e), we found this effect to be dependent on RNA length. Next, we asked which "density" of guanosines is sufficient to trigger NELF release, and thus we tested synthetic 96-mer poly(G₂A) RNA, poly(G₂A₃) RNA, and poly(G₂A₆) RNA. As for poly(GU)₄₀ RNA and poly(GA)₄₈ RNA,

we found all of them to be able to fully dissociate NELF. However, at higher RNA concentrations poly(G₂A₆) RNA, bearing the widest G spacing did not cause DSIF dissociation (Fig. 2i). In summary, RNA-driven NELF dissociation from the PEC seems critically dependent on unpaired guanosines. In building on this, we were able to rationalize our prior data on eRNA fragments (Fig. 2b–e). There, we only found the short fragment of Nr4a1-(a)

**Fig. 2 eRNAs trigger NELF release from the paused elongation complex in a length- and the guanosine-dependent manner in vitro. a** Nucleic acid scaffold used for radioactive electrophoretic mobility shift assays (EMSAs). Template DNA (49 nt) is shown in dark blue, fully-complementary non-template DNA (49 nt) is shown in light blue, and nascent RNA (25 nt) is shown in red. **b** EMSAs demonstrate the ability of *Arc* eRNA variants to release NELF from the PEC (paused elongation complex). Results for *Arc* eRNAs of three different lengths are shown. Top: Arc eRNA (1–200), middle: Arc eRNA (1–100), bottom: Arc eRNA (1–50). **c** EMSAs with *Nr4a1*-(a) variants, experimental setup as for (**b**). **d** EMSAs with *Nr4a1*-(b) variants, experimental setup as for (**b**). For all EMSAs the PEC was assembled on the nucleic acid scaffold shown in (**a**) using 0.8 pmol $^{32}$P-labeled RNA. The Pol II-DSIF complex was then formed by using 1.2 pmol Pol II and 2.4 pmol DSIF. Subsequently, 1.2 pmol NELF were added to form the PEC (final concentration 0.1 μM). The addition of increasing amounts of eRNAs (final concentrations: 0.15, 0.3, 0.6, 0.9, 1.2 and 1.8 μM) triggered detachment of NELF from the PEC. **e** NELF release from the PEC is quantified by plotting the formation of the Pol II-DSIF complex against the eRNA concentration (mean of two experimental replicates). The resulting pseudo-binding curve was fitted using a single-site binding model, apparent $K_d$ values are indicated. **f** EMSAs performed with the highly-structured *Arc* eRNA (96–200) fragment (104 nt), a mostly single-stranded *Arc* eRNA Δstem mutant (100 not), and a *Nr4a1*-(a) Δloop 12 mutant (102 nt). The *Nr4a1*-(a) Δloop 12 mutant lacks prominent single-stranded regions present in *Nr4a1*-(a) eRNA (see Supplementary Fig. 2d). **g** EMSAs performed with single-stranded, low complexity RNAs, such as poly(A), poly(C), poly(U), poly(CA), poly(UA), and poly(CU) RNAs. **h** EMSAs were performed with 80 nt long poly(GU) RNA and different lengths of poly(GA) RNA (40 and 96 nt). **i** EMSAs were performed with 96 nt long poly($G_2A$), poly($G_2A_3$), and poly($G_2A_6$) RNA. EMSAs shown in (**f–i**) were carried out as described for (**b–d**), except for an additional RNA concentration step of 2.4 μM in (**g–i**). **j** Nucleotide frequency plot for all eRNAs, whose secondary structure was determined by SHAPE-MaP (Fig. 1d and Supplementary Data 2). To prevent biases only a single TSS was utilized for this analysis in the case eRNAs possessed two alternative TSSs within a distance of <40 nt. eRNA sequences were extended to 1 kb and divided into bins of 200 nt. Guanosines are only significantly overrepresented (as determined by a pairwise *t*-test with *p* values (A/G) = 0.018; (C/G) = 0.020; (U/G) = 0.036) in the 5′-terminal 200 nt. Source data for (**b–j**) are provided in a Source Data file.

eRNA to induce the partial release of NELF (Fig. 2c, e). This effect can now be explained by focusing on the number of unpaired guanosines that do not form stable base pairs with cytidines. Despite the fact that, overall, *Arc* eRNA (1–50) and *Nr4a1*-(b) eRNA (1–50) contain more guanosines, these are mostly paired with cytidines, which is not the case for *Nr4a1*-(a) (1–50) (Supplementary Fig. 2f). To experimentally substantiate the functional role of unpaired guanosines, we generated G-less *Nr4a1*-(a) eRNA (1–50) and (1–100) variants, in which we replaced all guanosines with adenosine (G-to-A mutant) or cytidine (G-to-C mutant). Indeed, the G-less mutants showed a strongly reduced ability to trigger NELF release, as reflected by 10x higher apparent $K_d$ values (Supplementary Fig. 2g, h). Interestingly, the restoration of two guanosines in the middle (2 G middle) or three guanosines at the 3′- or at the 5′-end of the G-less mutant (3 G at 5′-end or 3 G at 3′-end) enhanced eRNA-driven NELF detachment 2.5 to 5-fold, respectively ($K_d = 0.38$ μM for 3 G at the 5′-end and $K_d = 0.32$ μM for 3 G at the 3′-end) (Supplementary Fig. 2g, h). However, the additional restoration of three guanosines at the RNA's 5′- or 3′-end (six guanosines in total) did not facilitate NELF dissociation any further. As the apparent $K_d$ for NELF release by WT *Nr4a1*-(a) (1–100) eRNA is lower the any of the mutants with restored guanosines (0.14 μM, Fig. 2e), our data argue that potent eRNAs must comprise several widely-spaced unpaired guanosines and not only one cluster. Last, we sought to support our in vitro data with a more global analysis of the prevalence of guanosine in eRNAs. Thus, we calculated the nucleotide distribution across our set of 39 neuronal eRNAs (Supplementary Data 2). Indeed, we found guanosine to be overrepresented in their first 200 nt in a statistically significant manner (Fig. 2j), but not amongst the larger collection of eRNAs (Supplementary Data 1). This suggests that only a subset of eRNAs (e.g., activity-induced eRNAs) possess elevated levels of guanosine within their first 200 nt. Intriguingly, we only find guanosine to be overrepresented, which sets our finding apart from 5′-UTRs of coding genes. These are highly structured and hence show a higher content of both guanosine and cytidine[36].

**eRNAs bind to a positive patch on NELF-C and to the NELF-E RRM domain.** For eRNAs to displace NELF from the PEC they must very likely directly contact NELF, a complex of four proteins termed NELF-A, -B, -C, and -E[37]. The RRM (RNA recognition motif) domain of NELF-E was shown to bind both single-

stranded, as well as structured RNAs, such as the HIV TAR element, in vitro[38–41]. Previously, we had shown that the RRM domain is essential for eRNA function in vivo[18]. Moreover, RNA binding studies with recombinant NELF had identified two additional RNA binding sites on NELF-A/C and on NELF-B[42]. Whereas the NELF-E RRM domain might be involved in contacting nascent RNA during pausing[40], the biological relevance of the other two RNA binding interfaces has not been established. To decipher how eRNAs are able to dissociate NELF from paused Pol II, we, therefore, purified a NELF variant that lacked the RRM domain of NELF-E (NELFΔRRM) and examined its dissociation from the PEC after eRNA addition. All mutant PEC complexes could be assembled as efficiently as wild-type complexes (Supplementary Fig. 3a), but we observed significantly diminished NELF dissociation induced by *Arc* eRNA (1–200 and 1–100) fragments ($K_d$s of 0.33 μM and 7.8 μM) (Fig. 3c), suggesting that the RRM domain of NELF-E is directly involved in eRNA-driven NELF release (Fig. 3a). We detected an analogous effect for *Nr4a1*-(b) eRNA fragments (Supplementary Fig. 3b). However, the absence of the NELF-E RRM domain did not fully abolish NELF release, a finding which was particularly obvious for the eRNA 1–200 nt variants and for *Nr4a1*-(a) (1–100) eRNA (Fig. 3a, c and Supplementary Fig. 3b). This demonstrates that eRNA-driven dissociation of NELF from the PEC does not solely depend on the RRM domain. It is also in line with the observed dependency of NELF dissociation on eRNA length. To uncover the missing eRNA binding site on NELF, we next turned to a NELF mutant, in which one of the two additional RNA binding sites, a positively charged surface patch on NELF-A/C had been mutated to uncharged residues (NELF-C patch mutant)[42]. Using this mutant, we found dramatically diminished eRNA-driven NELF dissociation from the PEC (Fig. 3b, c and Supplementary Fig. 3c). More specifically, *Arc* and both *Nr4a1* eRNA (1–100) fragments hardly induced NELF detachment, whereas the 1–200 nt fragments did, but to a much lesser extent as compared to the NELFΔRRM variant. Last, we combined both mutants to perform EMSAs with a NELF variant that lacked both the NELF-E RRM domain and whose NELF-C patch was mutated (Fig. 3d–f). Intriguingly, this double mutant has entirely lost its ability to dissociate from the PEC, even if eRNA (1–200) fragments were added in large molar excess. Thus, we conclude that both the NELF-E RRM domain and the positively charged surface patches on the NELF AC-lobe are essential to enable eRNA-induced NELF dissociation from the PEC (Fig. 3).

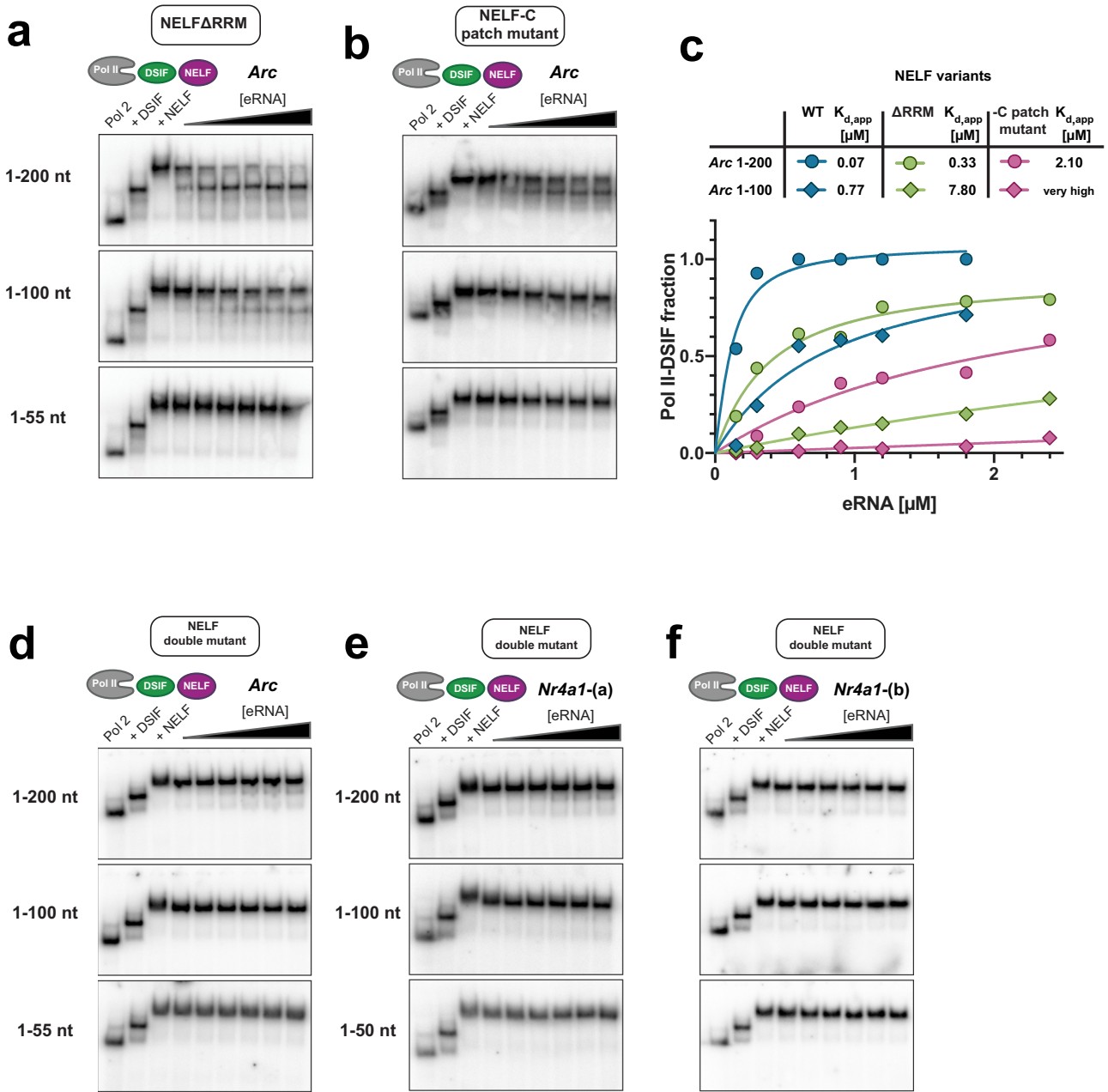

**Fig. 3 Positive patches on NELF-AC and the NELF-E RRM domain are both involved in eRNA-driven NELF dissociation.** EMSAs carried out with PEC variants that comprise **a** the NELF-ABC-EΔRRM complex (NELF ΔRRM), or **b** a mutant NELF–ABCE complex, in which lysines and arginines situated in positively charged surface patches on NELF-C are mutated to uncharged residues (R291Q, K315M, K371M, K372M, K374M, K384M, K388M, R419Q, and R506Q) (NELF-C patch mutant)[42], **c** Quantification of NELF release using data from **a**, **b**. Data for wild-type NELF (Fig. 2b) are shown to aid comparison. Binding curves were plotted and fitted as in Fig. 2e. **d** EMSA using a NELF double mutant (NELF patch mutant + NELF-E ΔRRM). The three NELF mutants utilized in these experiments were described previously[42]. All assays were performed as in Fig. 2b–d, using *Arc* eRNAs (1–50), (1–100), or (1–200). In **e**, **f** EMSAs of the NELF double mutant using *Nr4a1*-(a) and *Nr4a1*-(b) eRNA fragments are shown, respectively. Source data for (**a**–**f**) are provided in a Source Data file.

**An extended RNA–protein interaction network between eRNAs and NELF subunits -E and -A enables eRNA-driven NELF dissociation from the PEC.** To corroborate the presence of at least two distinct eRNA binding sites on NELF and to more exactly determine these sites, we sought to apply protein–RNA crosslinking coupled to tandem mass spectrometry to both eRNA-PEC and eRNA-NELF complexes. To provide evidence for the formation of specific eRNA-NELF complexes, the likely product of eRNA-driven NELF release from the PEC, we utilized analytical size-exclusion chromatography. Indeed, our data confirmed the

formation of such complexes (Supplementary Fig. 4a). In contrast, we were unable to form such specific protein–RNA complexes with the NELF double mutant complex (NELF-C patch mutant + deletion of NELF-E RRM domain, Supplementary Fig. 4b). Collectively, these results confirm that eRNAs are able to form stable and specific complexes with the isolated NELF complex. Based on this finding we utilized UV light of 365 and 254 nm, respectively, to induce cross-links between the NELF complex and bound eRNAs that had either been labeled with 4-thiouridine (4SU) or that had been left unlabeled (Supplementary Fig. 4c–e). In total, we collected

protein–RNA crosslinking data of the NELF complex bound to different eRNAs, as well as a repetitive poly $(GU)_{40}$-RNA, under six experimental conditions (Supplementary Fig. 4c). Importantly, all tested RNAs are highly potent in dissociating NELF from the PEC (Fig. 2b, c, h). We obtained high-quality crosslinking data under all conditions, in particular for the 4SU-labeled eRNAs (Supplementary Data 4, 5 and Supplementary Fig. 4d). The estimated false discovery rates for 4SU-labeled eRNAs and unlabeled eRNAs were ≤0.5 and ≤5%, respectively, for the selected settings. To corroborate these high-coverage crosslinking data in the context of the entire PEC, we also performed two crosslinking experiments with a purified PEC and both 4SU-labeled *Arc* eRNA (1–200) and *Nr4a1*-(a) eRNA (1–200) (Supplementary Fig. 4f and Supplementary Data 6, 7; Methods). The resulting crosslinking data are part of Fig. 4a–e, Supplementary Fig. 4f, and Supplementary Data 7, however, as they overwhelmingly agree with our data on the isolated NELF complex, we will not discuss them separately.

Remarkably, the vast majority of all protein–RNA crosslinks are restricted to both NELF-E (40–65% of all crosslinks) and NELF-A (20–55% of all crosslinks) (Fig. 4a upper panel and Supplementary Data 5). This crosslinking preference confirms our EMSA results (Fig. 3), and it is entirely consistent with an SDS-PAGE of a UV crosslinking experiment of NELF and *Nr4a1*-(a) (1–100) at 365 nm (Fig. 4a bottom panel and Supplementary Fig. 4e). A more detailed analysis of the RNA cross-link positions on NELF-E reveals distinct crosslinks or clusters thereof that are found under all tested experimental conditions (Fig. 4b and Supplementary Fig. 5a). Thus, our data highly likely reveal specific RNA binding sites on NELF that, at least partially, need to be contacted by eRNAs for the specific removal of NELF from the PEC. The NELF-E tentacle (residues 138–380 including the NELF-E RRM domain) harbors about half of all NELF-E-RNA crosslinks (Fig. 4c and Supplementary Fig. 5b). Amongst those, the observed crosslinks to G261-Y267 and to F299-V300 coincide with the conserved ribonucleoprotein motifs RNP2 ($L^{265}$YVY) and RNP1 ($CA^{299}$FV) that are part of the NELF-E RRM domain[38]. Other prominent RNA cross-link positions on NELF-E that are located close to its C-terminus (A330-Q333; W345; Y367-Y372) are congruent with previous publications that reported that the region C-terminal to the NELF-E RRM domain contributes to RNA binding[38,39]. The other half of all NELF-E RNA crosslinks, comprising the two distinct crosslinking clusters G94-P109 and K130-F136, are located in the unstructured N-terminal region of NELF-E (amino acids 1–138). This demonstrates that the RRM domain is not the sole eRNA binding site on NELF-E (Fig. 4b, c). Regarding NELF-A, almost all RNA crosslinks to this subunit (Q220-F232; K255-D258; L262-G267; K276-A280; E284-K288) are located in its largely disordered tentacle region (residues 188–528) (Fig. 4d, e and Supplementary Fig. 5c, d)[30].

Further inspection of the entirety of RNA crosslinks to NELF-E and NELF-A revealed that their positions overlap very well with the observed NELF crosslinks to Pol II and to the SPT5 subunit of DSIF (Fig. 4b, d)[30]. In particular, the NELF-A tentacle, itself heavily crosslinked to RNA (Fig. 4d, e), was shown to cross-link to Pol II and to be indispensable for the formation of the PEC (Fig. 4d)[30,43]. Moreover, we detected K371, a residue previously found to be crosslinked to both Pol II and SPT5, to be part of a cluster of RNA-crosslinked residues on NELF-A (P362-K371) (Fig. 4d). The striking agreement between RNA-NELF crosslinks and protein–protein crosslinks between the Pol II-DSIF complex and the unstructured regions of NELF-E and -A immediately suggests that eRNA binding to NELF might interfere with NELF binding to the Pol II-DSIF complex, this way contributing to NELF dissociation. Last, we also noticed a striking overlap between the eRNA cross-link positions on NELF-A and

experimentally confirmed P-TEFb phosphorylation sites on this NELF subunit that were shown to be critical for Pol II pause release (T157, T277, S363)[44]. In particular, T157 caught our interest, as this confirmed P-TEFb target site is located in patch 4, one of the positively charged surface patches that was shown to possess the capacity to bind RNA[42] and that we had found important for eRNA-driven NELF detachment from the PEC (Fig. 3b–f). Indeed, our crosslinking data encompass RNA crosslinks to the positively charged NELF surface patch 1 (RNA crosslinks to NELF-C residues P307, A308, M416-D321), to patch 3 (RNA crosslinks to NELF-C residues S378 and K388), and to patch 4 (RNA crosslinks to NELF-A residues E132, S134, P138, N147, L156, T157, V160, K161, K168, and an RNA cross-link to NELF-C residue F420) (Fig. 5b and Supplementary Data 4). The cryo-EM structure of the PEC T157 is surrounded by lysines that belong to NELF surface patch 4[30]. We, therefore, hypothesize that phosphorylation of T157 by P-TEFb might trigger a conformational rearrangement of the aforementioned lysines, an event which might alleviate NELF release from the PEC. A similar conformational change in NELF might be induced by eRNA binding to NELF-A in the vicinity of T157, thus offering a possibility for Pol II pause release under bypassing P-TEFb activity. We want to note that, overall, we did not observe as many RNA crosslinks to the positively charged patches on the NELF-A/C lobe than to the unstructured tentacles of NELF-A and NELF-E (Fig. 4b–e). On the one hand, this is likely due to more efficient RNA-protein crosslinking to unstructured regions of proteins, on the other hand, this may be a result of RNA-driven crosslinking being a primarily base-edge driven process that is much less likely in case the RNA backbone interacts with a given protein surface. Last, to further substantiate our crosslinking data and to demonstrate that a single eRNA molecule is indeed able to form multivalent interactions with different parts of the NELF complex, we utilized Rosetta's FARFAR2 algorithm[45] in conjunction with our experimentally determined secondary structure restraints to calculate 3-dimensional models of both *Arc* and *Nr4a1*-(a) eRNA structures (Supplementary Fig. 6a, b). These models confirmed that large eRNA molecules (200 nt) can simultaneously bind widely-spaced areas of the PEC to induce NELF dissociation, whereas small eRNA fragments (50 nt) cannot (Supplementary Fig. 6c, d).

**NELF directly binds to nascent eRNAs in vivo.** Previous EMSA experiments[32,46], as well as our own in vitro data on PEC assembly (Fig. 2a and Supplementary Fig. 2a), showed a stable association of DSIF and NELF with Pol II requires the synthesis of short nascent pre-mRNAs. This suggests that the interaction between the pausing factors and nascent RNA is critical for mediating Pol II pausing. However, a direct demonstration of such interaction in an in vivo context has never been provided. Moreover, we also wanted to test the in vivo relevance of the eRNA-mediated NELF release that we could observe in vitro (Figs. 2, 3). To this end, we performed eCLIP-seq[47] in primary neuron culture before and after KCl stimulation. We determined the genome-wide interaction map between NELF-E and RNA during the early stage of activity-induced transcription (Fig. 5a and Supplementary Fig. 7a, b). By nature of eCLIP-seq library preparation, we could estimate the NELF-E crosslinking sites from each sequenced read, which is the first nucleotide of the R2 read (see Methods). To assign the crosslinking sites separately to eRNAs and pre-mRNAs, we utilized the transcript units we had previously defined de novo from GRO-seq reads (Supplementary Fig. 7c and Supplementary Data 1). The crosslinking sites were strikingly biased toward the 5′-ends of pre-mRNAs, which is consistent with the previous model that NELF interacts with the

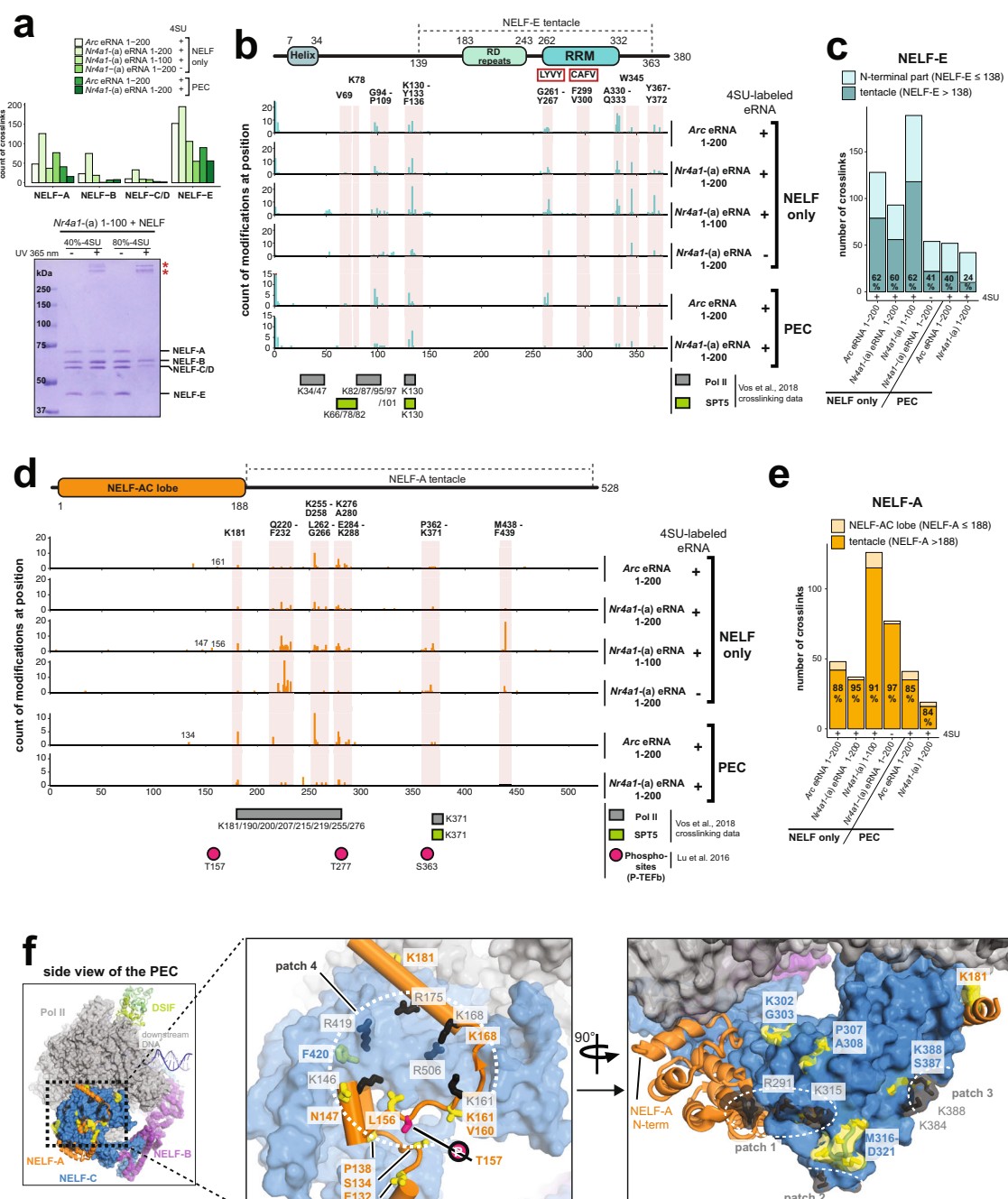

**Fig. 4 eRNAs form an extensive interaction network with the N-terminal region of NELF-E and the tentacles of NELF-A and NELF-E. a** The bar plot visualizes the total number of protein–RNA crosslinks (total count of nseen) to each NELF subunit for four (out of eight) eRNA crosslinking experiments to the isolated NELF complex and for two (out of two) experiments with the entire PEC (Supplementary Fig. 4a, experiments marked in red; Supplementary Data 4–7). Below is a 10% SDS-PAGE of two crosslinking experiments of 4SU-labeled *Nr4a1*-(a) eRNAs 1–100, before (−) and after (+) 365 nm-induced crosslinking to the NELF complex is shown. 40% and 80%-4SU denote the concentration of 4-thiouridine vs. uridine during in vitro transcription (Supplementary Fig. 4a, Methods). Upon crosslinking predominantly NELF-E and NELF-A disappears while high-molecular bands corresponding to crosslinked protein–RNA products appear (marked by red asterisks). **b** Total number of nucleotide-crosslinks at each amino acid (aa) position plotted along the NELF-E sequence (1–380 aa). Data were shown for the same samples as in (**a**). The crosslinking results for all eight experiments are shown in Supplementary Fig. 5. Clusters of crosslinks or single crosslinking positions that are present throughout all samples are highlighted. Representative amino acid residues are indicated above the highlighted boxes. The domain architecture of NELF-E is schematically displayed on top. **c** Distribution of all crosslinks plotted in (**b**) between the N-terminal portion of NELF-E and the NELF-E tentacle. **d** Nucleotide-crosslinks to NELF-A plotted as in (**b**). **e** Distribution of all crosslinks plotted in (**d**) between the NELF-AC lobe and the NELF-A tentacle. **f** Nucleotide-crosslinks to the NELF-AC lobe mapped onto the PEC (PDB code 6GML) displayed in the left panel. Crosslinked residues in the vicinity of the positively charged patches 4 (middle panel) and 1–3 (right panel) are displayed as sticks and colored in yellow. The referring amino acid is reported next to the structure and in the color of the corresponding NELF subunit (NELF-A = orange; NELF-C = blue). Lysine and arginine residues associated with the positively charged patches[42] are also shown as sticks and colored in black. T157 is highlighted in pink and marked with an encircled P. Source data for (**a**, **c**, **e**) are provided in a Source Data file. Data shown in (**b**, **d**) are part of Supplementary Data 4, 6.

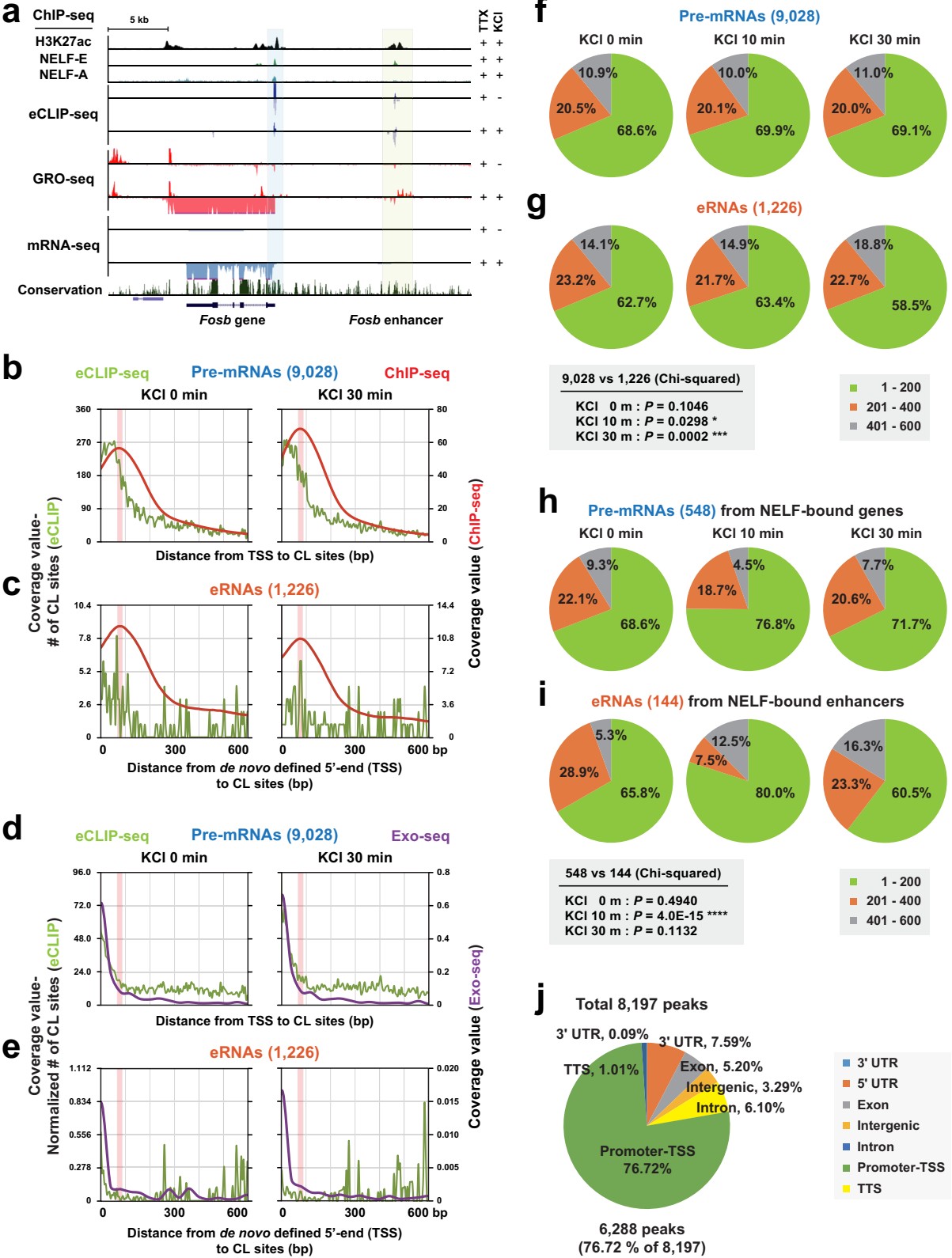

5′-ends of nascent RNAs emerging from transcribing Pol II[32,46] (Fig. 5b, f and Supplementary Fig. 8a, c). About 70% of the crosslinking sites were located within the first 200 nt of nascent pre-mRNAs in both unstimulated (Unst.; 0 min) and KCl-stimulated conditions with a notable enrichment in the first 50 nt. The distribution of the crosslinking sites within eRNAs was similar but more broadly spread toward downstream regions

(Fig. 5c, g and Supplementary Fig. 8b, d). To further examine whether or not the distribution of the crosslinking sites is influenced by the different abundance of individual nascent transcripts, we normalized the crosslinking numbers by GRO-seq signals. The proportion of the crosslinking sites near the 5′-end of eRNAs became much lower after normalization, resulting in more evenly distributed crosslinking events across the length of eRNAs.

**Fig. 5 NELF directly binds to nascent eRNAs in vivo. a** Example track showing one of the activity-induced genes, *Fosb* (Blue shaded at its promoter) and its nearby enhancer (Yellow shaded area) along with ChIP- (H3K27ac, NELF-E, and NELF-A), eCLIP-, GRO-, and mRNA-seq data. **b**, **c** The coverage profiles of the crosslinking sites (green line) from eCLIP-seq and NELF ChIP-seq coverage profiles (red line) present up to 600 bp downstream from the TSSs as defined by ENCODE annotation of pre-mRNAs (9028) (**b**) and from the de novo defined 5′-ends (TSSs) of intergenic eRNAs (1226) (**c**) at different time points after KCl stimulation (0 and 30 min). NELF peak (~60–70 bp downstream of the TSSs) portions (thin red box) from ChIP-seq data are also shown. The 607 overlapping eRNAs out of 1226 eRNAs were selected after transcript calling with GRO-seq (KCl 30 min). A total of 7242 annotated pre-mRNAs having one or more crosslinking sites out of 9028 pre-mRNAs (**b**) and a total of 240 eRNAs having crosslinking sites out of 607 eRNAs (**c**) were used for these analyzes. See also the figure legend of Supplementary Fig. 8a, b. **d**, **e** The coverage profiles of eCLIP-seq (green line) from the normalized number of crosslinking sites considering the expression level from GRO-seq and Exo-seq coverage profiles (purple line) for a total of 7242 annotated pre-mRNAs having crosslinking sites out of 9028 pre-mRNAs (**d**) and a total of 240 eRNAs having crosslinking sites out of 607 eRNAs (**e**) at KCl stimulation (0 and 30 min). NELF peak (~60–70 bp downstream of the TSSs) portions (thin red box) from ChIP-seq data were also shown. **f–i** Pie charts that show the proportion of crosslinking sites present in three distance windows (1–200, 201–400, or 401–600 nucleotides) for the total annotated (9028) (**f**) or KCl-up/NELF-bound (548) (**h**) pre-mRNAs, and the total intergenic (1226) (**g**) or KCl-up/NELF-bound (144) (**i**) eRNAs at different time points after KCl stimulation. *P* values were determined by Chi-squared. **j** ChIP-seq analysis of the NELF-E protein. The pie chart shows the genomic distribution of annotated NELF-E peaks. 5′UTR 5′ untranslated region, 3′UTR 3′ untranslated region, CDS coding sequence, TSS transcription start site, TTS transcription termination site. Source data for (**b–j**) are provided in a Source Data file.

On the other hand, pre-mRNAs still exhibit crosslinking occurring prominently near the 5′-ends (Fig. 5d, e). eRNAs are typically expressed at much lower levels than pre-mRNAs, but the difference in RNA abundance does not appear to have an impact on the crosslinking patterns, as 1632 pre-mRNAs whose expression levels were comparable to those of eRNAs (25–75 percentile range of eRNA expression levels), still exhibit the 5′-end-enriched crosslinking pattern (Supplementary Fig. 8g, h). Observed bias in the crosslinking sites toward the 5′-ends of nascent transcripts largely correlated with the average NELF-E occupancy, which is highly restricted to the immediate downstream regions of TSSs (Fig. 5b, c and Supplementary Fig. 8a, b). Given that low levels of NELF binding were detected at some enhancers, it is possible that nascent eRNAs could directly contact the NELF complex bound near genomic region of its origin. Although our eCLIP-seq analysis does not allow us to distinguish the origin of NELF that interacts with eRNAs, the following features observed from our analysis suggest the possibility of eRNA-dependent disruption of NELF association with paused Pol II at promoters, as previously suggested[18]. (1) There are much lower levels of NELF binding at enhancers than at promoters (Fig. 5j and compare the right y-axis values in Fig. 5b, c); (2) eRNA crosslinking sites show a broader distribution across the length of eRNAs in contrast to pre-mRNA crosslinking sites which are narrowly enriched near the 5′-ends (Fig. 5b–e and Supplementary Fig. 8a, b). There were also statistically significant differences in the distributions of NELF crosslinking sites between pre-mRNAs and eRNAs upon KCl stimulation, which was mainly caused by the changes in the crosslinking positions toward downstream regions in eRNAs rather than in pre-mRNAs (Fig. 5f–i and Supplementary Fig. 8c–f). (3) A comparison with the Exo-seq profile further confirmed that eRNA crosslinking results from NELF interactions at various positions along the length of transcribed eRNAs, not just near the 5′-ends of short eRNAs transcribed from alternative downstream TSSs at enhancers. Both the Exo-seq raw read density and the TSS peaks called by an Exo-seq read clustering algorithm were highly enriched near the 5′-ends of eRNAs defined de novo from GRO-seq data, with only a minor population present in the downstream regions (Fig. 5b, c and Supplementary Figs. 8a, b, 9a, b, respectively). Taken together, our eCLIP-seq analysis provides the correlative in vivo evidence that eRNAs are capable of making contacts with NELF associated with paused RNA Pol II at promoters following their synthesis at enhancers.

**eRNA-driven NELF release commands a more efficient release of Pol II from the paused state in vitro.** NELF release is a hallmark of the transition from promoter-proximal pausing to transcription elongation. As we found eRNA-driven NELF release

in vitro to resemble pause release in vivo (Fig. 2), and as we observed eRNA-NELF interactions in vivo (Fig. 5), we next asked whether eRNAs indeed trigger a more efficient release of Pol II from the paused state. To address this question, we established a mammalian in vitro pause release assay (see Methods)[30,32,48] (Fig. 6a). Before examining the effect of eRNAs, we verified that we indeed observe pause stabilization through the presence of both DSIF and NELF using our assay setup (Supplementary Fig. 10a). As expected, by adding *Nr4a1*-(a) or *Arc* eRNA fragments we observed a more efficient release of Pol II from the paused state (Fig. 6b and Supplementary Fig. 10b, upper gel). In striking similarity to our EMSA results (Fig. 2b, c), this effect was critically dependent on eRNA length and less so on the "structuredness" of the tested eRNA (Supplementary Fig. 10b, lower gel). Only *Nr4a1*-(a) eRNA variants with a length of 100 nt or exceeding this length showed increased Pol II pause release. We note that the second observed pause site at ~40–45 nt is a NELF-independent intrinsic pausing site that is caused by our experimental assay system. Next, we measured eRNA-induced Pol II pause release using a PEC variant that comprised the NELF-C patch mutant and utilizing both *Arc* and *Nr4a1*-(a) and -(b) eRNAs. As shown in Fig. 6c and Supplementary Fig. 10c, we observed hardly any Pol II pause release with this NELF mutant using *Arc* eRNA (1–50) or (1–100) and substantially diminished rates of release using longer eRNA variants. This is, again, in good agreement with our EMSA data (Fig. 3b and Supplementary Fig. 3b). We note here that the NELF-C patch mutant leads to an overall increase in Pol II pause stabilization (compare quantification in Fig. 6b, c). When using the NELF double mutant (both lacking the NELF-E RRM domain and comprising the NELF-C patch mutants, see Fig. 3c–e), we found Pol II pause release rates similar to those for the NELF-C patch mutant (Fig. 6d and Supplementary Fig. 10d). Taken together, our in vitro data on Pol II pause release confirm our prior findings and they further establish that, indeed, eRNAs of sufficient length are able to release Pol II from its paused state by stimulating NELF release. We note that the eRNA-induced increase in Pol II pause release efficiency was not as dramatic as the highly efficient release of NELF following eRNA addition in our EMSA assays (Fig. 2). This is likely due to the active engagement of Pol II in transcription with its active site in a tilted confirmation due to NELF and DSIF binding, as shown before[30]. Thus, jump-starting transcription after eRNA addition may be less efficient under our in vitro transcription assay conditions as compared to our EMSA setup (Figs. 2, 3). Moreover, our assay system is not able to recapitulate P-TEFb phosphorylation[35,49], which may also hinder the efficient resumption of transcription elongation in vitro.

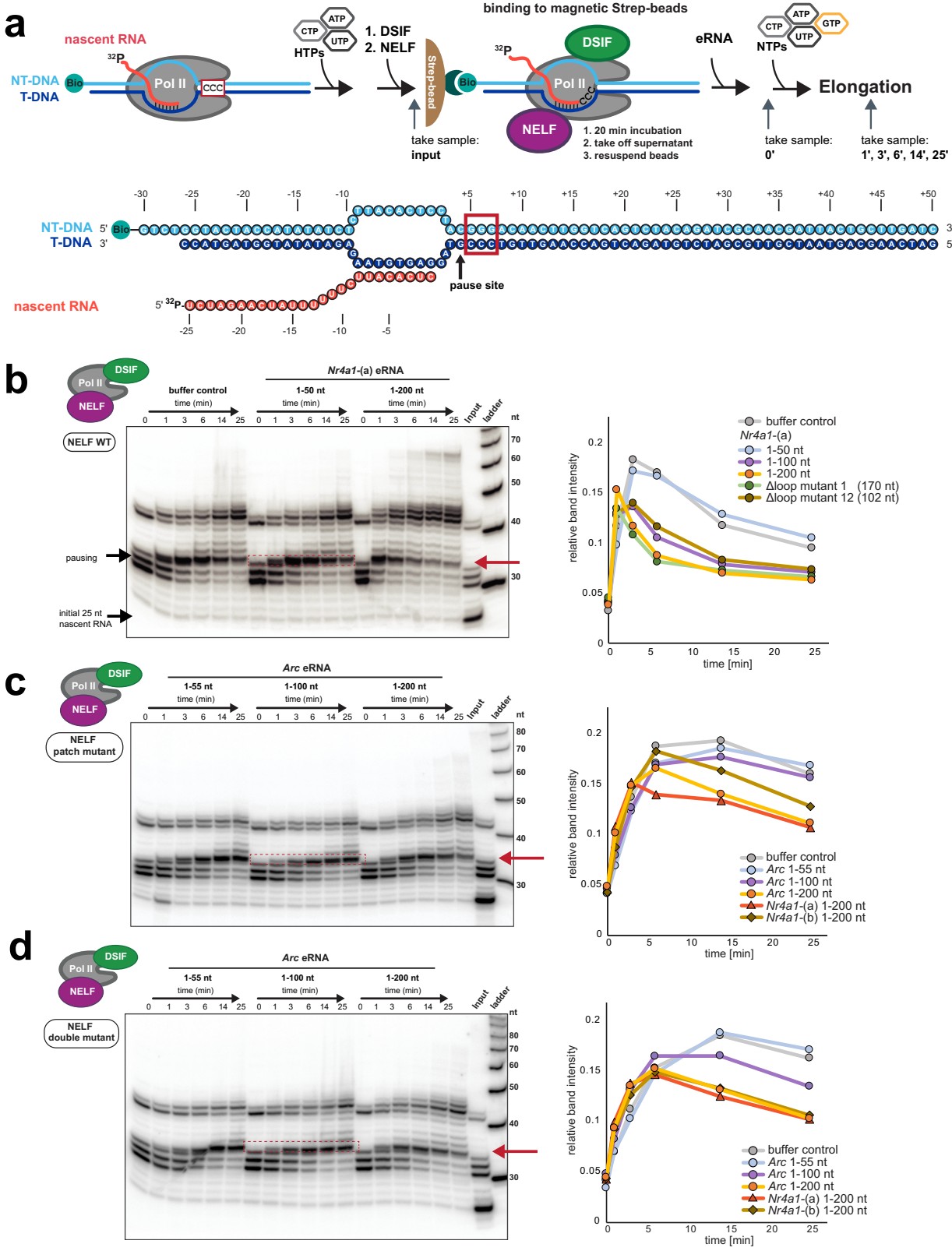

**NELF binding levels correlate with activity-induced transcription elongation.** Because our in vitro data revealed that eRNAs abrogate Pol II pausing by releasing NELF from the PEC (Fig. 6), we asked whether this mechanism plays a general role in activity-dependent transcription in neurons. To that end, we further analyzed our GRO-seq profiles as a proxy for nascent Pol II transcription (Fig. 7a–c and Supplementary Fig. 11). We grouped

all mRNA genes based on the binding level of NELF-E at the promoter as measured by ChIP-seq and/or their transcriptional inducibility in response to KCl stimulation (mRNA-seq signal FC >1.5 any time after KCl stimulation). A total of 623 activity-induced genes (KCl depolarization-induced [KCl-up]) were grouped as NELF-bound (548) and NELF-unbound (75). KCl-up/NELF-bound (548) genes were further sub-divided into three

**Fig. 6 eRNAs facilitate pause release in a length-dependent manner. a** Schematic of the mammalian pause release assay (top) and the utilized nucleic acid scaffold (bottom). Briefly, a transcription-competent complex that consists of Pol II and a transcription bubble, comprising a template (T-), non-template (NT-) DNA strand a nascent RNA, is assembled (see Fig. 2a). As the DNA contains a 4 nt long G-less cassette succeeded by three Gs (in the non-template strand), pausing can be induced by GTP omission. DSIF and NELF are then added to reconstitute the PEC and to stabilize the pause. The PEC is then isolated by binding the biotinylated DNA non-template strand to magnetic streptavidin-coated beads. Next, eRNAs are added, the supernatant (SN) is removed, and transcription is allowed to resume by the addition of NTPs. **b** Pause release assay using wild-type NELF and *Nr4a1*-(a) eRNA fragments (1–50 and 1–200). The left panel shows the urea PAGE analysis for the *Nr4a1*-(a) 1–50 and 1–200 fragments. Urea PAGE gels for *Nr4a1*-(a) eRNA (1–100) and two Nr4a1-(a) mutants (102 nt - Δloop 12 mutant; 170 nt - Δloop 1 mutant) are shown in Supplementary Fig. 5b. Samples for all experiments were taken just before NTP addition (0 min) and at different time points after NTP addition (1, 3, 6, 14, and 25 min). The "input" sample contains the PEC sample before its affinity purification using the streptavidin-coated beads. Thus, it allows for the visualization of unbound nascent RNA. For the quantification of pause release efficiency (shown on the right) the intensity of the first transcript elongation band past the pause (boxed in red) was analyzed. **c** Pause release assay using the NELF patch C mutant and *Arc* eRNA (1–55, 1–100, and 1–200). The left panel depicts the urea PAGE analysis for the *Arc* eRNA fragment (1–55), (1–100), and (1–200). Additional data for *Nr4a1*-(a) and *Nr4a1*-(b) 1–200 are shown in Supplementary Fig. 5c. The right panel shows the quantification of all experiments, as described for (**b**). **d** Pause release assay using the NELF double mutant. Performed under the same conditions as described in (**c**). Source data for (**b**–**d**) are provided in a Source Data file.

different groups based on the level of NELF-E binding (NELF-high [216], -mid [133], and -low [199]) (Fig. 7b and Supplementary Fig. 11b). We applied the same grouping strategy to the 252 activity-induced eRNAs (KCl-up; FC >1.5) out of 1226 intergenic eRNAs (Supplementary Data 1) and further divided them into NELF-bound (144) and -unbound (108) enhancers (Fig. 7c and Supplementary Fig. 11c). This analysis revealed that activity-induced genes and enhancers show a strong positive correlation between NELF binding levels and transcriptional induction (Fig. 7b, c). To further examine the function of NELF in Pol II pausing and elongation, we calculated the Pausing Index (PI) based on GRO-seq signals, which can estimate the relative density (and therefore transcription activity) of Pol II at the initiation and elongation stages[50] (Supplementary Fig. 11d). Activity-induced genes with high levels of NELF binding (KCl-up/ NELF-high) showed a decrease in PI at 30 and 60 min after KCl stimulation, indicating an increase in Pol II elongation (Fig. 7d and Supplementary Fig. 11e). However, such elongation stimulatory effect was not observed in activity-induced genes with weak or no NELF binding. Enhancers also exhibited a NELF-dependent increase in eRNA elongation was also observed but to a much lesser degree (Fig. 7e and Supplementary Fig. 11f). Taken together, these results suggest that in primary neuron culture NELF is part of the activity-dependent gene expression program through elongation control. NELF might also control eRNA transcription as about half of the activity-induced enhancers are bound by NELF, albeit at much lower levels than the activity-induced promoters.

## Discussion

**To detach NELF from the PEC, eRNAs must have sufficient length to bridge multiple RNA binding sites on NELF-A and NELF-E.** Our mechanistic study directly links eRNAs and mammalian Pol II for the first time and it fundamentally advances our previous findings[18]. What is more, in addition to the well-characterized NELF-E RRM domain[38–40], our data also shed light on the biological role of previously reported additional nucleic acid binding interfaces on NELF[42], in particular the NELF-A/C lobe and both the NELF-A and NELF-E tentacles, all of which we find to be contacted by eRNAs (Fig. 4 and Supplementary Figs. 4, 5). Our data clearly indicate that eRNA length cannot be compensated for by elevated concentrations of smaller eRNA molecules (Figs. 2, 3 and Supplementary Fig. 6). Thus, we speculate that individual eRNA-NELF contacts might be too weak and transient to allow for efficient eRNA-driven NELF dissociation from the PEC. In support, the NELF-A tentacle was hypothesized to significantly contribute to the overall affinity of NELF for RNA[42]. In more detail, we envision that the positively charged

patches on the NELF-AC lobe attract an eRNA molecule to the PEC. eRNA binding to the NELF-AC lobe then facilitates the establishment of further contacts between the eRNA and the NELF-A and/or the NELF-E tentacle (whose affinity towards the RNA is probably lower, see Fig. 4). Subsequently, the sum of these contacts induces the stripping off of both tentacles from Pol II-DSIF, which in turn induces NELF dissociation from the PEC (Fig. 8). In light of the proposed NELF dissociation mechanism, eRNA molecules require a certain dimension to span from the NELF-AC lobe (patch 2) to the middle of the NELF-A tentacle or to the NELF-E tentacle including the C-terminal RRM domain (Supplementary Fig. 6d). Short eRNA fragments such as *Arc* eRNA (1–55) are too short to span the distance between the AC lobe and any of the tentacles. Moreover, they are too rigid to adapt to the surface of the PEC (Supplementary Fig. 6c). Longer eRNA molecules can both span larger distances and they have greater conformational flexibility, which will increase their chance for adopting a conformation that triggers NELF dissociation. Intriguingly, RNA binding along subunits of NELF possesses transcription activation potential not only in the context of eRNAs. Part of the NELF-AC lobe and the NELF-A tentacle (NELF-A residues 89–248) show sequence similarity to the hepatitis delta antigen (HDAg)[43], a viral protein that binds the rod-like double-stranded viral genomic RNA and that is required for the replication of the Hepatitis delta virus (HDV)[51]. To prevent Pol II pausing and to stimulate HDV replication, HDAg competes with NELF-A for binding to a common Pol II surface[52]. To reveal the exact mechanism of eRNA-induced NELF dissociation from the PEC and any potential cooperativity between different eRNA-NELF binding events, detailed molecular studies are needed.

**Unpaired guanosines in enhancer RNAs and transcriptional condensates go hand in hand.** Against initial expectations eRNAs neither share common structural motifs nor do they possess sequence motifs that determine their function (Figs. 1, 2). In contrast, we find that enhancer RNAs must only meet two loose criteria to be able to efficiently abrogate Pol II pausing by detaching NELF from Pol II: (1) They need to be longer than 200 nt, and (2) multiple unpaired guanosines need to be distributed along the entire enhancer RNA (Figs. 2, 8). The preference of NELF for unpaired guanosines connects well to previous findings. First, the NELF-AC subcomplex was reported to bind single-stranded RNA with a GC content of 60%, but not RNA with a GC content of 44%[42]. A closer look at the utilized RNA sequences revealed, however, that, while the RNA representing a GC content of 60% comprised stretches of guanosines (14 Gs in total), the RNA representing a GC content of 44% contained no guanosines

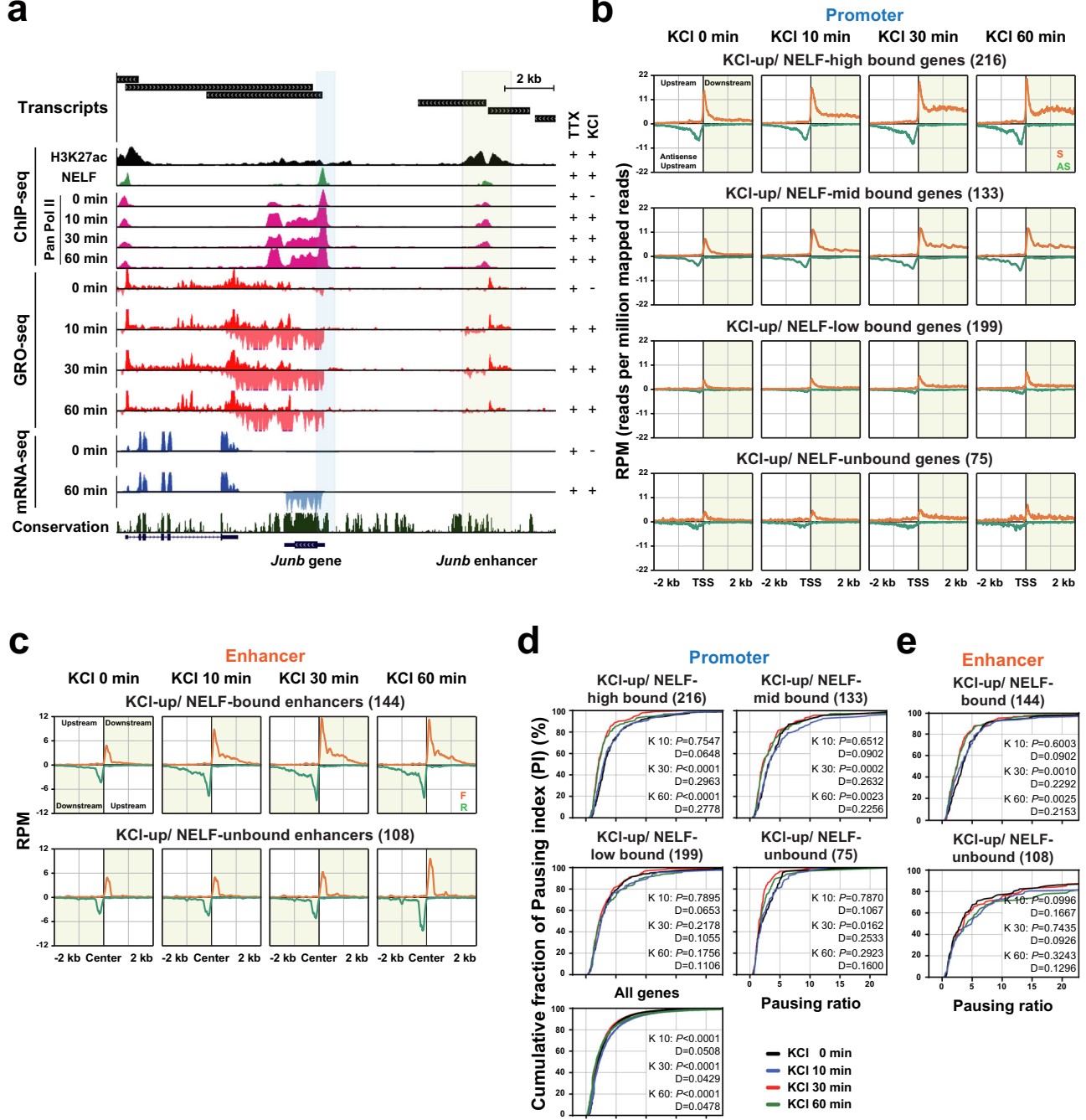

**Fig. 7 NELF occupancy is correlated with Pol II pause release and causal to transcription induction in vivo. a** Example track that shows an activity-induced gene, *Junb* (Blue shaded at its promoter) and its nearby enhancer (Yellow shaded area). Transcript units determined by de novo transcript calls are shown on the top together with tracks for ChIP- (H3K27ac, NELF-E, and Pan Pol II), GRO-, and mRNA-seq. **b, c** The average GRO-seq profiles of four different gene groups based on the levels of NELF binding (NELF-high [216], -mid [133], -low [199], or -unbound [75]) (**b**) and two different enhancer groups (NELF-bound [144] or -unbound [108] (**c**). The profiles are shown for ±2 kb regions centered on the TSSs or the enhancer centers upon 0, 10, 30, and 60 min KCl stimulation. Orange lines denote the sense strand at promoters and the forward strand at enhancers. Green lines denote the anti-sense strand at promoters and the reverse strand at enhancers. **d, e** Pausing index (PI) profiles of four different gene groups based on their levels of NELF binding with all genes (**d**) and KCl-up/NELF-bound (144) or -unbound (108) enhancers (**e**). Each gene group is analyzed at four time points (0, 10, 30, and 60 min KCl). The PI is defined in Supplementary Fig. 11d. Statistical significance between cumulative probability graphs was determined by the Kolmogorov–Smirnov test. Source data for (**b**–**e**) are provided in a Source Data file.

at all. Second, preferential binding of NELF to guanosines was also reported for the *D. melanogaster* homolog of NELF-E and its isolated RRM domain[41]. The study identified the sequence CUGGAGA as NELF binding element (NBE). Mutating all guanosines to adenosines within the NBE abolished *D. melanogaster*

NELF-E RNA binding, underscoring the preference of NELF for guanosines. Interestingly, a comparison of the sequences of fly and human NELF-E reveals that the *D. melanogaster* homolog is lacking the RD repeat domain[41]. This fact might explain why human NELF-E binds RNA in a less sequence-specific manner

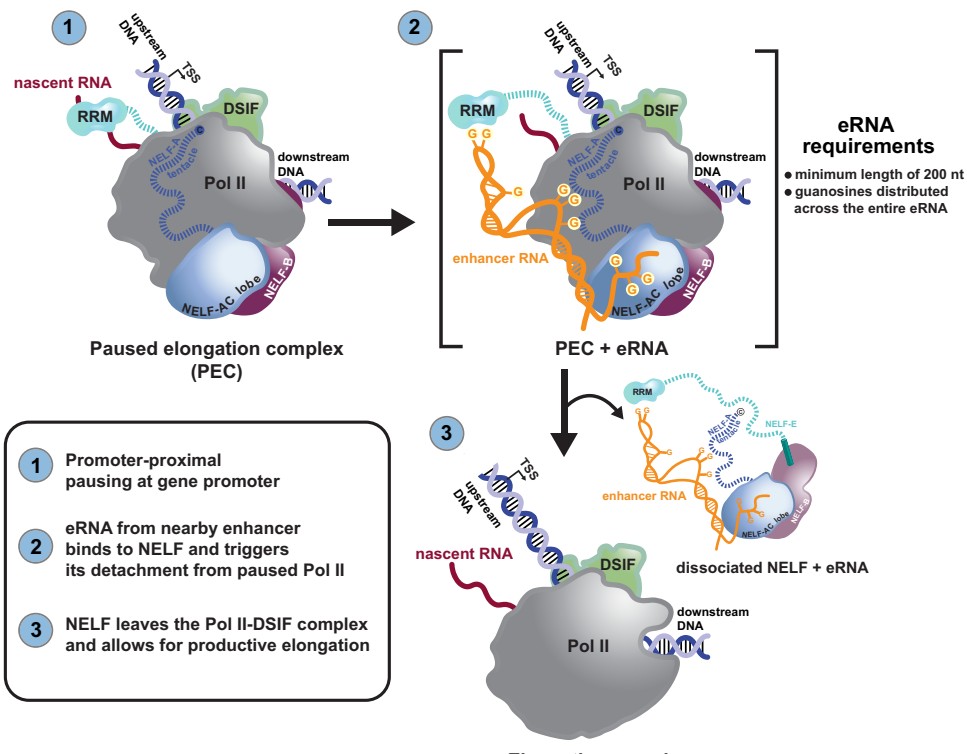

**Fig. 8 Model of how multivalent, allosteric interactions between an eRNA and the paused elongation complex (PEC) stimulate Pol II pause release.**
Following the transcription of a nearby, activated enhancer, an eRNA molecule contacts the paused elongation complex (consisting of Pol II, DSIF, and NELF) at the target gene promoter. The eRNA interacts with NELF at multiple sites (positive patches on NELF-AC, the NELF-A and -E tentacles, and the NELF-E RRM domain). These multivalent, allosteric interactions collectively trigger NELF detachment from the PEC. The resulting elongation complex (consisting of Pol II and DSIF) then resumes transcription elongation. The length of the eRNA (>200) and unpaired guanosines within an eRNA are essential for its ability to detach NELF from the PEC.

and allows for the binding of the broad range of RNAs that our data reveal.

A lack of well-defined motifs that render eRNAs active is not without mechanistic precedent. PRC2 (Polycomb repressive complex 2), a histone methyltransferase that establishes repressed chromatin by trimethylating histone H3 at lysine 27, overall shows promiscuous RNA binding[53,54]. However, the complex preferentially binds to unstructured G-rich sequences and to G-quadruplexes[55]. This is reminiscent of our RNA binding data on NELF (Figs. 2–4), with the notable exception that we find no evidence for a role of G-quadruplexes in eRNA function. Intriguingly, G-tract-containing RNAs sequester PRC2 from nucleosome substrates and thereby lead to gene activation[56,57]. Hence, decoying protein factors with overall promiscuous RNA binding activity by high-affinity binding to G-rich RNAs might be a general concept in RNA-dependent regulation of gene expression. Last, a lack of specificity of RNA-protein interactions might also be compensated for by compartmentalization into sub-organellar structures. This is exactly what was proposed for the regulation of eukaryotic transcription by transcriptional condensates[58,59]. As RNA binding increases the phase separation properties of RNA-binding proteins[60,61], we envision that eRNAs contribute to the formation of transcription condensates[62]. They likely do so by significantly increasing the valency of the network of interactions that lies at the heart of phase-separated foci[59]. Moreover, we hypothesize that the presence of enhancer RNAs in the same transcriptional condensate as Pol II and other pausing factors will greatly enhance their capacity to regulate gene expression, e.g., by abrogating promoter-proximal pausing. Indeed, a recent in vitro study under physiologically relevant conditions demonstrated that eRNAs transcribed from super-enhancers influence condensate formation by purified Mediator complex[62].

Finally, because we find P-TEFb phosphorylation sites to coincide with eRNA binding sites on NELF-A (Fig. 4f), our study offers the first insight into the possibility that eRNA binding to NELF might bypass the necessity of P-TEFb activity for Pol II pause release. Both eRNAs binding to NELF, as well as NELF phosphorylation by P-TEFb, might introduce negative charges into critical areas on the NELF surface. In analogy to the activation of cyclin-dependent kinases by T-loop phosphorylation and concomitant arginine rearrangement[63], these negative charges could then induce conformational changes in NELF that might trigger its dissociation from Pol II.

**eRNAs can directly interact with NELF in an in vivo context.**
Our eCLIP-seq analysis in primary neuron culture demonstrates that NELF directly interacts with both pre-mRNAs and eRNAs with a bias toward their 5′-ends. This provides the first genome-wide in vivo evidence that supports the model that a stable association of NELF with paused Pol II is mediated by its interaction with nascent RNA[30,44]. The observed direct interactions between eRNAs and NELF in primary neurons further suggest that the eRNA-mediated NELF release from the PEC demonstrated by our reconstituted system could be part of a transcriptional induction mechanism that operates in an in vivo context. However, we cannot rule out that some eRNA eCLIP-seq signal stems from crosslinking of eRNAs to NELF bound at enhancers, as low levels of NELF binding peaks are also present at

enhancers (Fig. 5). With this caveat, the following observations are consistent with the possibility that eRNAs mediate destabilization of NELF at promoters in vivo. The distribution of eRNA crosslinking sites is broader than that of pre-mRNAs relative to the average NELF binding peak area (Fig. 5b–e and Supplementary Fig. 8a, b). The pre-mRNA crosslinking sites are tightly enriched in less than ~50 nucleotide regions from their 5′-ends, which is even more upstream than the NELF peak (~60–70 bp downstream of the TSSs). In contrast, a significant number of eRNA crosslinking sites were observed up to ~1000 nt downstream from their 5′-ends, a finding that coincides with the typical size of an eRNA transcription unit[11]. As we had excluded all eRNAs that overlapped with mRNA TSSs from the analysis, this result indicates that, unlike pre-mRNAs, eRNAs interact with NELF during and even after the synthesis of their full-length transcripts, which indirectly suggests a possibility of making contacts with the promoter-bound NELF. The broad distribution of crosslinking sites also fits nicely with the length requirement of eRNAs for efficient NELF release proposed by our in vitro data (Figs. 2, 4). Another feature of eRNAs compatible with our model is their fast induction kinetics. Several studies have shown relatively faster induction kinetics exhibited by eRNAs compared to mRNAs in various enhancer-gene contexts[4,18,64–68]. Therefore, whether or not eRNA production is under the control of NELF activity, a population of eRNAs transcribed before pre-mRNA production rises can be available for interaction with promoter-bound NELF. In this regard, it is worth pointing out that a previous study in *Drosophila* also suggested that Pol II at enhancers undergoes more rapid pause release than at promoter regions[10]. This model is well suited to a broader distribution of the crosslinking sites across the length of eRNAs (Fig. 5). We can further infer that eRNA-mediated NELF detachment from paused Pol II is more feasible in a transcriptional condensate environment created by enhancer-promoter loops, where eRNAs are closely confined with promoter-associated transcription factors and Pol II.

## Methods

**Animals**. All animal experiments performed in this study were reviewed and approved by the IACUC committee at Pohang University of Science and Technology (POSTECH). Mice (Mus musculus) were housed in LD12:12 (12 light cycles/12 dark cycles) conditions at 40–60% humidity and a temperature of 65–75 °F (18–23 °C). All mice were on a C57BL/6 J genetic background (male and female). Timed matings were set up with males and females from 6 to 26 weeks of age and E16-E18 embryos of age were used for experiments.

**Mouse cortical neuron culture and stimulation**. Mouse cortical neurons were dissected at embryonic day 16.5 (E16.5) and cultured in Neurobasal media (NB) (21103-049, Thermo Fisher Scientific) supplemented with 2% B-27 (17504-044, Thermo Fisher Scientific) and 1% Glutamax (35050, Thermo Fisher Scientific). For KCl depolarization, neurons at DIV 7 were made quiescent by 1 μM tetrodotoxin (TTX; 1078, Tocris, Minneapolis, MN) overnight. 55 mM KCl was then added for the indicated length of time.

**Global run-on sequencing (GRO-Seq)**. GRO-seq was performed as previously described[19,69]. Briefly, 10 million nuclei per sample were used for global run-on, and base hydrolysis was performed as previously described[70]. Nascent RNA was immunoprecipitated with anti-BrdU antibody-conjugated beads (sc-32323 AC, Santa Cruz Biotech, Santa Cruz, CA). Purified run-on RNA was subjected to polyA tailing by Poly(A)-polymerase (14.06 U; M0276L, NEB, Ipswich, MA) for 12 min at 37 °C. PolyA-tailed RNA was subjected to another round of immunopurification by using anti-BrdU antibody-conjugated beads. Reverse transcription was then performed using SuperScript III Reverse Transcriptase (200 U; 18080-044, Thermo Fisher Scientific) with RT primer (pGATCGTCGGACTGTAGAACTCT/idSp/ CCTTGGCACCCGAGAATTCCATTTTTTTTTTTTTTTTTTTT-TTVN) for 2 h at 48 °C. Extra RT primers were removed by Exonuclease I (100 U; M0293S, NEB, Ipswich, MA) for 2 h at 37 °C. cDNAs were fragmented with basic hydrolysis and size-selected (130–500 nucleotides) in a 6–8% polyacrylamide TBE-urea gel. Purified cDNAs were circularized using CircLigase (50 U; CL4111K, Epicentre) for 2 h at 60 °C and relinearized at the basic dSpacer furan with Ape 1 (15 U; M0282S, NEB, Ipswich, MA) for 2 h at 37 °C. The relinearized single-stranded DNA

template was subjected to PCR amplification by using barcoded primers for Illumina TrueSeq small RNA sample and Phusion High-Fidelity DNA Polymerase (2 U; M0530L, NEB, Ipswich, MA). Subsequently, PCR products were size-selected in 6% polyacrylamide TBE gel (175–400 bp) and purified. The final libraries were sequenced using Illumina NEXTSEQ 500 following the manufacturer's instructions.

For analysis, low quality and adapter sequences of the raw FASTQ reads were trimmed using Trim_galore with default parameters. To trim polyA sequences, we used Trim_Galore again with the "--poly" parameter (https://www.bioinformatics. babraham.ac.uk/projects/trim_galore/). Trimmed reads were aligned to the mm10 GENCODE annotation using STAR (https://github.com/alexdobin/STAR/). Transcripts were called using "findPeaks" of the HOMER package underuse of the "--minReadDepth 200" parameter (http://homer.ucsd.edu/homer/). Samtools and the HOMER package were used to make visualization tracks and RPKM calculations. To calculate RNA expression values by gene accurately, we used the Homer package. To generate GRO-seq coverage plots, we used HOMER. For HOMER to draw promoter regions, "makeTagDirectory" program was used. Before making tag directories, we converted BAM files to BED files using "bedtools bamtobed" (https://github.com/arq5x/bedtools). Using HOMER's built-in Perl script 'annotatePeaks.pl' coverage plots were created. To compare read densities between pre-mRNAs and eRNAs in defined windows, the read counts for both pre-mRNAs and eRNAs were calculated in defined windows of 1 or 10 kb from the TSS of promoters and enhancers, respectively. Subsequently, GRO-seq coverage plots were generated using 'annotatePeaks.pl' of HOMER. Pausing indices were calculated as shown in Supplementary Fig. 12c using 'analyzeRepeats.pl' of HOMER. The widths of intervals used to calculate pausing indices were determined from analysis of GRO-seq alignments. The promoter-proximal region was defined as 100 bp upstream to 200 bp downstream of the TSSs. The gene body was defined from 400 to 800 bp downstream of the TSSs. In each window, GRO-seq read density was calculated.

**Identification of enhancers**. Enhancers were identified according to MACS-called H3K27ac-enriched peaks. Firstly, H3K27ac peaks were identified in KCl-stimulated neurons (GSM1467417 and GSM1467419)[20] with their corresponding input controls (GSM1467414 and GSM1467416, respectively) using MACS. Mapped reads of two biological replicates in each condition group were merged before the enhancer calling analysis using Bamtools. Then, the ranking of super-enhancer (ROSE) algorithm was used to define super-enhancers with the identified H3K27ac peaks. The H3K27ac peaks that were not overlapped with the super-enhancers (SEs) or promoter regions of known genes were defined as typical-enhancers (TEs). We used the pool of SEs and TEs as total enhancers in this study. To identify de novo enhancer transcripts from total GRO-seq transcripts, H3K27ac peaks within ±2 kb region from TSSs and gene body regions were removed from total enhancers. Then, total GRO-seq transcripts overlapped with total enhancers above were defined as de novo enhancer transcripts.

**mRNA-seq**. mRNA-seq library was constructed using TruSeq RNA Library Preparation Kit (Illumina) according to the manufacturer's instructions. FASTQ reads from Genomics Core at UTSW were mapped to UCSC's mm10 genome using Tophat with options "-a 8 -m 0 -I 500000 -p 8 -g 20 --library-type fr-firststrand --no-novel-indels --segment-mismatches 2". Since this data was strand-specific, we used the "-library-type fr-firststrand" option of Tophat. Reads with low mapping quality (<10) were removed using SAMtools. Duplicate reads were marked by Picard MarkDuplicates (https://broadinstitute.github.io/picard/). Tag directories for each sample were created using "makeTagDirectory" program. RNA expression was quantified using HOMER's inbuilt Perl script "analyzeRepeats.pl." These scripts offer flexibility to calculate expression values as reads per kilobase per million mapped reads (RPKM) normalized to 10 million at introns, exons, and gene body locations. "makeUCSCfile" from HOMER was used to create bedGraph files at 1 bp resolution and created bigWig files for visualization on UCSC genome browser. All coverage values were normalized to 10 million reads. We set expressed gene criteria as "RPKM values higher than 1 at least in one of six samples (three conditions, two replicates for each condition)". The subsequent RNA-seq analyses were performed with these 12,739 genes. To call significant differentially expressed genes (DEGs), we set our criteria as "fold change of RPKM is more than 1.5 and FDR of DESeq2 is less than 0.05 in both replicates".

**Depletion of ribosomal RNA and SRP-RNA from total RNA isolated from mouse cortical neurons**. To deplete rRNA from total RNA isolated from mouse cortical neurons a commercially available kit (RiboCop rRNA depletion kit V1.2, Lexogen) was used according to the manufacturer's instructions with some modifications. Briefly, for the simultaneous, additional depletion of the signal-recognition particle RNA (SRP-RNA) we utilized four biotinylated DNA oligos (see below) and mixed those with the probe mix (PM) directed against rRNA that is part of the Lexogen kit. About 500 ng of total RNA from unstimulated or stimulated (45′ KCl treatment) primary neurons were used for one round of depletion. In total, 115 μL of magnetic streptavidin beads (75 μL depletion beads from the kit + 40 μL Dynabeads MyOne Streptavidin C1, Thermo Fisher Scientific) were used. All buffer volumes of the depletion kit were scaled according to the

manufacturer's protocol. The hybridization mix was prepared from total RNA, 6.2 μL hybridization solution (HS), 5 μL probe mix (PM) (volume not scaled) + 3.5 pmol anti-SRP oligo-mix in a final volume of 54 μL. Following the first depletion step, the second round of depletion was performed to ensure that all depletion oligos were removed. To that end, the supernatant containing the rRNA- and SRP-RNA depleted RNA was transferred into a fresh reaction tube and supplemented with 30 μL of pre-equilibrated magnetic streptavidin beads. The mix was incubated at room temperature for 10 min and subsequently at 52 °C for 10 min. The cleared supernatant was then transferred to a fresh reaction tube and applied to spin-columns (RNA Clean & Concentrator-5 columns, Zymo-Research) according to the manufacturer's instructions. For one Exo-seq library sample, RNA from three depletion reactions (3x 500 ng total RNA) were pooled prior to column purification.

anti_SRP_oligos (for SRP-RNA depletion):
#1: 5′-Biotin-TACAGCCCAGAACTCCTGGACTCAAGCGATCCTCCTG
#2: 5′-Biotin-ATCCCACTACTGATCAGCACGGGAGTTTTGACCTGCTC
#3: 5′-Biotin-TCACCATATTGATGCCGAACTTAGTGCGGACACCCGATC
#4: 5′-Biotin-CTATGTTGCCCAGGCTGGAGTGCAGTGGCTATTCACAG

**Exo-seq library preparation and data analysis.** The Exo-seq library construction from two biological replicates of KCl (30 min)-treated and one untreated sample was performed as described[21], with the following alternations. Instead of poly(A)-selected RNA we started with an rRNA and SRP-RNA depleted sample (prepared from 1.5 μg total RNA input, see before). DNA oligos were altered to include a linker 1 (3′-adapter: 5Phos/TGGAATTCTCGGGTGCCAAGG/ddC) and a linker 2 (5′-adapter: 5Phos/GATCGTCGGACTGTAGAACTCTGAAC/ddC) sequence based on the adapters of the Illumina TruSeq small RNA sample prep kit. For the final PCR enrichment step, which was performed for 14 cycles, TruSeq RP1 (forward) and RPI Index (reverse) primers from the TruSeq kit were used. The library was cleaned up and size-selected by two consecutive rounds of binding to 1.2x and 0.8x SPRI beads (Agentcourt AMPure XP beads, Beckman Coulter). The integrity and size distribution of the final libraries (two biological replicates of KCl-treated and one of untreated) were analyzed on a Bioanalyzer before they were applied to a NextSeq 500 platform for a single-end 75 nt sequencing run.

Raw reads were processed using *Cutadapt* for adapter trimming, retaining all reads with a minimum read length of 20 nt after trimming. The remaining rRNA- and tRNA reads were removed using *SortMeRNA* and *Bowtie*, respectively. rRNA and tRNA reference files were obtained from the UCSC table browser. Processed reads were then aligned to the mouse genome (mm10, December 2009) using *STAR*, allowing two mismatches and removing multimappers. 5′-end coverage of mapped reads was calculated for both strands separately using the "bedtools genomecov" utility with -bg and -5 parameters. TSSCall was then run on the generated bedgraph files with standard parameters for the identification of transcription start sites (TSS). We defined extragenic TSSs as TSSs that do not occur within a RefSeq annotated gene ±2 kb. By using these criteria 129,161 extragenic TSSs were identified for Rep1 and 131,312 for Rep2. Called TSSs were then overlapped with de novo GRO-seq defined enhancer transcript units allowing an offset of ±200 nt. A single TSS was then selected for each identified enhancer locus. This selection was based on read coverage and distance to the 5'-end of the respective GRO-seq transcript. This resulted in 977 TSSs for Replicate 1 and 1039 for Replicate 2. Activity-induced enhancers were defined based on GRO-seq fold change (FC >1.5) between stimulated and unstimulated conditions. In the last step, 39 high-quality eRNA TSSs were selected for structure mapping. Among these 33 eRNA TSSs were derived from activity-induced eRNAs, as defined by GRO-seq data. To generate Exo-seq profiles (KCl 0 or 30 min), the de novo enhancer transcripts with GRO-seq (KCl 30 min) data were called using "findPeaks" of HOMER package and the coverage values of the first bp from TSSs of ENCODE annotation of pre-mRNAs or the de novo defined 5′-ends (TSSs) of intergenic eRNAs were plotted using the Ngs.plot R package[71]. To generate heatmaps, we also used the Ngs.plot R package. To generate the distribution of called TSSs, we first used a capped small RNA-seq (csRNA-seq) analysis program[72], which filters out noise and defines only significant clusters of reads (peaks) that represent TSSs by using the "findPeaks" option of the HOMER package with the "-style tss --ntagThreshold 5" parameter (see also http://homer.ucsd.edu/homer/ngs/csRNAseq/index.html).

**In vitro transcription and purification of enhancer RNAs.** eRNAs used for EMSAs, transcriptional pause release assays, and for SHAPE-MaP were produced by T7 RNA polymerase-mediated in vitro run-off transcription using PCR generated DNA templates (adapted from refs. [73,74]). For this purpose, mouse eRNA sequences were cloned via Gibson assembly into pUC18 vectors containing a T7 promoter sequence (5′-TAATACGACTCACTATAGG) in front of the eRNA sequence. Then, a PCR template was generated using a T7 promoter-spanning forward primer and an eRNA-specific reverse primer. Depending on the desired yield of each eRNA, 200 μL (for SHAPE-MaP eRNA) or 0.8–1.2 mL (for EMSA) transcription reactions were set up. The transcribed eRNAs were purified by urea PAGE, passively eluted into RNA buffer (25 mM K-HEPES pH 7.4, 100 mM KCl, 0.1 mM EDTA) by the crush and soak method[75] and concentrated to a volume of 0.5–1 mL using centrifugal filters (Amicon Ultra-4 /-15, Millipore) with a molecular weight cutoff (MWCO) depending on the molecular size of the produced fragment (1–50 nt—3 kDa; 1–100 nt—10 kDa; 1–200 nt—10 or 30 kDa).

Subsequently, eRNAs were subjected to size-exclusion chromatography using RNA buffer as running buffer on a Superdex 200 Increase 10/300 GL column to purify RNA monomers from aggregates. After elution from the column, the monodisperse peak fractions were pooled and concentrated to eRNA concentrations in the range of 10–20 μM.

**SHAPE-MaP library preparation and data analysis.** Thirty-nine eRNAs (1–200 nt) for SHAPE-MaP were produced as described above, except that eRNA fragments for SHAPE-MaP were flanked with a 20 nt long 5′-linker (5′-GGCCATC TTCGGATGGCCAA) and 43 nt long 3′-linker (5′-TCGATCCGGTTCGCCGG ATCCAAATCGGGCTTCGGTCCGGTTC)[76]. Prior to chemical probing, each purified eRNA (10 pmol in 18 μL) was incubated in 1x HEMK buffer (50 mM K-HEPES pH 7.0, 0.1 mM Na-EDTA, 150 mM KCl, and 15 mM MgCl$_2$) for folding at 37 °C for 30 min (adapted from ref. [77]). Chemical probing and library preparation were carried out according to the small RNA workflow[27]. The folded RNA sample was split into two samples: each sample was either treated with 1 μL of 100% DMSO (DMSO control sample) or 1 μL of 100 mM 1-methyl-7-nitroisatoic anhydride (1M7-modified sample) at 37 °C for 5 min. The treatment was repeated for the second round of modification to achieve higher modification rates. The volume of the modified RNA was adjusted to 30 μL with DEPC-treated water, the RNA was cleaned up with 1.8x AMPure XP beads (Beckman Coulter) according to the manufacturer's instructions and eluted in 15–25 μL of DEPC-treated water. Ten microliters of the purified RNA sample were subjected to reverse transcription using a specific SHAPE-MaP RT primer complementary to the 3′ linker region, as described in ref. [27]. For eRNAs which yielded no DNA product, the temperature during reverse transcription was increased to 50 °C. The cDNA was cleaned up with 1.8x AMPure XP beads and eluted in 35 μL DEPC-treated water. Subsequently, the first and second PCR steps were performed as follows: Two distinct Index-primers were used for the DMSO control and the 1M7-modified libraries. The DNA from the first PCR was purified with 1.0x AMPure XP beads and from the second with 0.8x. Final libraries were eluted in 25 μL. Each sample was analyzed on a Fragment Analyzer (Agilent) and concentrations were fluorometrically quantified using Qubit (Thermo Fisher Scientific). All individual samples were diluted with library dilution buffer (10 mM Tris-HCl pH 8.0, 0,1% (v/v) Tween20) to 5 nM and pooled equimolar. Libraries were sequenced on a NextSeq 500 platform with a Mid-Output NextSeq kit in a 150 bp paired-end mode and ~30% PhiX due to the low complexity of the library mix.

SHAPE-MaP sequencing reads were analyzed by the *ShapeMapper2* tool. The software was executed with default parameters, which align the reads to the given eRNA reference sequences using Bowtie2 (default option was used) and compute the read-depth of each eRNA along with the mutation rates and SHAPE reactivities. eRNA secondary structures were predicted with *RNAStructure*[29] using the SHAPE reactivities per nucleotide as constraints. The *MaxExpect* structure (maximum expected accuracy structure)[29], was selected for visualization of the secondary structure using the *StructureEditor* (belongs to RNAStructure software package). For the G-content plot in Fig. 2j sequences of the eRNAs were extracted from position 1 to 1000 using gffread. Sequences were then binned in non-overlapping bins of 200 nucleotides and analyzed for their sequence content. Differences in base content were tested for significance with a pairwise *t*-test using a custom R script.

**Expression and purification of proteins for in vitro experiments**

*Purification of mammalian RNA polymerase II. Sus scrofa* Pol II was purified as described with minor alterations as follows[31,78]. Pig thymus was collected from the Bayerische Landesanstalt für Landwirtschaft (LfL) in Poing, Germany. The bed volume of the 8WG16 (αRPB1 CTD) antibody-coupled sepharose column was 3 mL. The final SEC step was omitted. Instead, Pol II peak fractions after anion-exchange chromatography (UNO Q, Bio-Rad) were pooled and the buffer was exchanged to Pol II buffer (25 mM Na-HEPES pH 7.5, 150 mM NaCl, 10 μM ZnCl$_2$,10% (v/v) glycerol, 2 mM DTT) using Amicon centrifugal filter units with a 100 kDa cutoff (Millipore). Pol II was concentrated to 2.6 mg/mL (5 μM), aliquoted, flash-frozen in liquid nitrogen, and stored at −80 °C.

*Expression of NELF variants and DSIF.* Bacmids for protein expression were made in DH10EMBacY cells[79]. For the generation of V$_0$ virus, bacmids were transfected into SF21 insect cells in SF-4 Baculo Express medium (Bioconcept) at a density of 0.8 × 10$^6$ cells/mL. YFP fluorescence was monitored as a proxy for transfection efficiency and cells were kept at a density of 0.8 × 10$^6$ cells/mL until 100% of cells showed YFP fluorescence. The supernatant, containing the V$_0$ virus, was subsequently isolated and utilized to infect High Five cells at a 1:10 through 1:20 ratio. For protein expression High Five cells were kept at a density of 1.0–1.2 × 10$^6$ cells/mL. Protein expression was allowed to proceed for 48 h before cells were harvested by centrifugation. Cell pellets were resuspended in 60 mL (per 1 L expression culture) lysis buffer (for NELF: 20 mM Na-HEPES pH 7.4, 300 mM NaCl, 10% (v/v) glycerol, 30 mM imidazole, 2 mM DTT; for DSIF: 50 mM Tris-HCl pH 8.0, 150 mM NaCl, 2 mM DTT) supplemented with the protease inhibitors pepstatin A (1 μg/mL), leupeptin (1 μg/mL), benzamidin-hydrochloride (0.2 mM), and phenylmethyl sulphonyl fluoride (PMSF) (0.2 mM). Cells were directly used for protein purification or flash-frozen in liquid nitrogen and stored at −80 °C.

*Purification of NELF complex variants.* The purification protocol for the wild-type NELF complex and all NELF complex variants (NELF-EΔRRM, NELF patch mutant, and NELF double (patch + ΔRRM) mutant) were adapted from a prior protocol[42]. Following Ni-NTA affinity chromatography (Qiagen), the NELF containing protein fractions were pooled, filtered, and loaded onto a pre-equilibrated (20 mM Na-HEPES pH 7.4, 300 mM NaCl 10% (v/v) glycerol, 2 mM) Resource Q column (6 mL column volume). As the four-subunit NELF complex does not bind to the column, the flow-through was collected, diluted to 150 mM NaCl, reloaded onto the Resource Q column, and eluted with a linear salt gradient to 1 M NaCl (The first Resource Q step allows for the removal of trimeric NELF-ABD complexes). Peak fractions were pooled, concentrated using 30 kDa Amicon concentrators (Millipore), and loaded onto a Superose 6 10/300 column that had been equilibrated in SEC (size-exclusion chromatography) buffer (20 mM Na-HEPES pH 7.4, 150 mM NaCl, 2 mM DTT). Peak fractions after SEC were pooled and concentrated to ~30 μM (~7.0 mg/mL). Finally, glycerol was added to 10% (v/v) before the protein was aliquoted, flash-frozen in liquid N$_2$, and stored at −80 °C.

*Purification of DSIF.* DSIF carried a C-terminal 2xStrep-tag on SPT5. Cells were lysed by sonication, lysates were then cleared by ultracentrifugation (Type 45 Ti (Beckman), 35 min, 33,000 rpm (102,880 × *g*), 4 °C). Subsequently, the supernatant was applied to 1 mL Strep-Tactin beads (IBA) that had been equilibrated in binding buffer (50 mM Tris-HCl pH 8.0, 150 mM NaCl, 2 mM DTT). Beads were then washed with the same buffer (40 CV) and the protein was eluted with 20 mL (1 mL fractions) elution buffer (binding buffer supplemented with 2.5 mM desthiobiotin). Elution fractions were analyzed by 10% Bis-Tris PAGE, DSIF-containing fractions were pooled and applied to a Resource Q column. DSIF was eluted with a linear salt gradient from 150 mM to 1 M NaCl. Peak fractions were pooled, concentrated using 30 kDa Amicon concentrators (Millipore), and applied to a Superose 6 10/300 column in SEC buffer (20 mM Na-HEPES pH 7.4, 150 mM NaCl, 2 mM DTT). Peak fractions were pooled, concentrated to 7.5 μM (~1.1 mg/mL), glycerol was added to 10% (v/v) before the protein was aliquoted, flash-frozen in liquid nitrogen, and stored at −80 °C.

**Electrophoretic mobility shift assays (EMSAs).** The PEC was formed on a standard nucleic acid scaffold using the following oligos:

RNA (28% GC content) 5′-UCUAGAACUAUUUUUUCUUACACUC (25 nt), template DNA 5′-CTGTAGACTGACCAAGTTGTCCCGTAGGAGTGTAAGAG ATATATGGTAG (49 nt), and non-template DNA 5′-CTACCATATATCTCTTA CACTCCTACGGGACAACTTGGTCAGTCTACAG (49 nt). For the assay that is shown in Supplementary Fig. 2d an alternative RNA with a higher GC content was used: RNA (48% GC content) 5′-GCUGCAACUGUCGUUUCUUACACUC. The sequence of the shorter 15 nt RNA, used in control experiments, matched the 15 nucleotides from the 3′-end of the longer 25 nt RNA. Prior to the assay, 100 pmol of RNA were 5′-end-labeled with [γ-$^{32}$P]-ATP (3000 Ci/mmol, Perkin Elmer) using T4 polynucleotide kinase (NEB) according to the manufacturer's protocol. The labeled RNA was purified by ethanol precipitation and subsequently annealed to 100 pmol of template DNA in an annealing buffer (20 mM Na-HEPES pH 7.4, 100 mM NaCl, 3 mM MgCl$_2$, 10% (v/v) glycerol) in a final volume of 20 μL. To this end, the mixture was heated at 95 °C for 5 min and cooled down to 20 °C in 1 °C/min steps in a thermocycler (as previously described by Vos et al., 2018). After annealing, the sample was diluted 1:1 in DEPC-treated water to achieve a final concentration of 2.5 nM hybrid in 40 μL. Subsequently, all amounts refer to one individual EMSA sample loaded onto one gel lane. EMSA samples were typically prepared in an *n*-fold master mix, which was then split into *n* samples before the addition of different eRNAs. Each incubation step, if not explicitly mentioned, was performed at 30 °C for 15 min. The final sample volume was 8 μL. The PEC was assembled by incubating a pre-annealed RNA-template DNA hybrid (0.8 pmol) with Pol II (1.2 pmol). Then, non-template DNA (1.6 pmol) was added and incubated. Next, the reaction was supplemented with 5x EMSA buffer to achieve a final concentration of 20 mM Na-HEPES pH 7.4, 100 mM NaCl, 25 mM KCl, 3 mM MgCl$_2$, and 2 mM DTT. Subsequently, first DSIF (2.4 pmol) and then NELF (1.2 pmol) were added. Each added protein was incubated with the reaction mixture as described above. Finally, eRNAs were added in increasing amounts (1.2, 2.4, 4.8, 7.2, 9.6, 14.4, 19.2 pmol) to yield final concentrations of 0.15, 0.3, 0.6, 0.9, 1.2, 1.8, 2.4 μM. If an EMSA gel contains a series of six instead of seven different eRNA concentrations, then the highest concentration (2.4 μM) was omitted. The samples were incubated for 15–20 min at room temperature, subsequently supplemented with 1.5 μL of EMSA loading dye (20 mM Na-HEPES pH 7.4, 60% (v/v) glycerol), and loaded on a pre-chilled vertical 3.5% native acrylamide gel (0.5x TBE), which had been pre-run for 30 min at 90 V and 4 °C (used pre-chilled 0.5x TBE running buffer and put gel running chamber on ice during electrophoresis). Samples were separated for 1 h and 30 min. Subsequently, gels were dried and exposed to storage phosphor screens for 3 h or overnight, depending on the level of radioactivity. The phosphor screens were read out by a CR 35 image plate reader (Elysia Raytest). Gels were densitometrically analyzed using ImageJ and the Pol II-DSIF fraction was plotted against the eRNA concentrations using Prism 9.

$$(\text{Pol II} - \text{DSIF})\text{fraction} = \frac{(\text{Pol II} - \text{DSIF})}{(\text{Pol II} - \text{DSIF}) + (\text{Pol II} - \text{DSIF} - \text{NELF})}$$

The pseudo-binding curves were fitted with a single-site quadratic binding equation as

previously described[42]. The obtained apparent $K_d$ values serve as comparative measures between the different experimental conditions. The supershift assay (Supplementary Fig. 2c) was performed under standard EMSA conditions with a final sample volume of 10 μl. The final concentration of *Arc* eRNA 1–200 nt in the eRNA-containing samples was 1 μM. The NELF antibody (anti-NELF-E, #ab170104, Abcam) and the DSIF antibody (anti-SPT5, #sc-133217X, Santa Cruz Biotech) were used in final amounts of 3.3 pmol (2 μl of 1.66 μM; 0.5 μg of undiluted antibody was added) and 4.8 pmol (2 μl of 2.4 μM; 0.72 μg of a 1:5.5 dilution), respectively. This corresponds to a ratio (antibody to target protein) of about 2.75:1 (for NELF) and 2:1 (for DSIF) of the antibody to protein per sample. The NELF titration experiment (Supplementary Fig. 3a) using the four different NELF variants (WT, NELFΔRRM, NELF patch mutant, NELF double mutant) was performed under standard EMSA conditions. NELF variants were added in the following amounts: 0.24, 0.4, 0.8, 1.0, 1.2, 1.6, 2.4 pmol (final concentrations: 0.03, 0.05, 0.1, 0.125, 0.15, 0.2, 0.3 μM).

**Protein–RNA crosslinking coupled to mass spectrometry**

*Sample preparation for crosslinking experiments.* For NELF only samples, 0.5–1 nmol of NELF protein was mixed in an equimolar ratio (1:1) with 4SU-labeled eRNAs (*Nr4a1*-(a) 1–100, 1–200 and *Arc* 1–200) resulting in a final concentration of 2 μM in 1x complex buffer (20 mM Na-HEPES pH 7.4, 100 mM NaCl, 3 mM MgCl$_2$, 1 mM DTT). The mixture was incubated at 25 °C for 20 min and directly crosslinked at 365 nm without prior purification of the formed complex. One of the two *Arc* eRNA (1–200) samples represents an exception to this procedure, as it was additionally purified by gel filtration (Supplementary Fig. 4a; labeled "gel filtration purified"). The overwhelmingly overlapping cross-link pattern between this gel filtration purified sample and its non-purified version confirms that the omission of a gel filtration step does not lead to unspecific crosslinks (Supplementary Fig. 5). Importantly, for all crosslinking experiments, the formation of NELF–eRNA complexes was confirmed by analytical gel filtration. eRNAs were labeled with 4SU during in vitro transcription (as described above). Specifically, 40–80% of UTP were substituted with 4-thio-UTP (NU-1156L, Jena Biosciences). The yield of eRNA transcript was strongly dependent on the percentage of the added 4-thio UTP and the frequency of uracil in the eRNA sequence. Therefore, *Arc*, which comprises 17.5% uracil was more easily transcribed in the presence of 80% 4-thio UTP, while *Nr4a1*-(a), containing longer stretches of uracil (31% in total), only produced RNA at <60% of 4-thio UTP. NELF only samples combined with unlabeled RNAs (*Nr4a1*-(a) (1–200) and poly(GU)$_{40}$ RNA) were mixed and incubated just as described above but using ca. 10x higher amounts of input material and higher final concentrations (10 nmol input and 7.5 μM final concentration for *Nr4a1*-(a) (1–200); 5 nmol and 12 μM final concentration for poly(GU)$_{40}$ RNA). Samples were subsequently purified by gel filtration on a Superose 6 10/300 GL column, to separate NELF-bound eRNA from the free eRNA. NELF–eRNA complex fractions were pooled and concentrated to ca. 1–1.5 mg/mL using centrifugal filters (30 kDa MWCO, Amicon Ultra-4) before they were crosslinked at 254 nm. In the case of *Nr4a1*-(a) (1–200) the NELF–eRNA peak spanned several fractions (A7-B1). Fractions A7-A12 and B1 were concentrated and crosslinked separately. As the final mass spectrometry data for both samples showed no notable differences, the samples were denoted replicates (rep1 originating from fractions A7-A12 fractions, and rep2 originating from fraction B1).

For the PEC-samples, the PEC was assembled stepwise as described for the EMSA on the same nucleic acid scaffold that was used for the pause release assay, except that the nascent RNA was not radiolabeled and that a twofold excess of RNA–DNA hybrid over Pol II was used. The individual components were combined in the following amounts and a final total volume of 450 μL: 1.5 nmol RNA-template DNA hybrid, 3 nmol non-template DNA, 0.75 nmol Pol II, 1.5 nmol DSIF, and 1.5 nmol NELF. Subsequently, the PEC was purified from the unbound proteins and nucleic acids by gel filtration on a Superose 6 10/300 GL column (Supplementary Fig. 4f). The PEC fractions were pooled and concentrated to ca. 0.75 mg/mL (in 250 μL). The concentrated PEC sample was split in two, 4SU-labeled 0.375 nmol *Arc* or *Nr4a1*-(a) 1–200 was added to each half, and samples (total volume: 300 μL) were immediately subjected to crosslinking at 365 nm.

*Protein–RNA crosslinking.* The samples to be crosslinked were split in 50 μL aliquots into a 96-well plate (Nunc™ MicroWell™ 96-Well). The plate was put on a custom-made metal plate and then on ice, to guarantee a constant and uniform distribution of the cooling. For crosslinking of NELF only samples with unlabeled RNAs, the ice-cooled plate was placed at a distance of ~2 cm to the lamps in a Spectrolinker XL-1500 UV Crosslinker (Spectronics Corporation). Samples were irradiated with 3.2 J cm$^{-2}$ of 254 nm UV-C light in four steps of 0.8 J cm$^{-2}$. For crosslinking of samples with 4SU-labeled eRNAs the sample was placed ~1 cm beneath a UV hand lamp (Type: UV- 8 SL, Cat. No. 29 50 740; Herolab GmbH) and irradiated for 15 min with 365 nm UV-A light (30 min in case of the *Nr4a1*-(a) 1–100 40% 4SU sample).

*Sample preparation for mass spectrometric analysis.* Following the crosslinking step at 254 or 365 nm, samples were precipitated by adding 3 M sodium acetate (pH 5.2) and ice-cold (−20 °C) ethanol at 0.1 and three times the sample volumes, respectively, and incubation at −20 °C overnight. On the following day, the pellet was washed with 80% (v/v) ethanol and dried in a vacuum. The samples were dissolved in 50 μl of 4 M

urea in 50 mM Tris (pH 7.9) and further diluted to 1 M urea with 50 mM Tris (pH 7.9). For nuclease digestion, 5 U RNase T1 (Thermo Fisher Scientific) and 5 µg of RNase A (Roche Diagnostics) were added per µg of the sample, and the solutions were incubated at 52 °C for 2 h. Disulfide bonds in RNase-treated samples were then reduced with tris(2-carboxyethyl)phosphine (2.5 mM final concentration) for 30 min at 37 °C, and free cysteines were alkylated by addition of iodoacetamide (5 mM final concentration) for 30 min at 22 °C in the dark. Following the reduction and alkylation steps, the proteins were digested by the addition of sequencing-grade trypsin (Promega) at an enzyme-to-substrate ratio of 1:25. Proteolysis proceeded at 37 °C overnight. Digestion was stopped by the addition of 100% formic acid to 2% (v/v) and digested samples were purified by solid-phase extraction using Sep-Pak tC18 cartridges (Waters). The eluate was evaporated to dryness in a vacuum centrifuge and peptide-RNA cross-link products were enriched by titanium dioxide metal oxide affinity chromatography using 10 µm Titansphere PhosTiO beads (GL Sciences). Samples were redissolved in 100 µl of loading buffer (50% acetonitrile, 0.1% trifluoroacetic acid, 10 mg ml$^{-1}$ lactic acid in water) and added to 5 mg of pre-washed TiO$_2$ beads and incubated for 30 min on a Thermomixer (Eppendorf) at 1200 rpm to keep the beads in suspension. Beads were settled by centrifugation, the supernatant was removed, and the beads were washed, once with 100 µl of loading buffer and once with 100 µl of washing buffer (loading buffer without lactic acid), by shaking for 15 min, followed by centrifugation steps. Bound peptide-RNA conjugates were eluted from the beads in two steps by adding 75 µl of elution buffer (50 mM diammonium hydrogen phosphate, pH 10.5 adjusted with ammonium hydroxide) using the same procedure (15 min incubation with shaking followed by centrifugation). The eluates were combined and immediately acidified with 100% trifluoroacetic acid to a pH of 2–3. The acidified eluates were purified using Stage tips[80] prepared with three plugs of Empore C$_{18}$ disks (3 M), and the eluate was again evaporated to dryness. Dried samples were redissolved in water/acetonitrile/formic acid (95:5:0.1, v/v/v) for mass spectrometry analysis.

*Liquid chromatography-tandem mass spectrometry analysis and identification of protein–RNA crosslinks.* LC-MS/MS analysis was performed on an Easy-nLC 1200 HPLC system coupled to an Orbitrap Fusion Lumos mass spectrometer (both Thermo Fisher Scientific). Peptides were separated on an Acclaim PepMap RSLC C$_{18}$ column (25 cm × 75 µm, Thermo Fisher Scientific) using gradient elution with solvents A (water/acetonitrile/formic acid, 95:5:0.1, v/v/v) and B (water/acetonitrile/formic acid, 20:80:0.1, v/v/v) at a flow rate of 300 nl min$^{-1}$. The gradient was set from 6–40% B in 60 min. The mass spectrometer was operated in data-dependent acquisition mode using the top speed mode and a cycle time of 3 s. The precursor ion scan was performed in the orbitrap analyzer at a resolution of 120,000. MS/MS sequencing was performed on precursors with charge states in the range of +2 to +7, with quadrupole isolation (isolation width 1.2 m/z) and with stepped HCD using normalized collision energies of 23 ± 1.15% (CE23) or 28 ± 2.8% (CE28). Detection of fragment ions was performed in the orbitrap analyzer at a resolution of 30,000 or. Repeated selection of the same precursor m/z was prevented by enabling dynamic exclusion for 30 s after one scan event. For data analysis, the Thermo raw files were converted into mzXML format using msconvert[81] and searched using the crosslinking search engine xQuest[82] (version 2.1.5, available from https://gitlab.ethz.ch/leitner_lab/xquest_xprophet). The protein sequence databases contained the subunits of the NELF complex or the PEC and abundant contaminants identified in the respective samples identified by LC-MS/MS analysis of non-enriched samples. Depending on the type of RNA, different crosslinking products were considered based on preliminary searches and specified as *monolinkmw* in xquest.def: For experiments with eRNA containing unmodified nucleotides, adducts of up to three nucleotides were allowed in their unmodified form or with neutral losses of H$_2$, H$_2$O, or HPO$_3$. For the (GU)$_{40}$ RNA, combinations of neutral losses were also considered. For experiments with eRNA containing 4SU, only adducts that contained at least one 4SU were considered. The main cross-link product of 4SU is identical to U-H$_2$O; additional neutral losses of H$_2$ or H$_2$O were also allowed. xQuest search parameters included: enzyme = trypsin, number of missed cleavages = ≤2, MS mass tolerance = ±10 ppm, MS/MS mass tolerance = ±20 ppm (orbitrap detection), fixed modification = carbamidomethylation on Cys, variable modification = oxidation of Met. Results were additionally filtered with stricter mass error tolerances according to the experimentally observed mass deviation, a minions value of ≥7, a TIC subscore of ≥0.15, and an xQuest ID score of ≥20. The false discovery rate was assessed using reversed sequences of all respective database entries. Crosslinking site localizations were directly taken from the xQuest output. In the case of "NELF-only" samples, only the results of the measurement with a collision energy of 28% (CE28) were used for further analysis, as they showed generally lower FDR rates as compared to CE23. In the case of PEC-samples data from CE23 and CE28 measurements were combined to achieve higher data density. All cross-link identifications are listed in Supplementary Data 4 (NELF-only, CE28) and Data 6 (PEC, CE23, and CE28). For visualization of crosslinks in Fig. 4a–e and Supplementary Fig. 5a, b, redundant spectral counts ("nseen" values in xQuest result tables) that fulfilled the above-mentioned filter criteria were calculated and aggregated for each amino acid position. Aggregated redundant spectral counts per protein target and decoy and per crosslinking experiment are listed in Supplementary Data 5 and 7. For the plot shown in Fig. 4c the amino acid position 1 of NELF-E (Met1) was treated as an outlier and excluded from the calculation, as it was strongly overrepresented in the data and could bias the proportions towards the N-terminus.

**3D modeling of eRNA structures.** About 5000 ab initio 3D structure models of *Arc* and *Nr4a1*-(a) eRNA (1–200) and *Arc* eRNA (1–55) were generated with Rosetta (version 3.13) FARFAR2[45] using the rna_denovo command and a fasta file of the eRNA sequences, along with a dot-bracket file describing the SHAPE-MaP-derived secondary structures (MaxExpect structures from RNAstructure). Scores were extracted from the silent pdb output file. The RMSD of the files relative to the lowest score structure was calculated in Rosetta using the rna_score command with -just_calc_rmsd and -native input parameter. The pdb file with the lowest score was used as a reference. The RNA dimensions of the 20 best scoring structures and of the PEC molecule shown in Supplementary Fig. 6c, d were determined with the PyMOL python script *Draw_Protein_Dimensions.py* available from the PyMOL script library (https://pymolwiki.org/).

**Enhanced UV crosslinking and immunoprecipitation sequencing (eCLIP-seq).** eCLIP-seq was performed as previously described[83]. Briefly, 10 million cells (0, 5, and 25 min KCl-treated) were crosslinked (400 mJ/cm$^2$ of 254 nM UV light) with a UV crosslinker (Stratagene) that took around 5 min, followed by lysis in lysis buffer (10 mM Tris-HCl [pH 7.6], 300 mM NaCl, 0.1% Sodium deoxycholate, 1% Triton X-100, 1 mM EDTA [pH 8.0], 0.5 mM EGTA [pH 8.0], 0.1% SDS, and protease inhibitors). RNA fragmentation was carried out by limited digestion with RNase I (AM2291, Ambion) and DNase I (M6101, Promega) for 5 min at 25 °C, followed by incubation with RNase inhibitor (Y9240, Enzymatics) to stop the reaction. Two different conditions were lysed in lysis buffers containing straight 0.1% SDS only or 1% SDS which was diluted to 0.1% SDS after 5 min incubation. The lysates were then incubated with antibody (NELF-E; ab170104, Abcam)-conjugated magnetic beads overnight at 4 °C. The immune-complexes were washed with each of the following buffers: high salt buffer (20 mM Tris-HCl [pH 7.6], 1 M NaCl, 1% Triton X-100, 2 mM EDTA, 0.5% *N*-lauroylsarcosine, 0.1% SDS, and 2 M Urea), LiCl buffer (10 mM Tris-HCl [pH 7.6], 250 mM LiCl, 1% NP-40, 1% Sodium deoxycholate, 1 mM EDTA, and 2 M Urea), and TET buffer (10 mM Tris-HCl [pH 7.6], 1 mM EDTA, and 0.2% Tween20). In each wash, the beads were incubated with wash buffer containing RNase inhibitor for 10 min at 4 °C. RNAs were dephosphorylated with Alkaline Phosphatase (20 U; M0290S, NEB, Ipswich, MA) and T4 PNK (10 U; M0437M, NEB, Ipswich, MA). Subsequently, a 3′ RNA adapter was ligated onto the RNA with T4 RNA Ligase 1, High Conc. (75 U; M0437M, NEB, Ipswich, MA). Protein–RNA complexes were run on a Bis-Tris SDS-PAGE and the gel containing protein–RNA complexes shifting upwards from the expected size of protein (around 40 up to 150 kD) were cut out. The RNAs were recovered from the gel slices by digesting the protein with proteinase K (8 U; P8107S, NEB, Ipswich, MA) leaving a polypeptide remaining at the crosslinked nucleotide. After precipitation, RNAs were reverse transcribed with SuperScript III Reverse Transcriptase (200 U; 18080-044, Thermo Fisher Scientific), the free primer was removed by Exonuclease I (80 U; M0293S, NEB, Ipswich, MA), and a 3′ DNA adapter was ligated onto the cDNA product with T4 RNA Ligase 1, High Conc. (60 U; M0437M, NEB, Ipswich, MA). Libraries were then amplified with Q5 High-Fidelity Master Mix (M0492S, NEB, Ipswich, MA). Subsequently, PCR products were size-selected in 6% polyacrylamide TBE gel (210–350 bp) and purified. The final libraries were sequenced using Illumina NEXTSEQ 500 according to the manufacturer's instructions. For analysis, adapter sequences of FASTQ reads were trimmed using cutadapt with "-a AGATCGGAAGAGC" of universal Illumina adapter sequences. The trimmed reads were aligned against a repetitive element-masked mouse genome (mm10) using the STAR aligner. PCR duplicate reads on paired-end (R1 and R2) with the same start/endpoints and same mapped inserts were removed using Sambamba. Then, only R2 reads, which contained information about potential truncation events, were used for further analyses. bamCoverage was used to make visualization tracks. The first nucleotides from the eCLIP R2 reads represent the crosslinking sites, which were extracted using htseq-clip under use of the "-e 2 --site s" parameter[84]. To examine the distance between cross-link sites and the defined 5′-ends of eRNAs and TSSs, we used "bedtools intersect". The crosslinking coverage values within 600 bp from TSSs were normalized to the GRO-seq profiles of transcripts that contain crosslinking sites, while also taking their expression level under consideration. Subsequently, coverage profiles were generated. Then we calculated the proportion of the crosslinking sites present in six distance windows (up to 200, 201–400, 401–600, 601–800, 801–1000, or 1001–2000 nucleotides) for enhancer and promoter.

**Chromatin immunoprecipitation (ChIP).** ChIP assays were carried out as previously described with minor modifications[3,18]. At DIV 7, 12 million cultured cortical neurons were treated with the indicated conditions, then fixed in a crosslinking-buffer (0.1 M NaCl, 1 mM EDTA, 0.5 mM EGTA, 25 mM Hepes-KOH, pH 8.0) containing 1% formaldehyde (252549, Sigma-Aldrich, St. Louis, MO) for 10 min at RT. Crosslinking was quenched by glycine (final 125 mM) for 5 min at RT and harvested in PBS protease inhibitors on ice. Pelleted neurons were lysed in ice-cold buffer I (50 mM HEPES-KOH [pH 7.5], 140 mM NaCl, 1 mM EDTA [pH 8.0], 10 Glycerol, 0.5 IGEPAL CA630, and protease inhibitors) to isolate nuclei. Nuclei were sonicated in ice-cold buffer III (10 mM Tris-HCl [pH 8.0], 300 mM NaCl, 0.1 sodium deoxycholate, 1 Triton X-100, 1 mM EDTA [pH 8.0], 0.5 mM EGTA [pH 8.0], and protease inhibitors). The resulting nuclear extracts were centrifuged at 13,200 rpm for 15 min at 4 °C to separate insoluble fraction. The supernatant was then incubated with 2 µg each of anti-NELF-A (sc-23599, Santa Cruz Biotech), anti-NELF-E (ab170104, Abcam), or anti- Pol II (N-20)

(sc-899X, Santa Cruz Biotech) overnight at 4 °C. Protein A/G PLUS Agarose (sc-2003, Santa Cruz Biotech, Santa Cruz, CA) was added and incubated for 2 h at 4 °C. The immune-complexes were pelleted and washed twice with each of the following buffers: low salt buffer (0.1% SDS, 1% Triton X-100, 2 mM EDTA, 20 mM Tris-HCl [pH 8.1], 150 mM NaCl), high salt buffer (0.1% SDS, 1% Triton X-100, 2 mM EDTA, 20 mM Tris-HCl [pH 8.1], 500 mM NaCl), and LiCl buffer (250 mM LiCl, 1% IGEPAL CA630, 1% sodium deoxycholate, 1 mM EDTA, 10 mM Tris [pH 8.1]). In each wash, the beads were incubated with wash buffer for 10 min at 4 °C. The washed beads were then rinsed once with 1x TE (10 mM Tris-HCl [pH 8.0], 1 mM EDTA). The immune-complexes were eluted from the beads twice by elution buffer (10 mM Tris-HCl [pH 8.0], 1 mM EDTA [pH 8.0], 1% SDS) at 65 °C for 10 min. The crosslinking was reversed by incubation at 65 °C for 5–6 h. The resulting eluate was treated with RNase A (10 µg; Qiagen, Hilden, Germany) for 1 h at 37 °C and Proteinase K (4 U; P8107S, NEB, Ipswich, MA) for another 2 h at 55 °C. The DNA was purified by Phenol:Chloroform extraction, followed by PCR purification kit (28106, Qiagen).

**ChIP-seq library construction and data processing.** ChIP-seq library construction was performed using NEBNext ChIP-Seq Library Prep Master Mix Set (E6240, NEB, Ipswich, MA) following the manufacturer's instruction with modifications. Briefly, the end-repaired ChIP DNA fragments were size-selected (100–300 bp), dA-tailed, and then ligated with adapters. The adapter-ligated ChIP DNA fragments were digested by USER enzyme and amplified by 14–16 cycles of PCR. The amplified ChIP DNA library was size-selected (250–350 bp) and proceeded to sequencing. ChIP-seq libraries were sequenced on Illumina HISEQ 2500 with 50-bp or NextSeq 500 instrument with 75-bp single-end reads according to manufacturer's instructions (Illumina) by the UTSW McDermott Next-Generation Sequencing Core and Genomics Core. Adapter and low-quality score sequences of FASTQ reads were trimmed using Trim Galore with default parameters. The trimmed FASTQ reads were aligned to UCSC's mm10 genome using Bowtie2 with default parameters. Duplicated mapped reads were removed using "Sambamba" (https://lomereiter.github.io/sambamba/). Reads with mapping quality less than 10 were removed using SAMtools[85]. Aligned BAM formatted files were converted to the BED format using bedtools. Tag directories for each sample were created using "makeTagDirectory" of HOMER with the respective BED files. To normalize the differences in sequencing depths, mapped reads were "down sampled" to the lowest number of the uniquely mapped reads with duplicates followed by duplicate reads removal using "Sambamba". The bigWig files were generated using "bamCoverage" included in the "Deeptools" package for visualization on the UCSC genome browser platform. The coverage values in bigWig files were normalized to RPGC (Reads per genomic content). ChIP peaks were called using MACS with parameters "--gsize mm --broad" against input chromatin samples as control data. To find overlapping peaks under different conditions, we merged the peaks from different samples, called "merged peaks", using "mergePeaks" of HOMER. To generate ChIP coverage plots, we used 'annotatePeaks.pl' of HOMER.

**Transcriptional pause release assays on magnetic beads.** Pause release assays were adapted from Vos et al., 2018 and performed with a fully-complementary scaffold which is similar to the nucleic acid scaffold used for EMSA experiments. The same RNA sequence (25 nt) was used as for EMSA experiments. The template DNA (5′-GATCAAGCAGTAATCGTTGCGATCTGTAGACTGACCAAGTTGT CCCGTAGGAGTGTAAGAGATATATGG TAGTACC; 76 nt) and the non-template DNA (5′-BiotinTEG-GTCTGGTACTACCATATATCTCTTACACTCC TACGGGACAACTTGGTCAGTCTACAGATCGCAACGATTACT GCTTGATC; 80 nt) sequences were longer than the ones used for the EMSAs in order to allow for the transcription of longer RNA products. Moreover, the non-template DNA had a 4 nt overhang at its 5′-end, which carried a biotin tag to enable binding to streptavidin magnetic beads. The transcription scaffold contains a 9-base pair (bp) DNA-RNA hybrid, 16 nt of exiting RNA bearing a 5′-$^{32}$P label, 17 nt of upstream DNA, and 50 nt of downstream DNA. The radioactively labeled hybrid of RNA and template DNA was prepared as described for the EMSA experiment. Samples were generally prepared in an n-fold master mix (usually a 14x master mix was prepared, which was split into six 2.33x master mixes, from which each master mix was used to produce one time-course transcription experiment). All amounts in the following are related to a 1x mixture. The assembly of the Pol II transcription-competent complex originated from the hybrid of RNA and template DNA (2.5 pmol), Pol II (3.75 pmol), and non-template DNA (5 pmol). It was carried out as described for the EMSA experiment, except for the increased amounts of all components. After a final incubation with non-template DNA, the reaction was supplemented with 5x transcription buffer to achieve 1x transcription buffer conditions of 20 mM Na-HEPES pH 7.4, 150 mM NaCl, 3 mM $MgCl_2$, 10 µM $ZnCl_2$, 4% (v/v) glycerol, and 2 mM DTT in a final volume of 5 µL. Addition of ATP/CTP/UTP (HTP)-nucleotide mix (1 µL, 100 µM) and incubation at 30 °C for 10 min allowed Pol II to transcribe four nucleotides of the implemented G-less cassette before it stalled at +4 position due to GTP omission. Subsequently, DSIF (7.5 pmol) and then NELF (7.5 pmol) were added and each time incubated at 30 °C for 15 min. An *Input* control sample was taken before proceeding to the bead binding step. The master mix (14x) of the assembled PEC complex was then diluted with 1x transcription buffer to a final volume of 180 µL (1.5x of initial bead

volume) and was applied to magnetic streptavidin beads (120 µL beads per 14x master mix) (Dynabeads® MyOne™ Streptavidin C1, ThemoFisher Scientific). Prior to this, beads were three times washed with 1x BW buffer (20 mM Na-HEPES pH 7.5, 100 mM NaCl, 4% (v/v) glycerol, 0.04% (v/v) Tween20, 0.02% (v/v) IGEPAL CA630, 2 mM DTT) and finally resuspended in 120 µL 1x transcription buffer. The binding mix (300 µL total volume) was incubated at room temperature on a tube rotator for 20–30 min. After taking off the supernatant, beads with the bound PEC complexes (usually 1/3 of initial RNA was bound, judged by the radioactivity ratio between supernatant and beads) were washed three times with 1x BW buffer and split into six samples (2.33x of original master mix). The BW buffer was removed, and the beads were resuspended in 60 µL of 5 µM eRNA sample in 0.5x RNA buffer (10 mM Na-HEPES pH 7.4, 50 mM KCl, 0.5 mM EDTA) and incubated at room temperature on a tube rotator for 15 min. Subsequently, the supernatant was taken off, beads were resuspended in 24 µL of 1x Transcription buffer and the 0 min sample (4.3 µL) was taken and quenched with 2x Stop buffer (6.4 M urea, 50 mM EDTA pH 8.0, 1x TTE buffer). Transcription was allowed to resume by the addition of NTPs (3 µL, 100 µM), and the bead mix was immediately returned to a thermomixer and incubated at 30 °C and 900 rpm to avoid sedimentation of the beads. Samples (4.9 µL) were taken after 1, 3, 6, 14, and 25 min and immediately quenched with 2x Stop Buffer (5 µL). Multiple time-course experiments were processed in parallel in a phased manner (10 min phasing). Each sample was proteinase K (5 µg) treated at 37 °C for 30 min and then boiled at 90 °C for 4 min to release the bound molecules from the beads. Upon this treatment, samples were separated on a pre-run (30 min at 400 V) denaturing 15% Urea PAGE gel (0.5xTTE, 16 × 18 cm × 0.4 mm) for 3 h at 500 V (first 30 min at 400 V). Usually, 4–6 µL of samples were loaded depending on the amount of radioactivity. After the gel run, gels were dried and exposed overnight as described for the EMSA experiments. For quantification of pause release, the first band above the triple pausing band (as highlighted in Fig. 6b–d) was quantified by densitometric analysis. The band intensity was normalized against the total intensity of the corresponding lane and plotted against the time (min).

**Statistics and reproducibility.** All electrophoretic mobility shift and pause release assays that are part of the main and supplementary figures were carried out at least twice ($n \geq 2$) with similar results.

**Reporting Summary.** Further information on research design is available in the Nature Research Reporting Summary linked to this article.

## Data availability

The Exo-seq, GRO-seq, ChIP- (NELF-A, NELF-E, and Pan Pol II)-seq, eCLIP-seq, and SHAPE-MaP data generated in this study have been deposited in the NCBI Gene Expression Omnibus (GEO) database under accession code GSE163113. Crosslinking mass spectrometry data generated in this study have been deposited to the ProteomeXchange Consortium via the PRIDE partner repository (dataset identifier PXD030569)[86]. The mRNA-seq data used in this study had previously been generated and are available in the GEO under accession code GSE139309 (GSM4137778-GSM4137789). ChIP- (H3K27ac, Pol II, and CBP)-seq data used in this study are available in GEO under the accession codes GSE60192 and GSE21161. Source data are provided with this paper.

## Code availability

All custom code is available from the authors upon request. Details are provided in the reporting summary.

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

## Acknowledgements

We thank Elizabeth Duncan, Olaf Stemmann, and Alan Cheung for critically reading the manuscript. We thank Dr. Kunz and Marcel Bowens from the Bayerische Landesanstalt für Landwirtschaft (LfL) for access to pig thymus of unparalleled freshness. We thank Dr. Felix Klatt and Silke Spudeit for technical help, Chris Sarnowski for support with the xQuest software, and we are grateful to the Core Unit Systems Medicine of the University of Würzburg for next-generation sequencing. We thank Patrick Cramer, Seychelle Vos, and Carrie Bernecky for providing NELF expression plasmids and for helping in establishing the purification of mammalian Pol II. We further thank Birgitta Wöhrl for the DSIF expression plasmid and Norbert Eichner and Gunter Meister for access to a miSeq instrument. This work was supported by the German Research Foundation (DFG, grants KU 3514/1-1 and KU 3514/3-1), the Oberfrankenstiftung (P-Nr. 05474), the Elite Network of Bavaria, the University of Bayreuth and the Paul Ehrlich and Ludwig Darmstaedter Prize for Young Researchers (to C.-D.K.). This work was also supported by the National Research Foundation of Korea (NRF) grant funded by the Korean government (MSIT), 2019R1A2C2006740 (T.-K.K.), 2019R1A5A6099645 (T.-K.K.), 2017M3A9G7073033 (T.-K.K.), 2020H1D3A1A04104610 (T.-K.K.), and 2020R1I1A1A01067189 (S.-K.K.), the Brain Research Program of the National Research Foundation (NRF) funded by the Korean government (MSIT), 2019M3C7A1031537 (T.-K.K.), Samsung Science & Technology Foundation (SSTF-BA2102-09), and a Simons Foundation Autism Research Initiative-Pilot Award 575147 (T.-K.K.). A.L. acknowledges funding from an ETH Research Grant (ETH-24 16-2) and the ETH Domain Strategic Focus Area "Personalized Health and Related Technologies" (PHRT-503) and would like to thank Paola Picotti (ETH Zurich) for access to instrumentation and infrastructure. The Orbitrap Fusion Lumos mass spectrometer used in this work was purchased using funding from the ETH Scientific Equipment program and the European Union Grant ULTRA-DD (FP7-JTI 115766). Open Access publication is in part funded by the German Research Foundation (DFG, Project No. 491183248) and by the Open Access Publishing Fund of the University of Bayreuth.

## Author contributions

V.G. and C.-D.K. conceived and designed all Exo-Seq, SHAPE-MaP, and in vitro experiments. V.G., F.K., and T.B. acquired data. A.P. and V.G. performed computational analyses. L.-M.S. helped to establish the pause release assay. T.-K.K. and S.-K.K. designed all experiments related to the characterization of NELF and eRNAs in primary neuron culture. S.-K.K. performed all ChIP-seq, GRO-seq, and eCLIP-seq experiments under the guidance of T.-K.K., D.U., and K.K. performed the bioinformatics analysis of ChIP-seq, GRO-seq, and eCLIP-seq through discussion with T.-K.K. and S.-K.K. M.G. and A.L. performed crosslinking/mass spectrometry experiments and performed data analysis. C.-D.K., T.-K.K., V.G., and S.-K.K. drafted the manuscript with input from all authors.

## Funding

## Competing interests

The authors declare no competing interests.
