## [Peer Review File · Nature Communications]

Title: Enhancer RNAs stimulate Pol II pause release by harnessing multivalent interactions to NELFREVIEWER COMMENTS

Reviewer #1 (Remarks to the Author):

I have reviewed the paper titled “Enhancer RNAs stimulate Pol II pause release by harnessing multivalent interactions to NELF” by Vladyslava Gorbovytska et al. Overall this paper demonstrates the mechanistic insight of NELF release from PEC complex by eRNAs. I think this paper has some novel aspects and will be of interest to the broader scientific community. The paper has done a very good job experimenting step by step the key criteria for eRNAs to dissociate NelF from the pause elongation complex (PEC). After taking care of some major/minor concerns, I think this paper has merit for publications in nature communications.

The most important and also most ambiguous part of this paper is the cross-linking sections. For cross-linking experiments, the authors utilized UV cross-linking and provided a list of cross-linked peptides in table 4 after mass spectrometry analysis. XQuest software was used and provided confident scores for the linked peptides. Interestingly, not a single mass spectrum was provided for the cross-linking data either in the main text and or in the supplementary data. UV is a nonspecific cross-linker so few cross-linked MS/MS data must be provided with figures in the supplementary data. Besides, there is no false positive rate calculation (FDR) was provided. With unbound and bound eRNAs before and after cross-linking, data should be provided with decoy search, which can provide confidence of the datasets. Besides, some manual screening of MS/MS data for false-positive ID will be very useful. Previous cross-linking of PEC protein complex was done with BS3 cross-linker, which reacts with the lysine residues, here UV cross-linking was used which cross-linked several amino residues and nucleotides nonspecifically. Any replicate study was done for the reproducibility of the cross-linking experiments?

Other concerns I have about the samples. PEC with eRNA peaks were split into main and shoulder peaks. This is not clear from the supplementary figure 4. Both samples provided a similar cross-linking profile according to the data. Please specify the shoulder peaks.

According to the model in figure 8, eRNA dissociates and is bound to NelF from the PEC complex, should not that be present in samples? I think a cross-linking study of eRNAs and NELF complex in vitro would be very useful for additional insight on the binding sites. This could be a very nice comparison with eRNAs bound to PEC complex. Additionally, just a suggestion, it will be really good to see if a cross-linking study can be done using one of the RRM deleted NELF mutants.

Supplementary figure 8 should be in the main text.

Reviewer #2 (Remarks to the Author):

This manuscript described an interesting in-depth study of the mechanisms of eRNA-mediated transcriptional pause release. The authors found that the single-stranded G residues are critical for

relieving NELF-mediated repression of elongation. CLIP and mass spec studies further elucidated the interactions between eRNAs and NELF. The in vitro studies are also complemented by some in vivo work showing potential relevance during neuronal gene induction upon stimulation. Overall, the in vitro conclusions are supported by strong evidence. However, the in vivo analysis of the pause-release mechanism is mostly correlative, and it is hard to determine the causation. There are several major concerns that need to be adequately addressed.

1. The authors stated that the mechanisms remain under debate (refs 5-8). More recent reviews on this topic should be cited to reflect the status of the field.

2. In Fig. 1d-f, the base pairing probability plot is confusing. I recommend directly plotting the SHAPE reactivity on the structure model, which will give a more direct view of reliability of the structure in the context of experimentally determined constraints. The summary of reactivity in a binary format (structured vs. non-structured) in Fig. 1c is also misleading. A better approach is to plot the distribution of reactivity directly.

3, The author performed SHAPE-MaP on 39 in vitro transcribed eRNAs to determine their secondary structure. However, there are two concerns: 1, the 5'-end structure could be different between the full length eRNA and transcribed 5'-terminal 200nt eRNA; 2, It is well known that RNA secondary structures may be very different in vitro vs. in vivo. Additional experiments on in vivo RNA structure analysis may be needed to strengthen the conclusions.

4, The authors found that there were no common structural motifs among 39 eRNAs. Are their sequences and secondary structures conserved in evolution?

5. Given that eRNAs have a big range of lengths, are their activities enhanced by continuous eRNA elongation, simultaneous with the transcription of the target mRNA genes? Or do their function after finishing their own transcription?

6. eRNA are transcribed bidirectionally, as shown in Fig 1. However, the authors only analyzed one of the two transcripts from each eRNA. How was this selected? Given that the length and single stranded guanines play a much bigger role in releasing NELF, selecting the eRNAs based on abundance alone may lead to biases.

7. The apparent enrichment of crosslinking sites near the 5' end of the eRNAs could be due to the abundance of shorter transcripts, and as a result, this may be giving a false impression that the NELF prefers the 5' end. Ideally, a normalization should be applied to take into consideration of the expression level relative to transcript length.

8. Fig. 4e suggested that appropriate 3D structure of Arc eRNA would render simultaneous eRNA binding to NELF-A/C and NELF-E. How about the other 38 eRNAs? Do they have a common 3D structure? In addition, given the lack of strong constraints for the 3D model, a unique conformation presented in the

figure is misleading at the minimum. Explicit discussions of this limit should be included. Ideally, the authors should present an ensemble of potential 3D conformations to demonstrate the possible interactions.

9. Can the authors speculate on why continuous long transcripts are necessary for optimal dissociation, while higher copy numbers of shorter transcripts do not work as well. Is it due to cooperative binding, and is there any evidence for that? What is the biological implication of such a length dependence? Given the importance of the length, are there any genetic, epigenetic signatures at these enhancers that correlate with eRNA length (more precisely exposed guanines) and their activity?

10. CLIP has inherent bias in crosslinking to nucleotides. What is nucleotides frequency of eRNAs in mass spectrometry data? Is it also elevated level of guanosine interacting with Pol II paused elongation complex?

11. Given that most experiments were performed in vitro, and the in vivo validations are largely correlative. The authors should either perform transgene or mutational analysis in cells, or tone down the claims about the biological relevance of the proposed model. For example, if the Arc enhancer is replaced with poly(GU/GA) in vivo, can its eRNA also perform the similar function?

Reviewer #3 (Remarks to the Author):

In this manuscript titled “Enhancer RNAs stimulate Pol II pause release by harnessing multivalent interactions to NELF” by Gorbovytska et al, the authors discovered a novel mechanism of eRNA affecting target gene activation by releasing the promoter proximal paused RNA polymerase through an allosteric interaction between the eRNA and NELF. This is a novel mechanism that explains, at least partially, how the eRNA molecule itself can mechanistically affect target gene expression, one of the critical questions in the transcription field, and move the field significantly forward.

The authors used a combination of global transcriptome assays to identify eRNAs that are responsive to neuronal stimuli. They probed the structures of these eRNAs using the chemical probing method SHAPE-MaP, and tested their biochemical effects on paused RNA polymerase (Pol 2) complexes (PEC), and discovered that the eRNAs can dissociate Pol 2 pausing factor NELF. They found a novel mode of eRNA-NELF interaction that is distinct from the known NELF RRM domain interaction through crosslinking-mass spectrometry, and also showed in vitro that this interaction leads to the release of PEC to RNA synthesis. They also used eCLIP and to show that these eRNA-NELF interactions take place in vivo, and the association between target gene activation and NELF bound eRNA activation using nascent RNA sequencing (GRO-seq).

Overall, I am positive about publishing this manuscript in Nature Communications in terms of the impact, novelty, and the quality of the study. However, I identified a number of critical points that need to be addressed to fully accept the authors' claims.

Major points.

1. The authors used an elegant biochemistry using EMSA to show that eRNA displaces NELF from the PEC, and revealed the features in the RNA and NELF that are critical for this effect. I raise two concerns in these sets of experiments in Fig 2 and Fig 3.

1.1. Does the sequence of the 25 nt nascent RNA (Fig 2a) affect this effect? The 25 nt nascent RNA appears particularly AU-rich (GC content < 25%) that may not reflect the features in mouse nascent RNA such as the Arc TSS (GC content ~ 50%). It will be necessary to rule out the RNA sequence dependent interaction between eRNA and 25 nt nascent RNA by using at least one other 25 nt nascent RNA sequence, and reproduce Fig 2b.

1.2. On every EMSA gel image, the authors show stoichiometric assembly of PEC in the first 3 lanes, but in some panels, +DSIF lanes show unbound Pol 2 bands (2d,2g,3b,3c, 3d,3e, S3b-right, S3c-right). This appears to be a technical variation, but coincidentally, these are the conditions where eRNA dissociates NELF less efficiently. The unbound band intensity in the +DSIF lanes seems to correlate with the decreased NELF dissociation. It might be possible that excess DSIF unbound Pol 2 can sequester eRNAs and dampen the NELF dissociation. The critical panels, such as Fig 3c,3d, and 3e need to be replicated with a gel that does not show remnant (unbound) Pol 2 band in +DSIF lane.

2. The authors presented UV crosslinking data to claim that eRNA binds to the extended domains of NELF subunits. As presented in Fig 4b, the UV crosslinked sites are spread across the NELF subunits, and it is hard to distinguish if the crosslinking is significantly enriched at specific domains (e.g. NELF tentacle) above non-specific background binding or sites that RNA crosslinking is unlikely to happen (e.g. internal surface of the dimerization domains). A quantitative statistical analysis of the cross-linking sites will be necessary to support their claim from the data presented. It is also unclear if a negative control experiment (no eRNA control) is in the analysis, to distinguish if the crosslinking is from the nascent RNA, not the eRNA. Although nascent RNA is short and largely buried in the PEC complex crosslinked to the Pol 2 subunits, it is possible that nonspecific RNAs can crosslink to the NELF subunits. Therefore the negative control would be critical. I would further suggest 4-sU RNA crosslinking experiment, labeling only the eRNA with 4-sU, and validate that the reactive RNA crosslinking sites are significantly enriched in the NELF subunits.

3. One of the important claims in this manuscript is that eRNA binds to NELF in vivo (Fig 5), and this binding is not only the NELF bound to the nascent eRNA engaged with PEC near the eRNA TSS, but likely to be the NELF from the target gene promoter. The argument is that NELF binding site to the eRNA from the eCLIP data shows more downstream distribution from the eRNA TSSs (> 200 nt). I address two critical points for a more robust argument.

3.1. The enrichment of downstream NELF binding sites in eRNAs needs to be shown with statistical significance (e.g. Fisher exact tests in panels Fig 5d, 5e, 5h, 5i). Also, it is possible that eRNAs have overall lower expression level than mRNA, and the TSS-proximal NELF peak in the eCLIP data may not appear as prominent, simply because of the lower signal to noise ratio (compare the y-axes in Fig 5b and 5c). A more robust analysis is to use the mRNA NELF eCLIP signals from the mRNAs with lower expression levels, matched to the expression levels of the eRNAs by stratified sampling, and show that

eRNA eCLIP positions are still more downstream.

3.2. It is also possible that the eRNAs have multiple TSSs that are more widely distributed, and the downstream NELF eCLIP sites are actually from the NELF bound to TSS-proximal paused eRNAs. For example, I inspected the eCLIP, nascent RNA, and Exo-seq data at the Fosb enhancer site from the presented GEO archive using the reviewer token, and while Fosb enhancer does show a downstream NELF eCLIP site, it also shows some level of downstream Exo-seq reads near the eCLIP site, raising a concern that the downstream eCLIP signals are actually coming from the NELF at eRNA TSS proximal pausing, not the NELF at the target gene promoter. Therefore, it will be critical to show the Exo-seq profile in parallel to NELF eCLIP profiles in Fig 5b/5c, and confirm that most of the other eRNAs do not have significant downstream TSSs. Also, it will be possible to further select only the eRNAs that have a sharp focused TSS from the Exo-seq data, and perform the NELF eCLIP position analysis.

4. The in vitro transcription assay is a critical evidence that the eRNA mediated NELF dissociation leads to gene activation. Overall the effects were modest, but the results were convincing. However, the assay itself may need further clarifications on certain aspects of this biochemical assay.

4.1. The authors assume that there will be the release of NELF through the eRNA addition as seen in the EMSA experiment, but it will be necessary to confirm the release by quantifying NELF protein bound the in vitro transcription template (e.g. using Western blot after the eRNA addition on the bead bound PEC fraction).

4.2. There is a second pause site at ~ 40-45 nt, in addition to the one at ~30 nt that the authors focused on. The nature of this second pause site, whether it is physiological, or an artifact of the biochemical system, will need clarification. Upon NELF release by eRNAs, it appears that the intensity at this second pause site increases while the ~30 nt pause site decreases, and it seems simply a downstream redistribution of pausing. However, I suspect that the second pause site at ~40-45nt is a NELF independent pausing that is closer to productive Pol 2 elongation physiological pausing. The NELF quantification, as suggested above, e.g. at the 45 min timepoint in Fig 6b, can show this NELF independent pausing if the decrease in the bound NELF correlates with the decrease in the ~30nt band intensity.

4.3. It is unclear whether the run-off transcripts are detected. The expected size of the run-off transcripts is ~ 75nt based on the description of the template in the Methods, but it is unclear if the band at ~65 nt in Fig 6b correspond to the run-off transcripts, or whether they are stalled Pol 2 near double strand breaks. Also, it is possible that the second pause site at ~40-45nt may also be affected by the double strand break. It will be useful if the authors can provide the same experiment with longer template (~10-20 nt added to the end), and determine the nature of these bands other than the ~30 nt that the authors focus on.

5. In Fig 7, the authors claim that NELF bound eRNAs are more rapid and responsive to KCl stimulus, and correlated with the mRNA expression at the Pol 2 pause escape level. The timing of the eRNA activation that precedes the mRNA activation (Fig 7b) makes the claim more convincing. However, it is unclear how quantification was performed, in particular, whether the GRO-seq read counts were from the whole body of the genes or enhancer transcription units, or from a defined window from the TSSs. Longer genes may appear to respond slowly if the whole gene body was used, since the Pol 2 elongation speed

is $\sim 2\text{kb}/\text{min}$, and genes longer than $\sim 20\text{kb}$ will not be fully activated after 10 mins of KCl, while eRNAs are shorter and may appear to change more rapidly. But, if the read counts were calculated from a defined window from the TSS in the mRNA and eRNAs equally, this concern is resolved.

Minor comments

1. Part of the next-gen sequencing data seems to be a merged to a larger dataset that may be included in another study that I assume has not been published yet (GSE139309), and I recommend separating the sequencing data used in this study to an independent GEO archive, so that the data can be made open upon the publication of this manuscript in Nature Communications, if it is accepted. It was also somewhat complicated to review the GEO archive that had a mix of data used in this study and the ones used in other studies.
2. Fig 5b/c, red line appears to be the NELF ChIP-seq profile, but unclear in the legend. Also distribution of eCLIP sites (dots) in 5c is difficult to access the downstream spread. It might be better to also show the average density/profile as the ChIP-seq line.
3. In the discussion, the authors' reasoning that "In contrast, a significant number of eRNA crosslinking sites are observed to occur throughout up to $\sim 1,000$ nt regions from their 5'-ends, which coincides with the typical size of an eRNA transcription unit" is apparently an important point, but would need to clarify that these eRNA transcription units are away from mRNA TSS.
4. In Fig 6a, it would be helpful to indicate the lengths of the DNA and nascent RNA templates, as they are different from the EMSA assays, to help reading the gel images.

Response to Reviewers

Before addressing all concerns on a point-by-point basis, we would like to thank all three reviewers for the very detailed and thorough review. All comments were highly appreciated and addressing them substantially improved our manuscript.

Response to Reviewer #1

The most important and also most ambiguous part of this paper is the cross-linking sections. For cross-linking experiments, the authors utilized UV cross-linking and provided a list of cross-linked peptides in table 4 after mass spectrometry analysis. XQuest software was used and provided confident scores for the linked peptides. Interestingly, not a single mass spectrum was provided for the cross-linking data either in the main text and or in the supplementary data. UV is a nonspecific cross-linkers so few cross-linked MS/MS data must be provided with figures in the supplementary data. Besides, there is no false positive rate calculation (FDR) was provided. With unbound and bound eRNAs before and after cross-linking, data should be provided with decoy search, which can provide confidence of the datasets. Besides, some manual screening of MS/MS data for false-positive ID will be very useful. Previous cross-linking of PEC protein complex was done with BS3 cross-linker, which reacts with the lysine residues, here UV cross-linking was used which cross-linked several amino residues and nucleotides nonspecifically. Any replicate study was done for the reproducibility of the cross-linking experiments?

Other concerns I have about the samples. PEC with eRNA peaks were split into main and shoulder peaks. This is not clear from the supplementary figure 4. Both samples provided a similar cross-linking profile according to the data. Please specify the shoulder peaks. According to the model in figure 8, eRNA dissociates and is bound to Nelf from the PEC complex, should not that be present in samples? I think a cross-linking study of eRNAs and NELF complex in vitro would be very useful for additional insight on the binding sites. This could be a very nice comparison with eRNAs bound to PEC complex. Additionally, just a suggestion, it will be really good to see if a cross-linking study can be done using one of the RRM deleted NELF mutants.

(Please note that all underlined parts of the reviewer comments have been addressed in the revised version of the manuscript, as discussed below.)

Reply: We thank the reviewer for her/his valuable suggestions concerning our protein-RNA crosslinking data analyzed by mass spectrometry. To address the reviewer's concerns, we decided to redo and expand our crosslinking experiments for the revised version of our manuscript (as detailed in Fig. 4, Supplementary Fig. 4,5 and the reworked section in the main text). In particular, we added redundant crosslinking data for isolated NELF-eRNA complexes that are of high quality (Fig. 4 and Supplementary Fig. 4,5). Moreover, we corroborated our data on the isolated NELF complex by eRNA crosslinking data to the entire PEC (Fig. 4 and Supplementary Fig. 5). To further improve the efficiency of our crosslinking experiments, we also utilized eRNAs that are labeled with 4-thiouridine (4SU). Taken together, our data are of high quality and redundancy. While they confirm our initial conclusions, their quality now also allows us to more specifically delineate the eRNA binding sites on NELF and they also give first insights into how

pause release by eRNAs and P-TEFb might be interrelated. Please see more detailed replies to the reviewer's mass-spec specific questions below.

Reproducibility of mass spectrometry data: In following the reviewer's suggestion, we not only crosslinked eRNAs to NELF in the presence of the entire PEC complex, but also to the isolated NELF complex (labeled "NELF only" in Supplementary Fig. 4c). Furthermore, we performed crosslinking experiments with different kinds of unlabeled and 4SU-labeled eRNAs at 254 nm for unlabeled eRNAs and at 365 nm for 4SU-labeled eRNAs. Crosslinked RNAs include *Arc* eRNA (1-200), *Nr4a1*-(a) eRNA (1-200) and (1-100), as well as a repetitive poly (GU)₄₀-RNA. A summary of all executed experiments can be found as Supplementary Fig. 4c, data for NELF-A and NELF-E can be found as Supplementary Fig. 5. Please also see Supplementary Tables 4,6 for detailed lists of crosslinks. As is easily apparent from Fig. 4 and from Supplementary Fig. 5, our crosslinking data for **all ten samples is consistent**, a fact that forms a solid basis for our data interpretation. Moreover, crosslinking of native RNAs or 4SU-labeled samples displayed good experimental agreement. This confirms that the replacement of the native uridine by 4SU did not lead to major structural alterations and did not affect the eRNA binding mode to NELF.

False discovery rates (FDR): The dramatically improved data quality in the revised version of this manuscript allowed us to use more sophisticated data analysis strategies. **First**, a larger diversity of nucleotide adducts was considered (up to 3 nucleotides, different neutral losses, other adducts than those containing U (for native bases)). **Second**, a target/decoy approach could be used to estimate false discovery rates. Stringent filtering criteria for spectral quality significantly reduced the number of decoy hits, so that the FDR for the NELF only data sets is close to 0% in most cases. For the PEC, we observed that the data quality for the two different fragmentation settings (different HCD collision energies) was quite different, so that robust FDR determination for the lower collision energy data was difficult to perform at low FDR thresholds. We therefore used the same score cut-off than for the other data sets, which resulted in an FDR closer to 10%. Nevertheless, these data sets also contain both confirmatory and complementary data and were therefore included. **Overall, the data acquired for the revision of this study represent the richest protein-RNA crosslinking data sets generated in the Leitner laboratory (ETH Zurich) and compare very favorably to data reported in the literature.**

Exemplary mass spectra for the crosslinking data: Annotated mass spectra for selected crosslinks are now shown as Supplementary Fig. 4d. The presented spectra represent different crosslinking approaches to native RNA nucleotides and to 4SU-labeled RNAs. Moreover, adducts ranging from single nucleotides to dinucleotides are shown.

Supplementary figure 8 should be in the main text.

Reply: We have incorporated Supplementary Fig. 8 in the main text. It is now Fig. 8.

Response to Reviewer #2

This manuscript described an interesting in-depth study of the mechanisms of eRNA-mediated transcriptional pause release. The authors found that the single-stranded G residues are critical for relieving NELF-mediated repression of elongation. CLIP and mass spec studies further elucidated the interactions between eRNAs and NELF. The in vitro studies are also complemented by some in vivo work showing potential relevance during neuronal gene induction upon stimulation. Overall, the in vitro conclusions are supported by strong evidence. However, the in vivo analysis of the pause-release mechanism is mostly correlative, and it is hard to determine the causation. There are several major concerns that need to be adequately addressed.

1. The authors stated that the mechanisms remain under debate (refs 5-8). More recent reviews on this topic should be cited to reflect the status of the field.

Reply: We thank the reviewer for this suggestion and have included the following references in our revised manuscript:

5. Sartorelli, V. & Lauberth, S. M. Enhancer RNAs are an important regulatory layer of the epigenome. *Nat Struct Mol Biol* **27**, 521–528 (2020).

6. Li, W., Notani, D. & Rosenfeld, M. G. Enhancers as non-coding RNA transcription units: recent insights and future perspectives. *Nature* **17**, 207–223 (2016).

7. Field, A. & Adelman, K. Evaluating Enhancer Function and Transcription. *Annu Rev Biochem* **89**, 1–22 (2020).

8. Lewis, M. W., Li, S. & Franco, H. L. Transcriptional control by enhancers and enhancer RNAs. *Biochem Soc Symp* **10**, 1–16 (2019).

2. In Fig. 1d-f, the base pairing probability plot is confusing. I recommend directly plotting the SHAPE reactivity on the structure model, which will give a more direct view of reliability of the structure in the context of experimentally determined constraints. The summary of reactivity in a binary format (structured vs. non-structured) in Fig. 1c is also misleading. A better approach is to plot the distribution of reactivity directly.

Reply: In following the reviewer's advice, we re-plotted the eRNA secondary structure models and colored the nucleotides according to their SHAPE reactivity (see Fig. 1c,e,f and Supplementary Fig. 1d). Furthermore, we replaced the former binary format plot by a boxplot that shows the distribution of median SHAPE reactivity across all tested eRNAs (Fig. 1d), and we altered the respective part in the manuscript (p. 6).

3. The author performed SHAPE-MaP on 39 in vitro transcribed eRNAs to determine their secondary structure. However, there are two concerns: 1, the 5'-end structures could be different between the full length eRNA and transcribed 5'-terminal 200nt eRNA; 2, It is well known that RNA secondary structures may be very different in vitro vs. in vivo. Additional experiments on in vivo RNA structure analysis may be needed to strengthen the conclusions.

Reply: We thank the reviewer for raising both points that we will address in the following.

Point 1: As the reviewer correctly points out there is the theoretical possibility of altered eRNA structures in longer/“full-length” eRNA molecules. However, most RNA base pairing is found to be local (<200 bp). This is true for RNAs studied by ligation-based RNA structure probing methods (Aw et al. 2016; Sugimoto et al. 2015; Lu et al. 2016), as well as when studying entire mRNAs structuromes (Mustoe et al. 2018). Moreover, due to the lack of a polyA tail and their degradation in 3’->5’ direction, the 5’-ends of eRNAs are the most stable parts of the molecules and, thus, also the ones that most likely exert a critical function. Due to their degenerated 3’-ends a further extension of the studied eRNA towards their hypothetical 3’-ends might also require their experimental verification in neurons. Last, our nucleotide composition analysis (Fig. 2j) revealed that guanosines, critical for NELF dissociation, are only found to be enriched in the 5’-terminal 200 nt of our set of eRNAs. To address the reviewer’s concern, we changed our manuscript as follows (p. 6):

“While long-range RNA-RNA interactions exist, RNA base pairs in diverse lncRNAs and mRNAs predominantly form locally (<200 bp), ...“

Fig. R1. In vivo SHAPE-MaP results for 4 eRNAs of immediate early genes. Mutation data for the respective in vitro dataset are shown on the right to aid comparability.

Point 2: As suggested by the reviewer, we had also attempted to experimentally determine the secondary structures of eRNAs *in vivo*. To that end we treated mouse cortical neurons with 10 mM 1M7 as described (Smola et al. 2015). To accomplish robust 1M7 modification rates of activity-induced eRNAs, we performed this treatment while simultaneously stimulating neurons by the KCl method. To enrich for eRNAs (that are found in nuclei), we subsequently extracted the neurons' nuclei before RNA isolation. Moreover, in hoping to further increase 1M7 modification rates, we also performed the 1M7 treatment under nuclear run-on conditions. However, when applied to four eRNAs of immediate early genes (*Nr4a1*-(a), *Fosb*, *Fos-e2* and *Gadd45b*), neither approach yielded sufficiently high mutation rates that would have allowed us to properly determine the secondary structure of the respective eRNAs (see Fig. R1 on p. 4). Please note that for these *in vivo* SHAPE-MaP experiments we attempted to map eRNA structures for 400 nt.

Taken together, we conclude that cortical neurons – at least in our hands – are refractory to efficient SHAPE modification by 1M7 that would allow for reliable modeling of eRNA structures *in vivo*. This is likely due to a lack of permeability and very rapid and potent quenching of the reagent in the medium (Smola and Weeks 2018). Moreover, we note that eRNAs are short-lived and, as such, mapping their structures using SHAPE reagents is inherently difficult. To reflect our *in vivo* experiments, we added the following sentence to the manuscript (p. 6):

“Unfortunately, eRNAs in cortical neurons proved refractory to sufficient levels of 1M7 modification, thus precluding us from confirming our data *in vivo*.”

Last, we would like to note that in our study we find that eRNA secondary structure is not the determining factor for eRNA functionality, at least in Pol II pause release. We therefore do not envision that further *in vivo* SHAPE data on eRNAs would significantly improve our study.

4. The authors found that there were no common structural motifs among 39 eRNAs. Are their sequences and secondary structures conserved in evolution?

Reply: To address the reviewer's question, we plotted the average phastCons scores of our set of 39 eRNAs in three different clusters of higher organisms (Vertebrates, Placental animals, and Euarchontoglires). While the enhancer center defined as up to 200 bp upstream of eRNA TSSs was reasonably conserved (~0.5 average score) in all clusters, the average conservation scores drop quickly in the eRNA transcription unit (Fig. R2, page 6). The enhancer center is known to be conserved, as it contains transcription factor binding sites. In contrast, this analysis strongly suggests that the eRNA sequences themselves are not conserved. Please note that the analysis presented in Fig. R2 does not take into account actual eRNA expression, which may not be found across all aligned species. In addition to a lack of eRNA sequence conservation, our study demonstrates that even eRNAs from the same enhancer can have distinct structures (*e.g.* compare the secondary structures of *Nr4a1* variants (a) and (b) in Fig. 1e,f). Thus, without chemical RNA mapping information our structure-mapping results discourage all bioinformatics-driven comparison efforts for the analysis of eRNA secondary structure between species.

In summary, we do not envision that, beyond the conservation of unpaired guanosines and the striking length dependence of eRNA function, there are any evolutionarily conserved

sequence or structure features of active eRNAs, at least regarding their function in Pol II pause release.

Fig. R2. Conservation analysis of our set of 39 eRNAs across higher organisms.

5. Given that eRNAs have a big range of lengths, are their activities enhanced by continuous eRNA elongation, simultaneous with the transcription of the target mRNA genes? Or do they function after finishing their own transcription?

Reply: As eRNAs were consistently shown to regulate target gene transcription in a locus-specific manner, they are believed to predominantly act in *cis*. Accordingly, eRNAs cause elevated transcription rates from promoters that are in close contact with the eRNA-producing enhancer locus (Li et al. 2013; Rahnamoun et al. 2018; Lam et al. 2013; Sartorelli and Lauberth 2020). Given their predominant activity in *cis*, their short half-lives (Schwalb et al. 2016) and their rapid degradation by the nuclear exosome (Andersson et al. 2014), it is highly likely that eRNAs exert their function while they are being transcribed. In support, a study by Rahman et al. (Rahman et al. 2016) found that, while nascent eRNAs levels are significant, their steady-state levels are low. In addition, dCas9-based tethering experiments by Sigova et al. (Sigova et al. 2015) showed that eRNAs (>60 nt) can act as a trap for the transcription factor YY1. eRNAs were found to increase YY1's residence time on enhancer DNA, a fact suggesting that eRNAs exert their function while just being transcribed. Last, eRNAs were shown to interact with BRD4 and CBP (Rahnamoun et al. 2018; Bose et al. 2017). In more detail, a correlation analysis of GRO-seq, CBP ChIP-seq and PAR-CLIP data revealed that eRNAs bound to CBP localize to sites of nascent RNA transcription and within regions of chromatin bound by CBP, further supporting a model where eRNAs exert their function while being transcribed. Regarding the dependence of mRNA transcription on the length of the respective nascent eRNA, there is a recent study (Carullo et al. 2020), in which the authors mapped neuronal activity-induced enhancers in different brain cells of rats. They present data resulting from a CRISPR-display approach (Shechner et al. 2015; Sigova et al. 2015) on enhancer RNAs that regulate the *Fos* gene. The authors found that tethering *Fos* enhancer RNAs of various length (150 nt, 350 nt and 450 nt) to the *Fos* e1 enhancer leads to a significant increase in *Fos* mRNA expression. Remarkably, the reported increase in *Fos* mRNA expression was already seen with a 150 nt long eRNA fragment, while extending the eRNA to 350 and 450 nt did not further boost the expression. This finding is entirely in line with our results (Fig. 2,3).

As a note of caution serves the fact that multiple eRNA-dependent action mechanisms have been reported, such as enhancer-promoter looping, transcription factor trapping, modulation of enzymatic activity, and phase separation (Sartorelli and Lauberth 2020). This opens up the possibility that - in different gene contexts - the minimum length requirement or the length-dependent effects of eRNAs might vary depending on the target molecules on which they act.

6. eRNA are transcribed bidirectionally, as shown in Fig. 1. However, the authors only analyzed one of the two transcripts from each eRNA. How was this selected? Given that the length and single stranded guanines play a much bigger role in releasing NELF, selecting the eRNAs based on abundance alone may lead to biases.

Reply: Transcript selection was based on the expression level of each respective eRNA, in analogy to prior studies (Schaukowitch et al., 2014; Bose et al., 2017). Due to their short half-lives we decided to focus on the predominantly expressed eRNA molecule for each enhancer region under consideration. We do not believe that this choice induces biases, as our data revealed no

particular structure, nor other specific sequence features – except the presence of unpaired guanosines - that dictate eRNA functionality (Fig. 2). Furthermore, we see this approach experimentally verified by prior knockdown data on *Arc* (minus) and *Gadd45b* (plus) eRNA strands, for which we observed a reduction of the target gene expression (Schaukowitch et al., 2014). Collectively, these results do not rule out the possibility that the opposite strand has a function, as long as the minimal requirements concerning eRNA length and guanosine content are met. However, these data suggest that eRNA abundance likely correlates with function.

7. The apparent enrichment of crosslinking sites near the 5' end of the eRNAs could be due to the abundance of shorter transcripts, and as a result, this may be giving a false impression that the NELF prefers the 5' end. Ideally, a normalization should be applied to take into consideration of the expression level relative to transcript length.

Reply: We appreciate this insightful comment and fully agree with the possibility of having a false enrichment of the crosslinking sites on eRNAs near the 5'-ends. Following the reviewer's suggestion, we normalized the crosslinking coverage data to the GRO-seq signal to account for differences in the abundance of the individual crosslinked transcripts. As a result of this normalization, the proportion of the crosslinking sites near the 5'-end of eRNAs was reduced, suggesting that the enriched crosslinking sites near the 5'-end of eRNAs may be due to the abundance of shorter transcripts (compare with GRO-seq profile shown in Fig. 7c), as commented by the reviewer (see Fig. 5d,e). Normalized crosslinking proportions were more evenly spread across the length of eRNAs, whereas the crosslinking sites of pre-mRNAs (9,028) remained concentrated near the 5'-end. This data is consistent with the model that NELF interacts with the 5'-ends of nascent pre-mRNAs emerging from transcribing RNA Pol II as a pausing mechanism (Missra and Gilmour 2010; Cheng and Price 2008). Taken together, although the eCLIP-seq analysis does not distinguish the origin of eRNA-interacting NELF (enhancer vs promoter), the broad distribution of the crosslinking sites across the length of eRNAs is in contrast to the 5'-end-concentrated distribution of NELF crosslink sites on pre-mRNAs. This finding supports our model that eRNAs transcribed at enhancers interact with promoter-bound NELF and facilitate its release from paused RNA Pol II. We amended the manuscript as follows (p.13-14):

" To further examine whether or not the distribution of the crosslinking sites is influenced by different abundance of individual nascent transcripts, we normalized the crosslinking numbers by GRO-seq signals. The proportion of the crosslinking sites near the 5'-end of eRNAs became much lower after normalization, resulting in more evenly distributed crosslinking events across the length of eRNAs. On the other hand, pre-mRNAs still exhibit the crosslinking occurring prominently near the 5'-ends (Fig. 5d,e)...."

8. Fig. 4e suggested that appropriate 3D structure of *Arc* eRNA would render simultaneous eRNA binding to NELF-A/C and NELF-E. How about the other 38 eRNAs? Do they have a common 3D structure? In addition, given the lack of strong constraints for the 3D model, a unique conformation presented in the figure is misleading at the minimum. Explicit discussions of this limit should be included. Ideally, the authors should present an ensemble of potential 3D conformations to demonstrate the possible interactions.

Reply: We thank the reviewer for this comment. In support of the crucial nature of eRNA length for triggering NELF release, we utilized Rosetta FARFAR2 (Lyskov et al. 2013; Watkins et al. 2020) and our SHAPE-MaP-derived secondary structure restraints to compute 5,000 3-dimensional models of *Arc* and *Nr4a1*-(a) eRNAs (1-200). As expected for RNA structures this large and without additional structural, *e.g.* cryo-EM information, the 3D models of both eRNAs did not converge to a cluster of low-energy structures. Instead, they showed a median RMSD of about 30Å (Supplementary Fig. 6a). In contrast, the 3D models for the short *Arc* eRNA (1-55) fragment displayed reasonable convergence to an RMSD of 5.3Å (top 1% RMSD = 2.7Å) (Supplementary Fig. 6b). These results underscore the 3-dimensional flexibility of eRNA fragments with a length of 200 nt. As our biochemical experiments (Fig. 2,3) revealed, an increase in eRNA length allows for multivalent eRNA interactions with the PEC. Intriguingly, *Arc* and *Nr4a1*-(a) eRNAs (1-200) fold into structures with dimensions that rival the dimensions of the PEC (Supplementary Fig. 6c). These modeling results are entirely in line with our results from EMSAs (Fig. 2,3) and our protein-RNA crosslinking MS/MS data (Fig. 4).

To incorporate our modeling results into the manuscript, we added the following text to the manuscript (p. 13):

“Last, to further substantiate our crosslinking data and to demonstrate that a single eRNA molecule is indeed able to form multivalent interactions with different parts of the NELF complex, we utilized Rosetta’s FARFAR2 algorithm⁴⁵ in conjunction with our experimentally determined secondary structure restraints to calculate 3-dimensional models of both *Arc* and *Nr4a1*-(a) eRNA structures (Supplementary Fig. 6a,b). These models confirmed that large eRNA molecules (200 nt) can simultaneously bind widely-spaced areas of the PEC to induce NELF dissociation, whereas small eRNA fragments (50 nt) cannot (Supplementary Fig. 6c,d).”

9. Can the authors speculate on why continuous long transcripts are necessary for optimal dissociation, while higher copy numbers of shorter transcripts do not work as well. Is it due to cooperative binding, and is there any evidence for that? What is the biological implication of such a length dependence? Given the importance of the length, are there any genetic, epigenetic signatures at these enhancers that correlate with eRNA length (more precisely exposed guanines) and their activity?

Reply: At this point our data clearly indicate that eRNA length cannot be compensated for by elevated concentrations of smaller eRNA molecules (Fig. 2,3 and Supplementary Fig. 6). We hypothesize that, initially, the positively charged patches on the NELF-AC lobe attract an eRNA molecule to the PEC. eRNA binding to the NELF-AC lobe then facilitates the establishment of further contacts between the eRNA and the NELF-A and/or the NELF-E tentacle (whose affinity towards the RNA is probably lower, see Fig. 4). These contacts then altogether induce stripping off of both tentacles from Pol II-DSIF, which in turn induces NELF dissociation from the PEC. In light of the proposed NELF dissociation mechanism, eRNA molecules require a certain dimension to span from the NELF-AC lobe (patch 2) to the middle of the NELF-A tentacle or to the NELF-E tentacle including the C-terminal RRM domain (Supplementary Fig. 6d). Short eRNA fragments such as *Arc* eRNA (1-55) are too short to span the distance between the AC-lobe and the tentacles. Moreover, they are too rigid to adapt to the surface of the PEC (Supplementary Fig.

6c). Longer eRNA molecules can both span larger distances and they have a greater conformational flexibility, which will increase their chance for adopting a conformation that triggers NELF dissociation. To reveal the exact mechanism and potential cooperativity between different eRNA-NELF binding events, future studies are needed. We plan to address the exact mechanism of how long eRNA molecules induce NELF dissociation, *e.g.* by using FRET approaches to measure protein-RNA distances accurately. We have included our hypotheses into the section “To detach NELF from the PEC, eRNAs must have sufficient length to bridge multiple RNA binding sites on NELF-A and NELF-E”, now part of the discussion in our revised manuscript.

Regarding the reviewer’s question about any genetic and/or epigenetic signatures that correlate with eRNA length and activity, we examined the relationship between the enhancer-specific epigenome profiles H3K27ac and H3K4me1 and eRNA length. We ranked eRNAs based on their length (Fig. R3, top panels, this page) or GRO-seq expression levels (Fig. R3, bottom panels, next page) and examined the levels of the aforementioned epigenetic marks in total or of three different clusters of eRNAs. Overall, the acetylation levels were largely correlated with both eRNA features. This result is consistent with previously known features of histone modification. Histone acetyltransferases often travel with RNA polymerase II to acetylate histones at various regions of genes. It was also shown that enhancer-specific histone modifications are dependent on RNA Pol II activity (Kim and Shiekhattar 2015; Selth et al. 2010). CBP is a coactivator that can interact with many TFs as well as RNA Pol II and is mainly responsible for H3K27ac modification. Moreover, we found enhancer-specific H3K4me1/2 modification also to be highly correlated with the length of eRNA transcripts (Kaikkonen et al. 2013). However, while enhancer-specific epigenome profiles generally correlate with eRNA length and eRNA expression levels, we do not know whether they can also quantitatively represent eRNA activity. Identifying any genetic or epigenetic features at enhancers that tightly correlate with eRNA activity or exposed guanines would require extensive correlative and functional analysis, which we believe is beyond the scope of the current study. Besides, our structural modeling data suggest that eRNA activity might not necessarily increase with length beyond a certain threshold size (*e.g.*, 200 nucleotides in length for triggering NELF dissociation).

Fig. R3. Correlation analysis between both eRNA length and expression level and epigenetic enhancer signatures (such as H3K27ac and H3K4me1). Top panels (on previous page) show the correlation between H3K27ac (left side) or H3K4me1 (right side) and eRNA length. eRNAs were ranked by their length and then grouped into three clusters, according to their length (from long to short). Heatmaps of two replicates (GRO-seq) are shown for each cluster. An average read density profile of the histone mark is shown for each cluster on top of the heatmaps. The bottom panels show the same as described for the top panels. However, here eRNAs were ranked and clustered according to their expression levels.

10. CLIP has inherent bias in crosslinking to nucleotides. What is the nucleotide frequency of eRNAs in mass spectrometry data? Is it also elevated level of guanosine interacting with Poll II paused elongation complex?

Reply: We thank the reviewer for this question. Please note that our reply discusses the nucleotide crosslinking frequencies of our redone and significantly improved mass spectrometry dataset that is now part of the revised manuscript (see also reply to reviewer #1, p. 1-2). First, in all crosslinking experiments that utilize 4-thiouridine-labeled RNAs we only detect 4-thiouridine-(4SU)-crosslinked nucleotides. Second, UV (254 nm)-induced photochemical crosslinking indeed has some bias, for example, cross-linking to uracil is reported to be preferred over the other bases. Our own data (Leitner lab) ranks the prevalence of residues approximately in this order (U > G ~ C >> A). This graduation we also observe in our new mass spectrometry data, where 94% of all unambiguously determined crosslinked mononucleotides are seen on uracil in the sample *Nr4a1*-(a) (1-200) rep 1 (87% in rep 2; compared to 31% U in the *Nr4a1*-(a) (1-200) sequence) (Supplementary Table 4). However, our mass spectrometry data do not show elevated levels for guanine as compared to adenine or cytosine. The U-bias is also met in the distribution of dinucleotide crosslinks, where about 92% of all unambiguously determined crosslinked dinucleotides contain a U (AU, CU, GU, UU). The dinucleotides UU (46%) and GU (35%) are overrepresented as compared to (CU (8%) and AU (10%)) as expected (distribution in *Nr4a1*-(a) 1-200 sequence: UU (35%), GU (33%), CU (31%), AU (22%)). However, it is hard to judge whether

the relatively high level of GU and the ratio between UU and GU is as expected by the above-mentioned crosslinking biases or whether it reflects preferential binding of Gs.

Note that in independent studies, we (the Leitner group and collaborators) have observed that, apart from the intrinsic reactivity of different bases, the spatial proximity and side chain orientation of residues plays a big role. For example, we could show that cross-linking on G may be preferred over crosslinking on an adjacent U if the G engages in π - π stacking with aromatic residues on the protein (see ChemRxiv preprint of Knörlein et al., 2021, DOI: 10.33774/chemrxiv-2021-05zhj; a slightly modified manuscript of the preprint is currently under review at Nat Commun). Considering these insights, different types of biases exist and may be at least partially offset by favorable spatial orientation. For MS analysis of cross-linking products, potential biases in nuclease cleavage efficiency will also need to be considered. In summary, it is difficult to distinguish the different biases that might be at play throughout the entire experimental procedure.

11. Given that most experiments were performed *in vitro*, and the *in vivo* validations are largely correlative. The authors should either perform transgene or mutational analysis in cells, or tone down the claims about the biological relevance of the proposed model. For example, if the Arc enhancer is replaced with poly (GU/GA) *in vivo*, can its eRNA also perform the similar function?

Reply: We appreciate the reviewer's concern and agree that the proposed functional study would strength our model. However, we feel that this is well beyond the scope of the current study. For this reason, we rephrased the text below to tone down our claim about the *in vivo* relevance of our results (p. 14):

“... Taken together, our eCLIP-seq analysis provides the correlative *in vivo* evidence that eRNAs are capable of making contacts with NELF associated with paused RNA Pol II at promoters following their synthesis at enhancers...”

Response to Reviewer #3

Major points

1. The authors used elegant biochemistry using EMSA to show that eRNA displaces NELF from the PEC, and revealed the features in the RNA and NELF that are critical for this effect. I raise two concerns in these sets of experiments in Fig 2 and Fig 3.

1.1. Does the sequence of the 25 nt nascent RNA (Fig 2a) affect this effect? The 25 nt nascent RNA appears particularly AU-rich (GC content < 25%) that may not reflect the features in mouse nascent RNA such as the Arc TSS (GC content ~ 50%). It will be necessary to rule out the RNA sequence dependent interaction between eRNA and 25 nt nascent RNA by using at least one other 25 nt nascent RNA sequence, and reproduce Fig 2b.

Reply: To rule out the possibility that the sequence of the nascent RNA is affecting the observed eRNA-driven NELF dissociation through base-pairing interactions between both RNAs, we performed an EMSA with an alternative nascent RNA. To that end, we used an additional 25 nt long nascent RNA with a higher GC-content in our EMSA setup.

- Standard nascent RNA [28% GC content]: UCU AGA ACU AUU UUU UCU UAC ACU C
- High GC nascent RNA [48% GC content]: GCU GCA ACU GUC GUU UCU UAC ACU C

As can be seen when comparing Fig. 2b with Supplementary Fig. 2d (plotted below), no effect of the nascent RNA on NELF dissociation for three differently sized Arc eRNAs was observed.

1.2. On every EMSA gel image, the authors show stoichiometric assembly of PEC in the first 3 lanes, but in some panels, +DSIF lanes show unbound Pol 2 bands (2d,2g,3b,3c, 3d,3e, S3b-right, S3c-right). This appears to be a technical variation, but coincidentally, these are the conditions where eRNA dissociates NELF less efficiently. The unbound band intensity in the +DSIF lanes seems to correlate with the decreased NELF dissociation. It might be possible that excess DSIF unbound Pol 2 can sequester eRNAs and dampen the NELF dissociation. The critical panels, such as Fig 3c,3d, and 3e need to be replicated with a gel that does not show remnant (unbound) Pol 2 band in +DSIF lane.

Reply: We thank the reviewer for catching this. To dispel any doubts about our conclusions, we repeated the EMSA experiments of critical panels Fig. 3c,d,e (and for Fig. 2g), as suggested by the reviewer. This repetition revealed efficient DSIF binding to Pol II, thus proving evidence that the previously observed inefficient DSIF binding to Pol II was due to technical variation. (We note here that the endogenous purification of Pol II from pig thymus is prone to lead to technical variation, depending on the specific source of thymus). Please see below a comparison between the previous EMSA experiments and the ones that are now part of the revised version of the manuscript.

2. The authors presented UV crosslinking data to claim that eRNA binds to the extended domains of NELF subunits. As presented in Fig 4b, the UV crosslinked sites are spread across the NELF subunits, and it is hard to distinguish if the crosslinking is significantly enriched at specific domains (e.g. NELF tentacle) above non-specific background binding or sites that RNA crosslinking is unlikely to happen (e.g. internal surface of the dimerization domains). A quantitative statistical analysis of the cross-linking sites will be necessary to support their claim from the data presented. It is also unclear if a negative control experiment (no eRNA control) is in the analysis, to distinguish if the crosslinking is from the nascent RNA, not the eRNA. Although nascent RNA is short and largely buried in the PEC complex crosslinked to the Pol 2 subunits, it is possible that nonspecific RNAs can crosslink to the NELF subunits. Therefore, the negative control would be critical. I would further suggest 4SU RNA crosslinking experiment, labeling only the eRNA with 4SU, and validate that the reactive RNA crosslinking sites are significantly enriched in the NELF subunits.

Reply: As detailed in our response to reviewer #1 (see p. 1-2 of this document), we were able to substantially improve our mass spectrometry experiments by collection of novel protein-RNA crosslinking data for eight experimental conditions (Supplementary Fig. 4a). The improvements are shown in Fig. 4, Supplementary Fig. 4,5 and in the Supplementary Tables 4-7 of the revised manuscript. Briefly, we added data for isolated NELF-eRNA complexes to our study and corroborated these with new eRNA crosslinking data to the entire PEC. To further improve the efficiency and specificity of our crosslinking experiments, we also utilized eRNAs that are labeled with 4-thiouridine (4SU), as suggested by the reviewer. Crosslinking of 4SU-labeled eRNAs to the paused elongation complex (PEC), that is assembled with an unlabeled nascent RNA, followed by its irradiation at 365 nm now assures that only the labeled eRNA will be crosslinked via 4SU. In contrast, nascent RNA, containing only native bases, will not crosslink at the chosen wavelength. This experimental scenario renders further negative controls dispensable. Moreover, a comparison of the results of unlabeled and 4SU-labeled eRNAs (*e.g.* Fig. 4b,d) shows that no specific binding sites are exclusively found with unlabeled eRNAs, further confirming that the discussed crosslinking patterns are unambiguously derived from eRNAs bound to NELF and the PEC.

Taken together, our mass spec data are now of exceptional quality and redundancy, resulting in defined crosslinking clusters concentrated on NELF subunits -A and -E. This increase in quality allowed us to search for a larger diversity of RNA adducts (up to 3 nt, see Methods). Moreover, we could now apply rigorous target/decoy approaches with estimated false discovery rates (FDR) of 0% for eRNA crosslinks to NELF and of 10% for eRNA-PEC crosslinks (see manuscript, Methods and Supplementary Tables 4-7). Overall, the data acquired for this study represent the richest protein-RNA cross-linking data sets generated in the Leitner Laboratory at ETH Zurich as of now. As mentioned in response to reviewer #1, our data compares very favorably to data reported in the literature, and, to the best of our knowledge, our crosslinking data are the first reported for a complex between RNAs with a length up to 200 nt and protein complexes up to 0.9 MDa (the PEC complex). Last, while our novel data confirm all our initial conclusions, their quality now also allows us to more specifically delineate the eRNA binding sites on NELF and they also give first insights into how pause release by eRNAs and P-TEFb might be interrelated (please see entirely rewritten mass spec paragraph and discussion in the manuscript).

3. One of the important claims in this manuscript is that eRNA binds to NELF *in vivo* (Fig 5), and this binding is not only the NELF bound to the nascent eRNA engaged with PEC near the eRNA TSS, but likely to be the NELF from the target gene promoter. The argument is that NELF binding site to the eRNA from the eCLIP data shows more downstream distribution from the eRNA TSSs (> 200 nt). I address two critical points for a more robust argument.

3.1. The enrichment of downstream NELF binding sites in eRNAs needs to be shown with statistical significance (*e.g.* Fisher exact tests in panels Fig 5d, 5e, 5h, 5i).

Reply: We have now included statistics as the reviewer suggested. We used chi-squared tests to determine the significance of differences in NELF crosslinking positions within eRNAs by comparing the number of crosslinking sites between pre-mRNAs and eRNAs in three or six distance windows at each time point (Fig. 5f-i and Supplementary Fig. 8c-f in this revision).

Comparisons were either carried out for total pre-mRNAs and eRNAs (Fig. 5f,g and Supplementary Fig. 8c,d), or only for those transcribed from NELF-bound promoters/enhancers (NELF-bound pre-mRNAs or eRNAs) (Fig. 5h,i and Supplementary Fig. 8e,f). For both populations we found statistically significant differences in the proportions of crosslinking windows between pre-mRNAs and eRNAs at either KCl 10 min or KCl 30 min conditions, respectively. The observed differences were mainly due to changes in the crosslinking positions toward downstream regions in eRNAs as compared to pre-mRNAs, further supporting our conclusions. For clarity, we removed Fig. 5f,j and Supplementary Fig. 5h,k from the original and significantly revised the entire results section entitled “NELF directly binds to nascent eRNAs *in vivo*”.

Also, it is possible that eRNAs have overall lower expression level than mRNA, and the TSS-proximal NELF peak in the eCLIP data may not appear as prominent, simply because of the lower signal to noise ratio (compare the y-axes in Fig 5b and 5c). A more robust analysis is to use the mRNA NELF eCLIP signals from the mRNAs with lower expression levels, matched to the expression levels of the eRNAs by stratified sampling, and show that eRNA eCLIP positions are still more downstream.

We appreciate this critical comment. As suggested by the reviewer, we newly analyzed the NELF eCLIP-seq data by selecting pre-mRNAs whose expression levels were comparable to eRNAs. We first generated box plots to compare the expression levels between pre-mRNAs and eRNAs, and selected the median values of eRNA expression in the 10 min KCl condition. The box limits indicate the 25th and 75th percentiles containing the median. We then selected 1,632 pre-mRNAs whose expression levels were within the box limits of eRNAs. Subsequently, we generated the coverage profiles of the crosslinking sites from eCLIP-seq for 1,632 pre-mRNAs. The NELF crosslinking sites of these pre-mRNAs were still biased toward the 5'-ends, suggesting that preferential NELF crosslinking of pre-mRNA near the 5'-ends occurs regardless of the expression level difference. We added this new data in Supplementary Fig. 8g,h.

Moreover, we also added the below to the text (p. 14):

“eRNAs are typically expressed at much lower levels than pre-mRNAs, but the difference in RNA abundance does not appear to have an impact on the crosslinking patterns, as 1,632 pre-mRNAs whose expression levels were comparable to those of eRNAs (25-75 percentile range of eRNA expression levels), still exhibit the 5'-end-enriched crosslinking pattern (Supplementary Fig. 8g,h)....”

3.2. It is also possible that the eRNAs have multiple TSSs that are more widely distributed, and the downstream NELF eCLIP sites are actually from the NELF bound to TSS-proximal paused eRNAs. For example, I inspected the eCLIP, nascent RNA, and Exo-seq data at the Fosb enhancer site from the presented GEO archive using the reviewer token, and while Fosb enhancer does show a downstream NELF eCLIP site, it also shows some level of downstream Exo-seq reads near the eCLIP site, raising a concern that the downstream eCLIP signals are actually coming from the NELF at eRNA TSS proximal pausing, not the NELF at the target gene promoter. Therefore, it will be critical to show the Exo-seq profile in parallel to NELF eCLIP profiles in Fig 5b/5c, and confirm that most of the other eRNAs do not have significant downstream TSSs. Also, it will be possible

to further select only the eRNAs that have a sharp focused TSS from the Exo-seq data, and perform the NELF eCLIP position analysis.

Reply: We appreciate the reviewer's insightful comment. We initially defined individual eRNA transcription units from GRO-seq data by a *de novo* transcript calling method using HOMER, from which the crosslinking positions were determined. We used 30 min KCl GRO-seq sample to define each eRNA unit *de novo*. To address the reviewer's concern, we compared the distribution profile of the 5'-ends (TSSs) defined by Exo-seq with the crosslinking profiles. The distribution of raw Exo-seq reads was significantly biased toward the very 5'-ends of crosslinked eRNAs. The Fosb example in the track file that the reviewer mentioned showed raw Exo-seq reads. However, individual raw reads cannot be considered as specific TSSs, as they can be caused by intrinsic noise resulting from random RNA fragments. Therefore, we first used a capped small RNA-seq (csRNA-seq) analysis program, which filters out noise and defines only significant clusters of reads (peaks) representing TSSs (Duttke et al. 2019), see also: <http://homer.ucsd.edu/homer/ngs/csRNAseq/index.html>. We then plotted 120 eRNAs having one or more crosslinking sites individually with called TSS peaks (Supplementary Fig. S9 in this revision). The Exo-seq-called TSSs (purple line) were densely positioned close to the 5'-ends of eRNAs determined by *de novo* transcript calling of GRO-seq sample (set to 0 bp position). Although a minor population of Exo-seq-called peaks were also located in the downstream region, the eRNA crosslinking sites (green line) were more prominently spread along the length of eRNAs. For comparison, we also analyzed pre-mRNAs. We used TSS positions provided by the GENCODE annotation database and examined the distribution of crosslinking sites and Exo-seq read clusters (peak). Pre-mRNAs showed more biased crosslinking sites toward the annotated 5'-ends compared to eRNA crosslinking sites. Taken together, this analysis demonstrates that eRNAs interact with NELF more broadly along their entire length, whereas pre-mRNAs primarily interact with NELF through their 5'-ends, which is consistent with the model that, following their synthesis, eRNAs can interact with NELF associated with RNA Pol II paused at the promoters.

We have added the following to our manuscript (p. 14):

“3. A comparison with Exo-seq profile further confirmed that eRNA crosslinking results from NELF interactions at various positions along the length of transcribed eRNAs, not just near the 5'-ends of short eRNAs transcribed from alternative downstream TSSs at enhancers. Both the Exo-seq raw read density and the TSS peaks called by an Exo-seq read clustering algorithm were highly enriched near the 5'-ends of eRNAs defined *de novo* from GRO-seq data, with only a minor population present in the downstream regions (Fig. 5b,c and Supplementary Fig. 8a,b, and 9a,b, respectively). Taken together, our eCLIP-seq analysis provides the correlative *in vivo* evidence that eRNAs are capable of making contacts with NELF associated with paused RNA Pol II at promoters following their synthesis at enhancers.”

4. The *in vitro* transcription assay is critical evidence that the eRNA-mediated NELF dissociation leads to gene activation. Overall, the effects were modest, but the results were convincing. However, the assay itself may need further clarifications on certain aspects of this biochemical assay.

4.1. The authors assume that there will be the release of NELF through the eRNA addition as seen in the EMSA experiment, but it will be necessary to confirm the release by quantifying NELF protein bound the in vitro transcription template (e.g. using Western blot after the eRNA addition on the bead bound PEC fraction).

Reply: As suggested by the reviewer, we had attempted to verify the release of NELF in our transcription assay setup by measuring NELF-E release after eRNA addition by Western Blotting.

Fig. R4. Western Blot analysis of NELF release upon transcriptional induction. The pause release assay was performed as described in Methods and outlined in Fig. 6a. Different concentrations of *Nr4a1-(a)* eRNA (1-200) were tested (0, 5, 50, 500 nM and 5 μM) to trigger NELF dissociation. After incubation of the PEC (bound to magnetic beads) with the different eRNA concentrations the eRNA-containing supernatant was removed and analyzed for the presence of NELF by Western blotting using a NELF-E-specific antibody. The blot also shows an input sample that contains the magnetic beads before eRNA addition and an additional NELF positive control (30 ng of purified NELF). [The

input beads is approximately 130 ng]. The presence of Pol II in the supernatant and on the beads was detected by and RPB1-specific antibody.

Fig. R4 demonstrates that NELF is indeed released from the PEC upon eRNA addition. However, the Western blot also shows that only a small fraction of NELF seems to be released from the PEC upon eRNA addition. This might explain the rather mild transcription induction effects we see upon eRNA addition (Fig. 6), in contrast to the efficient removal of NELF from the PEC in our EMSAs (Fig. 2,3). Another possible explanation for the apparent “over”-retention of NELF on the utilized magnetic beads (Dynabeads MyOne Streptavidin C1) might be its unspecific binding to the bead surface, which likely occurs due to the large proportion of low-complexity domains in NELF, e.g. in form of the NELF-A and NELF-E tentacles.

We further hypothesize that the low efficiency of NELF removal by eRNAs in our transcription assays is due to the transcriptional engagement of RNA polymerase II, which may alter the efficiency with which NELF can be removed from the PEC, e.g. by subtle changes to the Pol II structure. Please also note that DSIF and NELF binding to Pol II were found to induce a tilted active site in Pol II (Vos et al. 2018), which may hinder transcriptional induction after NELF dissociation *in vitro*. Moreover, we envision that due to the execution of the transcription assay “on beads” steric effects might play an additional role. To better include our thoughts on the differential NELF dissociation efficiencies between our EMSA and transcription assay setup, we added the following to our manuscript (p. 15):

“This is likely due to the active engagement of Pol II in transcription with its active site in a tilted conformation due to NELF and DSIF binding, as shown before (Vos et al. 2018). Thus, jump-starting transcription after eRNA addition may be less efficient under our *in vitro* transcription assay conditions as compared to our EMSA setup (Fig. 2,3).”

4.2. There is a second pause site at ~ 40-45 nt, in addition to the one at ~30 nt that the authors focused on. The nature of this second pause site, whether it is physiological, or an artifact of the biochemical system, will need clarification. Upon NELF release by eRNAs, it appears that the intensity at this second pause site increases while the ~30 nt pause site decreases, and it seems simply a downstream redistribution of pausing. However, I suspect that the second pause site at ~40-45nt is a NELF independent pausing that is closer to productive Pol 2 elongation physiological pausing. The NELF quantification, as suggested above, e.g. at the 45 min timepoint in Fig 6b, can show this NELF independent pausing if the decrease in the bound NELF correlates with the decrease in the ~30nt band intensity.

Reply: We thank the reviewer for raising this point. The “second pause site” is indeed a NELF-independent “intrinsic pausing” site of Pol II that is a result of the utilized DNA template sequence in our experimental assay system. We can substantiate this claim by carrying out a transcription assay in presence of all NTPs (not only CTP, ATP and UTP) at time point 0 min. In doing so, we would expect no pausing at all, however, we detect the “intrinsic pausing” site even under these experimental conditions (Fig. R5). This is the reason why we focused on the initial pausing site at 30 nt for our assay quantifications.

Fig. R5. “Pause”-release assay in presence of all NTPs (instead of HTPs).

In further support, we observed no additional pause site when we performed the assay with an alternative DNA template (Fig. R6, see next page).

Fig. R6. “Pause”-release assay with an alternative DNA template.

In summary, to clarify the nature of the second pause site, we added to our manuscript (p. 15):

“We note that the second observed pause site at ~40-45 nt is a NELF-independent intrinsic pausing site that is caused by our experimental assay system.”

4.3. It is unclear whether the run-off transcripts are detected. The expected size of the run-off transcripts is ~ 75nt based on the description of the template in the Methods, but it is unclear if the band at ~65 nt in Fig 6b correspond to the run-off transcripts, or whether they are stalled Pol 2 near double strand breaks. Also, it is possible that the second pause site at ~40-45nt may also be affected by the double strand break. It will be useful if the authors can provide the same experiment with longer template (~10-20 nt added to the end), and determine the nature of these bands other than the ~30 nt that the authors focus on.

Reply: Indeed, the expected size for a run-off transcript is exactly 75 nt. However, the uppermost band in our pause release assay runs at the height of about 65 nt and thus does not represent the run-off transcript. We hypothesize that, as suspected by the reviewer, this band might originate from stalling before the double strand break. The assay shown in Supplementary Fig. 10a demonstrates that we are able to detect run-off transcripts of ~80 nt under our experimental conditions. However, this is only possible in the absence of NELF. Taking into account the observed inefficient NELF removal under our transcription assay conditions (Reviewer #3, 4.1), we thus conclude that our transcription run-off bands at 65 nt represent the best possible results under conditions of only partial removal of NELF from the PEC.

5. In Fig 7, the authors claim that NELF bound eRNAs are more rapid and responsive to KCl stimulus, and correlated with the mRNA expression at the Pol 2 pause escape level. The timing of the eRNA activation that precedes the mRNA activation (Fig 7b) makes the claim more convincing. However, it is unclear how quantification was performed, in particular, whether the GRO-seq read counts were from the whole body of the genes or enhancer transcription units, or from a defined window from the TSSs. Longer genes may appear to respond slowly if the whole gene body was used, since the Pol 2 elongation speed is $\sim 2\text{kb}/\text{min}$, and genes longer than $\sim 20\text{kb}$ will not be fully activated after 10 mins of KCl, while eRNAs are shorter and may appear to change more rapidly. But, if the read counts were calculated from a defined window from the TSS in the mRNA and eRNAs equally, this concern is resolved.

Reply: We thank the reviewer for requesting a clarification about the rapid induction kinetics of eRNAs compared to pre-mRNAs, which is one of the features of eRNAs that is compatible with our model. We and others had previously shown that eRNAs exhibit a more rapid and transient induction than pre-mRNAs from their target genes in various enhancer-gene contexts (DeSanta and Natoli 2010; Schaukowitch et al. 2014; Arner et al. 2015; Shii et al. 2017; Hsieh et al. 2014; Kim et al. 2010; Hah et al. 2013). However, we agree with the reviewer's criticism, as the GRO-seq read counts in Fig. 7b of the original manuscript were based on the whole body of genes or enhancer transcription units.

Thus, for a robust comparison of the transcription induction kinetics between promoters and enhancers and to address the reviewer's concern, we analyzed the GRO-seq signals within defined windows (1 or 10 kb) from the TSSs of promoters and enhancers. In contrast to the analysis of total GRO-seq signal, this analysis allows for an unbiased comparison of eRNAs (1-2 kb) and pre-mRNAs (typically much longer). Moreover, each analyzed gene group [(KCl-up/NELF-bound pre-mRNAs (548), NELF-bound eRNAs (144), NELF-unbound eRNAs (108))] was subjected to k-means clustering to examine potential changes in the proportion of genes grouped within distinct induction kinetics clusters that might be caused by the defined windows (Fig. R7, next page). In using this strategy, we found NELF-bound enhancers to still exhibit faster induction kinetics than NELF-unbound enhancers. However, while the proportion of eRNAs within the fast induction cluster was not altered by the 1 kb vs. 10 kb window analysis, this was the case for pre-mRNAs that showed a high level of NELF binding. Here, some pre-mRNAs formed a cluster of genes with fast induction kinetics only when being analyzed using a defined window of 1kb (Fig. R7, next page).

Taken together, we therefore agree with the reviewer that the measurement of induction kinetics could be influenced by the size of defined window from the TSSs. For this reason, although a relatively larger proportion of eRNAs still show fast induction kinetics compared to pre-mRNAs, we decided to remove the description of our claim about the difference in the induction kinetics between pre-mRNAs and eRNAs in the result section of main text and related figures (Fig. 7b and Supplementary Fig 7b) from the original submission. Moreover, the relevant description about the kinetics difference in the discussion has been revised as below (p. 20).

".... Another feature of eRNAs compatible with our model is their fast induction kinetics. Several studies have shown relatively faster induction kinetics exhibited by eRNAs compared to mRNAs in various enhancer-gene contexts^{4,18,50,65-68}. Therefore, whether or not eRNA production is under the control of

NELF activity, a population of eRNAs transcribed before pre-mRNA production rises can be available for interaction with promoter-bound NELF.....”

Fig. R7. Heatmaps and k_means clusters of KCl-up/ NELF-bound pre-mRNAs (548), NELF-bound eRNAs (144) or -unbound eRNAs (108), determined based on GRO-seq data in response to 0, 10, 30 and 60 min KCl treatment. The read counts were calculated from the same defined windows (1 or 10 kb) relative to the TSS of promoters or enhancers. Clusters with fast induction rates are labeled and colored in green.

Minor comments

1. Part of the next-gen sequencing data seems to be merged to a larger dataset that may be included in another study that I assume has not been published yet (GSE139309), and I recommend separating the sequencing data used in this study to an independent GEO archive, so that the data can be made open upon the publication of this manuscript in Nature Communications, if it is accepted. It was also somewhat complicated to review the GEO archive that had a mix of data used in this study and the ones used in other studies.

Reply: A portion of the next-generation sequencing data that was utilized in this manuscript has been published in May 2021. It is now available from the GEO database (<https://www.ncbi.nlm.nih.gov/geo/query/acc.cgi?acc=GSE139309>). The Data Availability Section has been altered accordingly.

2. Fig 5b/c, red line appears to be the NELF ChIP-seq profile, but unclear in the legend. Also, the distribution of eCLIP sites (dots) in 5c is difficult to access the downstream spread. It might be better to also show the average density/profile as the ChIP-seq line.

Reply: As suggested by the reviewer, we revised those figures and the legends accordingly (Fig. 5b,c and Supplementary Fig. 8a,b in this revision).

3. In the discussion, the authors' reasoning that "In contrast, a significant number of eRNA crosslinking sites are observed to occur throughout up to ~1,000 nt regions from their 5'-ends, which coincides with the typical size of an eRNA transcription unit" is apparently an important point, but would need to clarify that these eRNA transcription units are away from mRNA TSS.

Reply: In our eRNA analysis, the *de novo* transcript units of eRNAs were first called and then examined for any overlap with annotated mRNAs. The eRNAs described in the manuscript do not overlap with any mRNAs. To further support this, please find a histogram of the distances between the eRNA transcription units and the nearest mRNA TSSs as Fig. R8 (next page).

For clarification, we rephrased the main text as below (p.19 in manuscript):

"... In contrast, a significant number of eRNA crosslinking sites were observed up to ~1,000 nt downstream from their 5'-ends, a finding that coincides with the typical size of an eRNA transcription unit¹¹. As we had excluded all eRNAs that overlapped with mRNA TSSs from the analysis, this result indicates that, unlike pre-mRNAs, eRNAs interact with NELF during and even after the synthesis of their full-length transcripts..."

Fig. R8. Distances between eRNAs and nearest mRNA TSSs. The boxplot on the left shows the distribution of distances between called TSSs of eRNAs (eTSSs) and annotated TSSs of mRNAs. The right panel shows log10 scale of the same distances on the y axis. eTSSs were at least 2 kb ($\log_{10} = 3.3$) away from annotated TSSs. The median distance was ~35 kb.

4. In Fig 6a, it would be helpful to indicate the lengths of the DNA and nascent RNA templates, as they are different from the EMSA assays, to help reading the gel images.

Reply: We addressed the reviewer’s comment by adapting Fig. 6a. Nucleotide positions are indicated relative to the active site of Pol II (which is denoted as position +1).

References

- Andersson R, Andersen PR, Valen E, Core LJ, Bornholdt J, Boyd M, Jensen TH, Sandelin A. 2014. Nuclear stability and transcriptional directionality separate functionally distinct RNA species. *Nat Commun* 5: 5336.
- Arner E, Daub CO, Vitting-Seerup K, Andersson R, Lilje B, Drablos F, Lennartsson A, Ronnerblad M, Hrydziuszko O, Vitezic M, et al. 2015. Transcribed enhancers lead waves of coordinated transcription in transitioning mammalian cells. *Science* 347: 1010–1014.
- Aw JGA, Shen Y, Wilm A, Sun M, Lim XN, Boon K-L, Tapsin S, Chan Y-S, Tan C-P, Sim AYL, et al. 2016. In Vivo Mapping of Eukaryotic RNA Interactomes Reveals Principles of Higher-Order Organization and Regulation. *Mol Cell* 62: 603–617.
- Bose DA, Donahue G, Reinberg D, Shiekhattar R, Bonasio R, Berger SL. 2017. RNA Binding to CBP Stimulates Histone Acetylation and Transcription. *Cell* 168: 135-149.e22.
- Carullo NVN, III RAP, Simon RC, Soto SAR, Hinds JE, Salisbury AJ, Revanna JS, Bunner KD, Ianov L, Sultan FA, et al. 2020. Enhancer RNAs predict enhancer–gene regulatory links and are critical for enhancer function in neuronal systems. *Nucleic Acids Res* 48: 9550–9570.
- Cheng B, Price DH. 2008. Analysis of factor interactions with RNA polymerase II elongation complexes using a new electrophoretic mobility shift assay. *Nucleic Acids Res* 36: e135–e135.
- DeSanta F, Natoli G. 2010. A Large Fraction of Extragenic RNA Pol II Transcription Sites Overlap Enhancers. *PLoS Biol* 1–17.
- Duttke SH, Chang MW, Heinz S, Benner C. 2019. Identification and dynamic quantification of regulatory elements using total RNA. *Genome Res* 29: 1836–1846.
- Hah N, Murakami S, Nagari A, Danko CG, Kraus WL. 2013. Enhancer transcripts mark active estrogen receptor binding sites. *Genome Res* 23: 1210–1223.
- Hsieh C-L, Fei T, Chen Y, Li T, Gao Y, Wang X, Sun T, Sweeney CJ, Lee G-SM, Chen S, et al. 2014. Enhancer RNAs participate in androgen receptor-driven looping that selectively enhances gene activation. *Proc Natl Acad Sci USA* 111: 7319–7324.
- Kaikkonen MU, Spann NJ, Heinz S, Romanoski CE, Allison KA, Stender JD, Chun HB, Tough DF, Prinjha RK, Benner C, et al. 2013. Remodeling of the Enhancer Landscape during Macrophage Activation Is Coupled to Enhancer Transcription. *Mol Cell* 51: 310–325.

- Kim T-K, Hemberg M, Gray JM, Costa AM, Bear DM, Wu J, Harmin DA, Laptewicz M, Barbara-Haley K, Kuersten S, et al. 2010. Widespread transcription at neuronal activity-regulated enhancers. *Nature* 465: 182–187.
- Kim T-K, Shiekhattar R. 2015. Architectural and Functional Commonalities between Enhancers and Promoters. *Cell* 162: 948–959.
- Lam MTY, Cho H, Lesch HP, Gosselin D, Heinz S, Tanaka-Oishi Y, Benner C, Kaikkonen MU, Kim AS, Kosaka M, et al. 2013. Rev-Erbs repress macrophage gene expression by inhibiting enhancer-directed transcription. *Nature* 498: 511–515.
- Li W, Notani D, Ma Q, Tanasa B, Nunez E, Chen AY, Merkurjev D, Zhang J, Ohgi K, Song X, et al. 2013. Functional roles of enhancer RNAs for oestrogen-dependent transcriptional activation. *Nature* 498: 516–520.
- Lu Z, Zhang QC, Lee B, Flynn RA, Smith MA, Robinson JT, Davidovich C, Gooding AR, Goodrich KJ, Mattick JS, et al. 2016. RNA Duplex Map in Living Cells Reveals Higher-Order Transcriptome Structure. *Cell* 165: 1267–1279.
- Lyskov S, Chou F-C, Conchúir SÓ, Der BS, Drew K, Kuroda D, Xu J, Weitzner BD, Renfrew PD, Sripakdeevong P, et al. 2013. Serverification of Molecular Modeling Applications: The Rosetta Online Server That Includes Everyone (ROSIE). *Plos One* 8: e63906.
- Missra A, Gilmour DS. 2010. Interactions between DSIF (DRB sensitivity inducing factor), NELF (negative elongation factor), and the Drosophila RNA polymerase II transcription elongation complex. *Proc Natl Acad Sci USA* 107: 11301–6.
- Mustoe AM, Busan S, Rice GM, Hajdin CE, Peterson BK, Ruda VM, Kubica N, Nutiu R, Baryza JL, Weeks KM. 2018. Pervasive Regulatory Functions of mRNA Structure Revealed by High-Resolution SHAPE Probing. *Cell* 173: 181–195.
- Rahman S, Zorca CE, Traboulsi T, Noutahi E, Krause MR, Mader S, Zenklusen D. 2016. Single-cell profiling reveals that eRNA accumulation at enhancer–promoter loops is not required to sustain transcription. *Nucleic Acids Res* 45: gkw1220.
- Rahnamoun H, Lee J, Sun Z, Lu H, Ramsey KM, Komives EA, Lauberth SM. 2018. RNAs interact with BRD4 to promote enhanced chromatin engagement and transcription activation. *Nat Struct Mol Biol* 25: 687–697.
- Sartorelli V, Lauberth SM. 2020. Enhancer RNAs are an important regulatory layer of the epigenome. *Nat Struct Mol Biol* 27: 521–528.
- Schaukowitch K, Joo J-Y, Liu X, Watts JK, Martinez C, Kim T-K. 2014. Enhancer RNA Facilitates NELF Release from Immediate Early Genes. *Mol Cell* 56: 29–42.

- Schwalb B, Michel M, Zacher B, Frühauf K, Demel C, Tresch A, Gagneur J, Cramer P. 2016. TT-seq maps the human transient transcriptome. *Science* 352: 1225–1228.
- Selth LA, Sigurdsson S, Svejstrup JQ. 2010. Transcript Elongation by RNA Polymerase II. *Annu Rev Biochem* 79: 271–293.
- Shechner DM, Hacısüleyman E, Younger ST, Rinn JL. 2015. CRISPR Display: A modular method for locus-specific targeting of long noncoding RNAs and synthetic RNA devices in vivo. *Nat Methods* 12: 664–670.
- Shii L, Song L, Maurer K, Zhang Z, Sullivan KE. 2017. SERPINB2 is regulated by dynamic interactions with pause-release proteins and enhancer RNAs. *Mol Immunol* 88: 20–31.
- Sigova AA, Abraham BJ, Ji X, Molinie B, Hannett NM, Guo YE, Jangi M, Giallourakis CC, Sharp PA, Young RA. 2015. Transcription factor trapping by RNA in gene regulatory elements. *Science* 350: 978–981.
- Smola MJ, Calabrese M, Weeks KM. 2015. Detection of RNA–Protein Interactions in Living Cells with SHAPE. *Biochemistry* 54: 6867–75.
- Smola MJ, Weeks KM. 2018. In-cell RNA structure probing with SHAPE-MaP. *Nat Protoc* 13: 1181–1195.
- Sugimoto Y, Vigilante A, Darbo E, Zirra A, Militti C, D’Ambrogio A, Luscombe NM, Ule J. 2015. hiCLIP reveals the in vivo atlas of mRNA secondary structures recognized by Staufen 1. *Nature* 519: 491–494.
- Vos SM, Farnung L, Urlaub H, Cramer P. 2018. Structure of paused transcription complex Pol II-DSIF-NELF. *Nature* 560: 601–606.
- Watkins AM, Rangan R, Das R. 2020. FARFAR2: Improved De Novo Rosetta Prediction of Complex Global RNA Folds. *Structure* 28: 963–976.e6.

REVIEWERS' COMMENTS

Reviewer #1 (Remarks to the Author):

Overall, this paper improved significantly after the incorporation of changes. The Cross-linker part was also properly demonstrated now. Although I would like to see the supplementary figure 4, d a little more detailed. In figure S4, d mass spec data should show the cross-linked fragments (for ex., UU+b/y ions). However, I find this paper is acceptable now.

Reviewer #2 (Remarks to the Author):

The authors have addressed all my concerns and I recommend publication of this manuscript. However, I do have another comment on the structural analysis that may be beneficial to the future studies on the eRNAs. In the past few years, chemical tools other than 1M7 has been developed to probe RNA flexibility/accessibility, such as DMS, NAI and NAI-N3. These tools have much better cell permeability and reactivity and are likely to solve the problem the authors encountered in the revision (see ref. here: Lee et al. 2017 RNA. <https://rnajournal.cshlp.org/content/23/2/169.short>).

Reviewer #3 (Remarks to the Author):

All of the major points were sufficiently addressed to resolve the concerns. The revised data and the manuscript are of high quality. I appreciate the authors' effort in compiling the response and improving the manuscript. All of my requested experiments and analysis were performed in high quality. Description of the methods and figures are presented with added clarity.

Response to my previous major point #5 regarding eRNA expression kinetics depending on the quantification method gave a mixed result compared to the original manuscript. Authors appropriately removed that claim from the data, but recruited existing literature to continue supporting the idea. I do not find this modification affects the overall impact of the manuscript significantly.

Response to Reviewers

Reviewer #1 (Remarks to the Author):

Overall, this paper improved significantly after the incorporation of changes. The Cross-linker part was also properly demonstrated now. Although I would like to see the supplementary figure 4, d a little more detailed. In figure S4, d mass spec data should show the cross-linked fragments (for ex., UU+b/y ions). However, I find this paper is acceptable now.

Reply: We thank the reviewer for her/his positive comments. Regarding the remaining comment of the reviewer, we want to draw the attention to the wording on top of the shown spectra. It indicates the nature of the crosslinked nucleotide.

Reviewer #2 (Remarks to the Author):

The authors have addressed all my concerns and I recommend publication of this manuscript. However, I do have another comment on the structural analysis that may be beneficial to the future studies on the eRNAs. In the past few years, chemical tools other than 1M7 has been developed to probe RNA flexibility/accessibility, such as DMS, NAI and NAI-N3. These tools have much better cell permeability and reactivity and are likely to solve the problem the authors encountered in the revision (see ref. here: Lee et al. 2017 RNA. <https://rnajournal.cshlp.org/content/23/2/169.short>).

Reply: We thank the reviewer for her/his positive comments. We are aware of recent developments in chemical RNA structure probing and will certainly consider these new chemicals in future attempts.

Reviewer #3 (Remarks to the Author):

All of the major points were sufficiently addressed to resolve the concerns. The revised data and the manuscript are of high quality. I appreciate the authors' effort in compiling the response and improving the manuscript. All of my requested experiments and analysis were performed in high quality. Description of the methods and figures are presented with added clarity.

Response to my previous major point #5 regarding eRNA expression kinetics depending on the quantification method gave a mixed result compared to the original manuscript. Authors appropriately removed that claim from the data, but recruited existing literature to continue supporting the idea. I do not find this modification affects the overall impact of the manuscript significantly.

Reply: We thank the reviewer for her/his positive comments.